# d²p: Structured Soft Attention Is All You Need

**Casey Sumagaysay Mogilevsky** [1]    **Kimberly Liang** [2]

## Abstract

Classical dynamic programming algorithms such as Smith–Waterman, edit distance, and constituency parsing solve structured combinatorial problems using hard constraints. Soft relaxations replace hard $\max$ or $\min$ operators with differentiable probabilistic models whose gradients correspond to alignment or parsing marginals, but existing approaches typically treat the DP algorithm itself as fixed, relying on hand-tuned gap penalties, edit costs, span penalties, and temperatures. We show how to learn these parameters directly from data: the marginals these algorithms produce are themselves first-order derivatives, so learning them is inherently a second-order problem. We derive efficient Hessian–vector products and cross-Jacobians for twelve dynamic programming algorithms spanning alignment, edit distance, and parsing; both derivative families admit closed-form covariance expressions under the induced Gibbs distribution. We implement these operators as fused CUDA kernels in d²p, achieving $100$–$20{,}000\times$ speedups over standard PyTorch and making end-to-end parameter learning practical at modern scales. We then demonstrate that learning these parameters is critical in practice. In protein structure alignment, freezing gap penalties collapses performance from $0.74$ to $0.32$ $F_1$ (and to $0.13$ at lower encoder capacity), while jointly learning them recovers biologically meaningful gap regimes and reaches $0.75$ $F_1$ and $0.445$ lDDT, $91\%$ of the TM-align lDDT ceiling. The same machinery transfers to constituency parsing, where a structured CKY CRF matches dense per-span supervision to within $0.003$ $F_1$ on English with no direct parse-tree structural supervision.

## 1. Introduction

Dynamic programming algorithms for alignment, edit distance, and parsing are usually understood as combinatorial procedures. They encode hard constraints (ordering and gap structure for alignment, grammatical consistency for parsing) that carve out a space of feasible solutions and return the optimum within it in polynomial time. In this viewpoint the algorithm is a fixed piece of inference machinery, and learning, when it happens, happens elsewhere: in the scores fed into it and the outputs it produces, while the DP algorithm itself remains a constant.

This view is incomplete in a way that turns out to matter for modern deep learning, because dynamic programming algorithms can be viewed as the zero-temperature limit of structured attention. An alignment is a correspondence: which elements of two sequences match. By analogy, a parse is a correspondence: which spans of one sequence are constituents. The soft, finite-temperature version of either, a matrix whose $(i, j)$ entry scores that pairing or bracketing, is exactly an attention map. Standard attention (Vaswani et al., 2017) places an unconstrained soft correspondence on every query–key pair, each query free to attend anywhere; a dynamic programming algorithm places the same kind of correspondence but restricts it to the valid global structures its recurrence admits: monotone and gap-respecting for pairwise alignment, grammatical and hierarchical for parsing. That restriction is the only difference between them, and it is what makes the attention *structured*.

This identity is not merely semantic. In classical hard form, each algorithm returns an optimal structure $Y^* = \arg\max_{Y \in \mathcal{Y}} \langle Y, S \rangle$ (the Smith–Waterman alignment, the optimal edit sequence, the CKY parse), and by Danskin's theorem (Danskin, 1966) this argmax is exactly the gradient $\partial V^*/\partial S = Y^*$ of the value $V^*(S) = \max_Y \langle Y, S \rangle$: the forward-pass-plus-traceback of classical DP is reverse-mode autodiff on the max-network (Eisner, 2016), the traceback being the gradient by another name. The finite-temperature relaxation replaces the $\max$ with a log-sum-exp, $V_T(S) = T \log \sum_Y \exp(\langle Y, S \rangle / T) = T \log Z$, whose gradient is the posterior marginal under the Gibbs distribution $p_T(Y) \propto \exp(\langle Y, S \rangle / T)$,

$$\frac{\partial V_T}{\partial S_{ij}} = \Pr_{p_T}[(i, j) \in Y] = P_{T,ij}, \quad (1)$$

[1]Orikata Bio PBC [2]Independent Researcher. Correspondence to: Casey Sumagaysay Mogilevsky <casey@orikata.ai>, Kimberly Liang <kimberly.yx.liang@gmail.com>.

*Proceedings of the 43ʳᵈ International Conference on Machine Learning*, Seoul, South Korea. PMLR 306, 2026. Copyright 2026 by the author(s).

**Temperature Effects in Structured and Unstructured Attention**

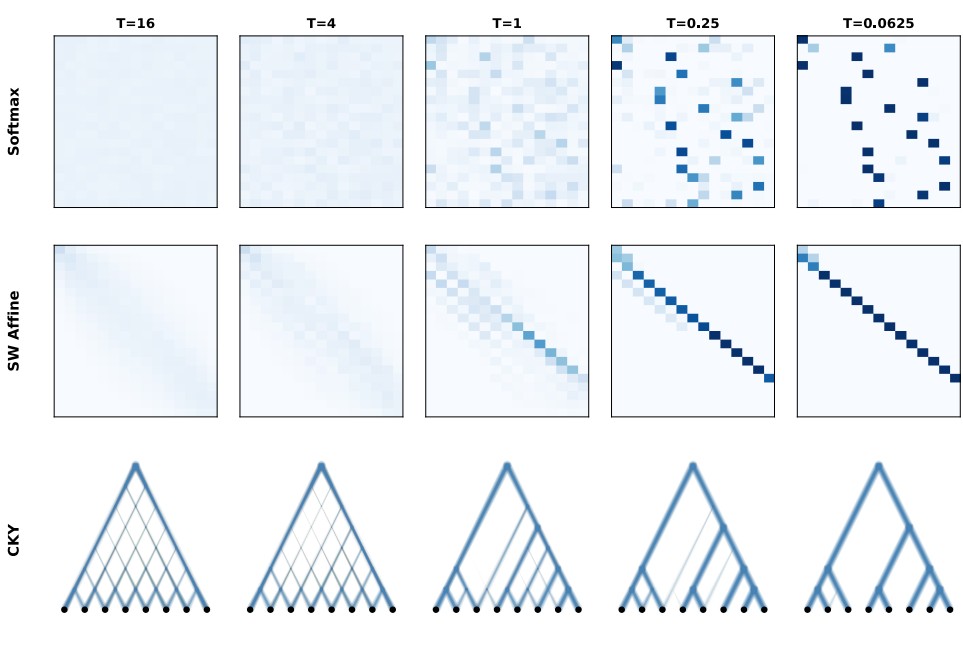

← Higher Temperature (high entropy)   Lower Temperature (low entropy) →

*Figure 1.* **Structured attention is temperature-controlled.** Each panel shows soft marginals (attention weights) at temperature $T$, decreasing from left ($T = 16$) to right ($T = 0.0625$), for three attention mechanisms: unstructured row-softmax (top), Smith–Waterman with affine gaps (middle), and CKY constituency parsing (bottom). As $T \to \infty$ (left) the weights relax toward the structure-independent prior $P_\infty$, the uniform edge frequency under $\mathcal{Y}$; as $T \to 0^+$ (right) they concentrate on the optimal structure $P^* = Y^*$, recovering the classical hard algorithm (the Smith–Waterman traceback in the middle row, the Viterbi parse in the bottom row). The structured rows preserve their constraints, monotone gap-respecting paths and nested spans, at every temperature, whereas softmax has none to preserve. The same matrix is the attention map a downstream loss is computed on; learning its structure end-to-end is the subject of this paper.

which is the structured attention matrix itself, the soft analog of $Y^*$: one identity at two temperatures (Figure 1), collapsing onto the optimal structure as $T \to 0^+$ and relaxing to a structure-independent prior $P_\infty$ as $T \to \infty$. Pair-HMM forward–backward, SCFG inside–outside, and temperature-scaled alignment are its classical instances; soft-DTW, structured attention, SparseMAP, differentiable DP, torch-struct, and differentiable Smith–Waterman its modern ones. Once the algorithm is recognized as attention rather than fixed inference, the gap penalties, edit costs, and temperatures that define it are not constants to set by hand but bona fide parameters to be learned.

Existing differentiable formulations train the encoder that produces $S$ but leave the program's own parameters fixed, for a structural reason: first-order autodiff through the program returns the marginals, which *are* the gradient $P_T = \partial V_T / \partial S$, so training an upstream encoder needs nothing more. But the marginals are the model's output, the attention matrix a loss is computed on; descending that loss, through the encoder or the program's own parameters, means differentiating the marginals, a second-order object (Table 1). Classical hard DP stops at the traceback and soft

DP at the marginals, both at a first derivative; d²p continues to the second: Hessian–vector products $\partial P_T / \partial S$ train the encoder, and cross-Jacobians $\partial P_T / \partial \boldsymbol{\eta}$ learn the hyperparameters $\boldsymbol{\eta}$ (gaps, edit costs, span weights, temperature), the gradient that would tune the ubiquitous $g{=}{-}10$ but was never available in closed form. Mensch & Blondel (2018) gave the score–score Hessian for Viterbi and DTW, and CherryML (Prillo et al., 2023) learns rate matrices by autodiff, but neither gives the cross-Jacobians that condition the structure space itself; we derive both, as Gibbs covariances in closed form, for all twelve algorithms.

These second-order quantities are not optional. Freezing Smith–Waterman's single gap penalty at the bioinformatics default ($g{=}{-}10$) collapses alignment $F_1$ from 0.74 to 0.32 at full encoder capacity, and to 0.13 at lower capacity, while learning it through the cross-Jacobian recovers full performance; the affine variant, with an extend penalty to fall back on, pays a far smaller price ($0.75 \to 0.71$), itself diagnostic that the learned penalty matters most where the algorithm is least expressive.

The recursion relation is itself a prior, and whether it is the

*right* one should matter more than the capacity upstream of it. We sweep encoder capacity from a fixed substitution matrix to a 128-dimensional neural network encoder, crossed against the right prior (Smith–Waterman), no prior (row-softmax cross-attention), and a deliberately wrong prior (DTW, monotone warping without gaps): at full capacity they reach $F_1 \approx 0.75$, $0.69$, and $0.05$, and at zero capacity $0.51$, $< 0.01$, and $0.05$. Capacity can substitute for absent structure but cannot repair the wrong one.

This sharpens Sutton's bitter lesson (Sutton, 2019): *hand-tuned* structure loses to scale, and loses most where data are abundant enough for scale to learn the structure itself, as in much of natural language. $\text{d}^2\text{p}$ fixes only the structure implicit in the attention computation, learning every parameter inside it, so its one structural commitment is exactly the one capacity cannot recover. The lesson is least bitter in the opposite regime: domains with rich structure and scarce, expensive data, the natural sciences chief among them, where the right prior is not a crutch to discard once data are abundant but a durable substitute for data that are difficult or costly to obtain.

Our two domains sit at opposite ends of this spectrum: protein structure alignment is structure-decisive, where the right recursion relation outweighs encoder capacity, while constituency parsing is data-abundant, where scaled language models parse better than our small encoders, so structure earns its place not through raw accuracy but through supervision efficiency (matching dense span labels from the gold tree's score alone) and an interpretable parse.

**Contributions.**

1. **Complete derivative hierarchy.** Because the attention matrix is itself a first derivative of the log-partition, learning through it is a second-order problem. We derive the full hierarchy, Hessian–vector products for encoder training and cross-Jacobians for every DP hyperparameter, all as Gibbs covariances under the induced distribution (Sections 3 and 5).

2. **Cross-Jacobian algorithm.** We give the U+W two-pass algorithm for computing $\partial P_T / \partial \boldsymbol{\eta}$ at the same asymptotic cost as the backward pass, instantiated for all twelve algorithms (Sections 5 and 6).

3. **Unified family.** We cast pairwise alignment, edit distance, and parsing as structured soft attention across twelve algorithms, extending structured-attention networks (Kim et al., 2017) and differentiable DP (Mensch & Blondel, 2018): grid algorithms compute cross-attention between two sequences, chart algorithms self-attention within one, with classical hard DP recovered as $T \to 0$ and pair HMMs and SCFGs as the finite-temperature instances (Section 3).

4. **Implementation.** We release $\text{d}^2\text{p}$ (https://github.com/caseysm/d2p), an open-source PyTorch library with fused CUDA kernels for forward, backward, HVP, and cross-Jacobian passes across all twelve algorithms, achieving $100$–$20{,}000\times$ speedups over standard PyTorch and $100$–$1000\times$ over torch-struct (Section 6).

5. **Applications.** On protein structure alignment, encoders trained through $\text{d}^2\text{p}$ reach $0.75$ $F_1$, structure substitutes for encoder capacity, and learned gap penalties recover biological priors; on parsing, a structured CKY CRF supervised only by the gold tree's score comes within $0.003$ $F_1$ of dense per-span supervision on English and exceeds it on three of nine domains spanning natural language and code (Sections 7 and H).

Section 2 reviews the temperature-scaled DP operator; Section 3 develops the first-order framework and its structured-attention interpretation; Section 4 establishes regularity; Section 5 derives the second-order hierarchy and the U+W algorithm; Section 6 instantiates Smith–Waterman with affine gaps as a running example, with the remaining eleven algorithms in Sections A to C; Section 7 presents the protein alignment, Section H the parsing transfer.

**Related work.** Differentiable dynamic programming has a long lineage: pair HMMs with forward–backward and SCFGs/PCFGs with inside–outside are classical differentiable models of alignment and parsing, probabilistic and temperature-scaled interpretations of alignment scoring date back decades and remain active (Kim et al., 1994; Bucher & Hofmann, 1996; Frith, 2020; Reifenrath et al., 2025), and DP gradients have long been identified with posterior marginals (Eisner, 2016). EM for continuous-time Markov chains (Holmes & Rubin, 2002) and autodiff estimation of rate matrices (Prillo et al., 2023) learn substitution parameters from these expectations, while soft-DTW (Cuturi & Blondel, 2017), structured attention (Kim et al., 2017), differentiable DP (Mensch & Blondel, 2018), torch-struct (Rush, 2020), SparseMAP (Niculae et al., 2018), and differentiable Smith–Waterman (Petti et al., 2023) bring these relaxations into learning pipelines. All expose the first-order marginals; our contribution is the second-order layer they do not derive or implement: Hessian–vector products and parameter cross-Jacobians that learn a program's own gap, edit, span, and temperature parameters end-to-end, as fused CUDA kernels across twelve algorithms.

**Notation.** $S$ is the score input (a pairwise matrix $S \in \mathbb{R}^{n \times m}$ for alignment; a score tensor over edges or spans in general) and $\boldsymbol{\eta}$ the DP hyperparameters (gap penalties $g_o, g_e$; edit costs $c_{\text{ins}}, c_{\text{del}}, c_{\text{sub}}$; span penalty $\lambda$; temperature $T$). The DP graph is a DAG $G = (V, E)$ where $\text{pa}(i)$ is

*Table 1.* The derivative hierarchy of temperature-softened dynamic programming. First-order quantities are computed by classical inference and prior differentiable-DP work; the second-order quantities are the contribution of this work, and are the mechanism by which encoder parameters and the algorithm's own hyperparameters are learned end-to-end. First-order derivatives are Gibbs expectations, second-order ones covariances, under $p_T$. The score–score Hessian was given for Viterbi and DTW by Mensch & Blondel (2018).

| Order | Derivative | Quantity (interpretation) | Pass | Status |
|---|---|---|---|---|
| 0 | $V_T$ | DP value (objective) | Fwd | classical |
| 1 | $\partial V_T/\partial S = P_T$ | marginals (attention matrix) | Bwd | classical |
| 1 | $\partial V_T/\partial \boldsymbol{\eta}$ | expected sufficient statistics | Bwd | classical |
| 1 | $\partial V_T/\partial T$ | $H(p_T)$ (max), $-H(q_T)$ (min) | Bwd | classical |
| 2 | $\partial P_T/\partial S$ | score–score Hessian (HVP) | Fwd+Bwd | Mensch & Blondel (2018) |
| 2 | $\partial P_T/\partial \boldsymbol{\eta}$ | score–parameter cross-Jacobian | U+W | **this work** |
| 2 | $\partial P_T/\partial T$ | score–temperature cross-Jacobian | U+W | **this work** |

the predecessor (parent) set of node $i$, $L_{\max}$ the longest-path length, and $D_{\max}$ the maximum in-degree; a structure $Y \in \mathcal{Y}$ is a binary edge-incidence vector scoring $\langle Y, S \rangle$, treated as a random vector under $p_T$ in the covariance identities below. We write $V^*$ and $V_T = T \log Z$ for the hard and soft value, $Y^*$ for the optimizer, $p_T$ for the Gibbs distribution, $P_T$ for its marginals (the attention matrix, with limits $P^* = Y^*$ as $T \to 0^+$ and $P_\infty$ as $T \to \infty$), $\alpha_i, \beta_i$ for forward/backward values, and $w_{ji}$ for the backward softmax transition weights. The two second-order objects are the HVP $\partial P_T/\partial S$ and the cross-Jacobian $\partial P_T/\partial \boldsymbol{\eta}$.

## 2. Background

### 2.1. Temperature-Scaled Log-Sum-Exp

The key to smoothing maximization DP is the following operator.

**Definition 2.1** (Temperature-Scaled Log-Sum-Exp)**.** For $T > 0$ and $v_1, \ldots, v_K \in \mathbb{R}$,

$$\text{LSE}_T(v_1, \ldots, v_K) = T \log \sum_{k=1}^{K} \exp(v_k/T).$$

In the notation of Mensch & Blondel (2018), $\text{LSE}_T$ is the regularized maximum $\max_\Omega$ with $\Omega = -T \cdot H$ ($H$ the Shannon entropy).

**Lemma 2.2** (Properties of $\text{LSE}_T$)**.** *The operator* $\text{LSE}_T$ *satisfies:*

1. *Limit:* $\lim_{T \to 0^+} \text{LSE}_T(v) = \max_k v_k$.

2. *Convexity:* $\text{LSE}_T$ *is convex and everywhere differentiable.*

3. *Gradient:* $\nabla \text{LSE}_T(v) = \text{softmax}(v/T)$.

4. *Hessian:* $\nabla^2 \text{LSE}_T(v) = \frac{1}{T}(\text{Diag}(p) - pp^\top)$ *where* $p = \text{softmax}(v/T)$, *which is positive semidefinite with operator norm at most* $1/T$.

All operations are computed in log-space with max-subtraction for numerical stability (Section I.11). As $T \to \infty$ the Gibbs distribution approaches uniform (marginals $\to$ prior edge frequencies) and as $T \to 0^+$ it concentrates on the optimum (marginals $\to$ indicators, splitting uniformly among ties); empty and degenerate inputs are the obvious limits.

### 2.2. Dynamic Programming on DAGs

**Definition 2.3** (DP Graph)**.** A DP graph is a finite DAG $G = (V, E)$ with a unique source $s$, a unique sink $t$, and edge weights $S_e \in \mathbb{R}$ for each $e \in E$.

Let $\mathcal{Y}$ denote the set of all $s \to t$ paths, each identified with its binary edge-incidence vector $Y \in \{0,1\}^{|E|}$. The *hard* DP objective is

$$V^*(S) = \max_{Y \in \mathcal{Y}} \langle Y, S \rangle,$$

computed in $O(|E|)$ time by the Bellman recursion $v_i = \max_{j \in \text{pa}(i)}(v_j + S_{ji})$ with $v_s = 0$.

*Remark* 2.4 (Generalization to Hypergraphs)**.** While this section presents the path-DAG formulation, the framework extends to hypergraph (tree-structured) DP such as CKY parsing and Eisner's algorithm. In these cases, $\mathcal{Y}$ represents parse trees rather than paths, and the Bellman recursion aggregates over hyperedges (span combinations). The gradient and Hessian identities transfer directly; we instantiate both path and tree algorithms in Section 6 with full derivations in the appendix.

## 3. Differentiable Dynamic Programming

We first present the framework (Figure 2 shows the end-to-end pipeline, instantiated for protein alignment in Section 7) for maximization problems such as alignment and parsing, and describe minimization problems like edit distance later in Section 3.5.

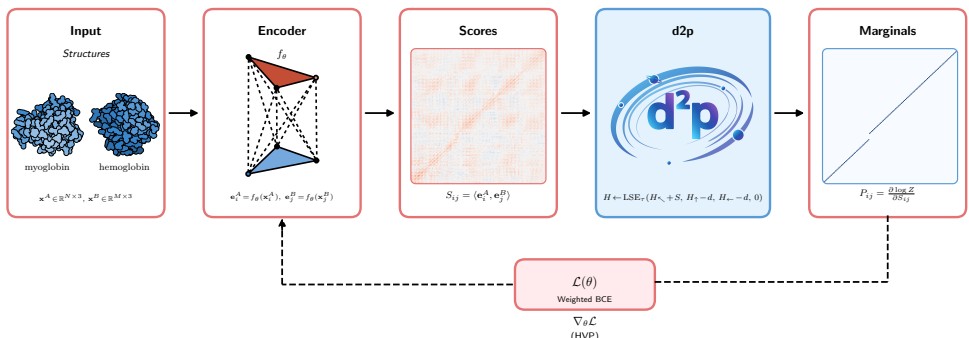

*Figure 2.* **End-to-end differentiable protein structure alignment with d²p.** Two protein structures are encoded by a neural network (ProteinMPNN, GVP, or IPA) to produce per-residue embeddings $h_i^{(1)}, h_j^{(2)} \in \mathbb{R}^d$. Inner products $S_{ij} = \langle h_i^{(1)}, h_j^{(2)} \rangle$ form a pairwise similarity matrix. The differentiable Smith–Waterman algorithm computes soft alignment marginals $P_{T,ij}$ (the posterior probability that residues $i$ and $j$ are aligned) via temperature-scaled dynamic programming. Gradients flow through the entire pipeline, enabling end-to-end learning of encoder parameters against TM-align ground truth.

### 3.1. Soft Bellman Recursion

We fix $T > 0$ and replace $\max$ with $\text{LSE}_T$ in the Bellman recursion:

$$\alpha_i = \text{LSE}_T\big(\{\alpha_j + S_{ji}\}_{j \in \text{pa}(i)}\big), \quad \alpha_s = 0. \quad (2)$$

We define the *soft DP value* as $V_T(S) = \alpha_t$.

### 3.2. Log-Partition Function

**Theorem 3.1** (Log-Partition Representation).

$$V_T(S) = T \log \sum_{Y \in \mathcal{Y}} \exp\big(\langle Y, S \rangle / T\big).$$

*Proof.* By induction over a topological ordering of $G$. At each node $i$, the $\text{LSE}_T$ aggregates partial path sums via exponentiation, matching the partition function over all $s \to i$ paths. □

$V_T$ therefore represents the log-partition function of a Gibbs distribution $p_T(Y) \propto \exp(\langle Y, S \rangle / T)$ over the path polytope, the same exponential family structure underlying conditional random fields (Lafferty et al., 2001; Sutton & McCallum, 2012) and graphical models more broadly (Wainwright & Jordan, 2008).

### 3.3. Gradient as Expected Path

**Theorem 3.2** (Marginal Gradient). $\nabla_S V_T(S) = \mathbb{E}_{Y \sim p_T}[Y]$.

This follows from Danskin's theorem (Danskin, 1966): differentiating a hard maximum yields an optimal structure (under the usual uniqueness/subgradient convention),

while differentiating the smoothed maximum yields the Gibbs marginal over structures. Each coordinate gives the *marginal probability* that edge $e$ is used with $\frac{\partial V_T}{\partial S_e} = \Pr_{p_T}[e \in Y]$.

**Backpropagation algorithm.** Following Mensch & Blondel (2018), the gradient can be computed in $O(|E|)$ by a backward pass through the DAG. We define the softmax transition weights

$$w_{ji} = \frac{\exp((\alpha_j + S_{ji})/T)}{\sum_{j' \in \text{pa}(i)} \exp((\alpha_{j'} + S_{j'i})/T)} \quad (3)$$

at each node $i$, using the forward values $\alpha$. We then initialize $\beta_t = 1$ and propagate in reverse topological order:

$$\beta_j \mathrel{+}= \beta_i \cdot w_{ji} \quad \text{for each successor } i \text{ of } j. \quad (4)$$

Then $\frac{\partial V_T}{\partial S_{ji}} = \beta_i \cdot w_{ji}$.

**Lemma 3.3** (Covariance Identity for Gibbs Distributions). *Let $p_T(Y) \propto \exp(\text{score}(Y; \eta)/T)$ be a Gibbs distribution over structures $Y$, with log-partition $V_T = T \log Z$. For any event $A \subseteq \mathcal{Y}$ and parameter $\eta$:*

$$\frac{\partial P_T(A)}{\partial \eta} = \frac{1}{T} \text{Cov}_{p_T}\left(\mathbf{1}_A, \frac{\partial \text{score}}{\partial \eta}\right). \quad (5)$$

*Proof.* Let $\varphi_\eta(Y) = \partial \text{score}(Y; \eta)/\partial \eta$ be the sufficient statistic. The standard log-partition derivative is $\partial_\eta p_T(Y) = \frac{1}{T} p_T(Y)(\varphi_\eta(Y) - \mathbb{E}_{p_T}[\varphi_\eta])$, so $\partial_\eta P_T(A) = \sum_{Y \in A} \partial_\eta p_T(Y) = \frac{1}{T}(\mathbb{E}_{p_T}[\mathbf{1}_A \varphi_\eta] - P_T(A) \mathbb{E}_{p_T}[\varphi_\eta]) = \frac{1}{T} \text{Cov}_{p_T}(\mathbf{1}_A, \varphi_\eta)$. □

This is the *fluctuation-response relation* from statistical mechanics where the sensitivity of an observable (marginal

probability) to a parameter equals the covariance between that observable and the parameter's sufficient statistic, scaled by temperature. All cross-Jacobian formulas in this paper are instantiations of this identity.

### 3.4. Hessian–Vector Products

The Hessian $\nabla^2 V_T(S)$ is the gradient of the marginals with respect to scores, needed whenever we optimize a loss that depends on the soft predicted structure.

**Theorem 3.4** (Covariance Interpretation).

$$\nabla^2_{S,S} V_T(S) = \frac{1}{T}\mathrm{Cov}_{p_T}(Y, Y).$$

The Hessian is an $|E| \times |E|$ matrix (where $|E|$ is the problem dimension: $mn$, $n^2$, or $n^3$ depending on the algorithm) that is too large to form explicitly. When scores form a matrix, the Hessian is a 4-dimensional tensor, where the "vector" $V$ is a matrix of the same shape as the scores, and the HVP is a tensor contraction. We compute $\nabla^2 V_T(S) \cdot V$ in $O(|E|)$ time using Pearlmutter's R-operator (Pearlmutter, 1994), following the approach of Mensch & Blondel (2018):

$$\nabla^2 V_T(S) \cdot V = \nabla_S \langle \nabla_S V_T(S), V \rangle. \tag{6}$$

This requires a forward tangent pass propagating $\dot{\alpha}_i = \sum_{j \in \mathrm{pa}(i)} w_{ji} \cdot (\dot{\alpha}_j + V_{ji})$, followed by a backward tangent pass propagating $\dot{\beta}_j \mathrel{+}= \dot{\beta}_i \cdot w_{ji} + \beta_i \cdot \dot{w}_{ji}$ where $\dot{w}_{ji} = w_{ji}(\dot{\alpha}_j + V_{ji} - \dot{\alpha}_i)/T$.

*Remark* 3.5 (HVP Correctness). Equation (6) follows from the identity $[\nabla^2 f(S) \cdot V]_i = \frac{d}{d\epsilon}\big|_{\epsilon=0}[\nabla f(S + \epsilon V)]_i$, which holds for any twice-differentiable $f$. The forward tangent $\dot{\alpha}$ computes $\frac{d\alpha}{d\epsilon}\big|_{\epsilon=0}$ under perturbation $S \mapsto S + \epsilon V$, and the backward tangent propagates this through the gradient computation. See Pearlmutter (1994) for the general derivation.

### 3.5. Softmin for Minimization Problems

Minimization problems (DTW, the edit distances) use the softmin $\mathrm{smin}_T(c) = -T\log\sum_k \exp(-c_k/T) = -\mathrm{LSE}_T(-c)$, the exact dual of $\mathrm{LSE}_T$ (Theorem J.2), with the Gibbs distribution $q_T(Y) \propto \exp(-\mathrm{cost}(Y)/T)$ favoring low-cost paths. The first-order marginal is unchanged, $P_{T,a} = \partial V_T/\partial c_a = \mathbb{E}_{q_T}[\mathbf{1}\{a \in Y\}]$, but the second-order objects acquire a sign flip: the Hessian $\nabla^2 V_T = -\frac{1}{T}\mathrm{Cov}_{q_T}(Y, Y) \preceq 0$ ($V_T$ concave) and the cross-Jacobians $\partial P_{T,a}/\partial\eta = -\frac{1}{T}\mathrm{Cov}_{q_T}(\mathbf{1}\{a \in Y\}, \varphi_\eta(Y))$, since raising a cost makes the structures that use it less likely.

## 4. Regularity Properties

The soft DP value function $V_T$ and its derivatives satisfy important regularity properties that ensure well-behaved

optimization. The value function is $L_{\max}$-Lipschitz in scores (where $L_{\max}$ is the maximum path length), and the gradient (marginals) is $(L_{\max}/T)$-Lipschitz. For maximization problems, $V_T$ is convex in scores with Hessian $\nabla^2 V_T = (1/T)\mathrm{Cov}_{p_T}(Y, Y) \succeq 0$; for minimization, it is concave. The soft value approximates the hard optimum with bounded error $O(T \cdot L_{\max} \cdot \log D_{\max})$, following Mensch & Blondel (2018). These properties give smoothness and curvature bounds for the score-to-marginal map, helping explain the stability of gradient-based optimization; they do not imply global convexity of downstream losses or of neural score parameterizations. Full theorem statements and proofs appear in Section E.

## 5. Mixed Second-Order Derivatives

Our core contribution is a unified framework for the cross-Jacobians of marginals with respect to algorithm-specific parameters.

### 5.1. Score–Score HVP

The score–score Hessian–vector product applies to *all* algorithms. For a score tensor $S$ (match scores, arc scores, etc.), the HVP $\nabla^2_S V_T \cdot V$ computes the action of the covariance matrix:

**Theorem 5.1** (Universal Score Covariance). *For any DP algorithm with score parameters $S$,*

$$\frac{\partial^2 V_T}{\partial S_a\, \partial S_b} = \frac{1}{T}\mathrm{Cov}_{p_T}(\mathbf{1}\{a \in Y\}, \mathbf{1}\{b \in Y\}),$$

*where $a, b$ index score entries and $\mathbf{1}\{a \in Y\}$ indicates that structure $Y$ uses score $S_a$.*

### 5.2. Score–Parameter Cross-Jacobians

Each algorithm has specific parameters beyond scores: gap penalties $(g_o, g_e)$ for alignment, insertion/deletion/transposition costs $(\mathrm{ins}, \mathrm{del}, \mathrm{trans})$ for edit distance, and temperature $T$ for all algorithms. We compute $\partial(\text{marginals})/\partial\eta$ for any such parameter $\eta$, again as a covariance.

**Theorem 5.2** (General Cross-Jacobian Covariance). *For any parameter $\eta$ with sufficient statistic $\varphi_\eta(Y)$ (the count of $\eta$-weighted transitions used by structure $Y$):*

$$\frac{\partial^2 V_T}{\partial S_a\, \partial\eta} = \frac{1}{T}\mathrm{Cov}_{p_T}(\mathbf{1}\{a \in Y\}, \varphi_\eta(Y)).$$

*Proof.* By equality of mixed partials, $\frac{\partial^2 V_T}{\partial S_a\, \partial\eta} = \frac{\partial}{\partial\eta}\left(\frac{\partial V_T}{\partial S_a}\right) = \frac{\partial P_{T,a}}{\partial\eta}$. Since $V_T$ is a log-partition function, its mixed second derivatives equal covariances of the corresponding sufficient statistics divided by $T$: $\frac{\partial^2 V_T}{\partial S_a\, \partial\eta} = \frac{1}{T}\mathrm{Cov}_{p_T}(\mathbf{1}\{a \in Y\}, \varphi_\eta(Y))$. $\square$

*Table 2.* Parameters and sufficient statistics for cross-Jacobians.

| Algorithm | Parameter $\eta$ | Statistic $\varphi_\eta(Y)$ |
|---|---|---|
| SW/NW Linear | gap $g$ | $N_{\text{gaps}}$ |
| SW/NW Affine | $g_o, g_e$ | $N_{\text{opens}}, N_{\text{extends}}$ |
| DTW | $\omega_d, \omega_h, \omega_v$ | $N_{\text{diag}}, N_{\text{horiz}}, N_{\text{vert}}$ |
| Levenshtein | ins, del | $N_{\text{ins}}, N_{\text{del}}$ |
| OSA/Damerau | trans | $N_{\text{trans}}$ |
| CKY | span $\lambda$ | $\sum (j - i)$ |
| Eisner | arc $\lambda$, bias $\delta$ | $\sum |h - d|, N_{\text{right}}$ |
| All | $T$ | $-\text{score}(Y)/T^2$ |

**Interpretation.** The cross-Jacobian $\partial P_{T,a}/\partial \eta$ measures how the marginal probability at position $a$ responds to changes in parameter $\eta$. Positions where the marginal co-varies positively with the $\eta$-count will increase when $\eta$ increases.

**Algorithm: Forward $U$ + Backward $W$.** The cross-Jacobian computation runs in three passes (forward, aggregation, and backward):

**Step 1: Forward $U$ pass.** Compute $U_i = \partial \alpha_i / \partial \eta$ by differentiating the forward recurrence. At each transition, add the direct contribution $\partial(\text{transition weight})/\partial \eta$ if the transition weight depends on $\eta$.

**Step 2: Aggregate.** Compute $\partial V_T / \partial \eta = \sum_i \beta_i^{(0)} \cdot U_i$.

**Step 3: Backward $W$ pass.** Initialize $W_i = \beta_i^{(0)} \cdot (U_i - \partial V_T / \partial \eta)/T$ and propagate backward using both the standard $\beta$ propagation and the tangent weights $\dot{w}$ induced by the $U$ values.

**Step 4: Output.** The cross-Jacobian at score position $a$ is $W_a$ (or appropriately projected for multi-state algorithms).

Table 2 lists the parameters and their sufficient statistics for all algorithms.

### 5.3. Score–Temperature Cross-Jacobian

The temperature cross-Jacobian is *universal*, applies to all algorithms and has a special entropy interpretation.

**Theorem 5.3** (Score–Temperature Covariance)**.**

$$\frac{\partial^2 V_T}{\partial S_a \, \partial T} = -\frac{1}{T^2} \text{Cov}_{p_T}\big(\mathbf{1}\{a \in Y\}, \text{score}(Y)\big).$$

*Proof.* We begin with the Gibbs distribution as $p_T(Y) = \exp((\text{score}(Y) - V_T)/T)$. Differentiating with respect to $T$ gives $\frac{\partial p_T(Y)}{\partial T} = \frac{p_T(Y)}{T^2}(\mathbb{E}(\text{score}) - \text{score}(Y))$. The marginal $P_{T,a} = \sum_{Y \ni a} p_T(Y)$, so $\frac{\partial P_{T,a}}{\partial T} = -\frac{1}{T^2}\text{Cov}_{p_T}(\mathbf{1}\{a\}, \text{score})$. $\square$

The $U$ update for temperature includes an entropy term at

*Table 3.* Twelve DP algorithms in the unified framework.

| Algorithm | States | Op | Parameters |
|---|---|---|---|
| *Alignment (Maximization)* | | | |
| SW Linear | 1 | $\text{LSE}_T$ | gap |
| SW Affine | 3 | $\text{LSE}_T$ | $g_o, g_e$ |
| NW Linear | 1 | $\text{LSE}_T$ | gap |
| NW Affine | 3 | $\text{LSE}_T$ | $g_o, g_e$ |
| LCS | 1 | $\text{LSE}_T$ | (none) |
| MAS | 1 | $\text{LSE}_T$ | (none) |
| *Edit Distance (Minimization)* | | | |
| DTW | 1 | $\text{smin}_T$ | $\omega_d, \omega_h, \omega_v$ |
| Levenshtein | 1 | $\text{smin}_T$ | ins, del |
| OSA | 1 | $\text{smin}_T$ | ins, del, trans |
| Damerau | 1 | $\text{smin}_T$ | ins, del, trans |
| *Parsing* $(O(n^3))$ | | | |
| CKY | chart | $\text{LSE}_T$ | span $\lambda$ |
| Eisner | 4 | $\text{LSE}_T$ | arc $\lambda$, bias $\delta$ |

each cell: $U_i \mathrel{+}= (\alpha_i - \bar{\alpha}_i)/T$, where $\bar{\alpha}_i$ is the weighted average of predecessor values.

### 5.4. Summary

Table 1 summarizes the full derivative hierarchy. The first-order quantities (marginals $\partial V_T / \partial S = P$, expected counts $\partial V_T / \partial \eta$, and entropy $\partial V_T / \partial T$) are computed by the backward pass; the score–score HVP $\nabla_S^2 V_T \cdot V$ by a forward sweep over the backward recursion (Section 3.4); and the cross-Jacobians $\partial P_T / \partial \eta$ and $\partial P_T / \partial T$ by the $U$+$W$ two-pass algorithm of Section 5.2. All run in $O(nm)$ (grid) or $O(n^3)$ (chart) via wavefront parallelism.

## 6. Algorithm Instantiations

We instantiate the framework in full on one running example, Smith–Waterman with affine gaps; the remaining eleven algorithms in Table 3 follow the same four-pass template (forward, marginals, HVP, cross-Jacobian) and are derived in Sections A to C.

### 6.1. Worked Example: Smith–Waterman with Affine Gaps

Affine-gap Smith–Waterman scores local alignments between sequences $X$ and $Y$ with pairwise score matrix $S \in \mathbb{R}^{n \times m}$, under hyperparameters $\boldsymbol{\eta} = (g_o, g_e)$ (gap-open and gap-extend, initialized negative but learned unconstrained) at temperature $T$. Its state sequences $(M/I/D)^*$ are in bijection with alignments: $M$ at $(i, j)$ is a match, $I$ a gap in $Y$ (insertion), $D$ a gap in $X$ (deletion). The gap-open penalty $g_o$ is paid once per contiguous gap block (on $M \to I$ or $M \to D$), $g_e$ on each extension ($I \to I$ or $D \to D$), yielding the standard affine penalty $g_o + (k - 1)g_e$ for a gap of

length $k$ (Gotoh, 1982). Local alignment can start anywhere, so the boundaries are $\alpha_{i,0}^M = \alpha_{0,j}^M = \alpha_{i,0}^I = \alpha_{0,j}^D = -\infty$, and the soft value $V_T = \text{LSE}_T(\{\alpha_{ij}^\sigma\}_{i,j,\sigma})$ sums over all states $\sigma \in \{M, I, D\}$. The three-state ($M/I/D$) soft recurrence is

$$\alpha_{ij}^M = \text{LSE}_T(\alpha_{i-1,j-1}^M, \alpha_{i-1,j-1}^I, \alpha_{i-1,j-1}^D, 0) + S_{ij}, \tag{7}$$

$$\alpha_{ij}^I = \text{LSE}_T(\alpha_{i-1,j}^M + g_o, \alpha_{i-1,j}^I + g_e), \tag{8}$$

$$\alpha_{ij}^D = \text{LSE}_T(\alpha_{i,j-1}^M + g_o, \alpha_{i,j-1}^D + g_e). \tag{9}$$

**The match marginal is the attention matrix.** By Theorem 3.2, the match-state marginal $P_{T,ij} = \partial V_T / \partial S_{ij} = \text{Pr}_{p_T}[(i, j) \in Y]$ is exactly the soft alignment matrix of Section 1: the posterior that residues $i$ and $j$ align. An encoder $f_\theta$ producing $S$ is trained by backpropagating through this matrix via the Hessian–vector product $\partial P_T / \partial S$ (Section 3.4).

**Learning the gaps.** The penalties $g_o, g_e$ are learned through the cross-Jacobian of Theorem 5.2. The sufficient statistic for $g_o$ is the number of gap-open events $N_{\text{open}}(Y)$, so

$$\frac{\partial P_{T,ij}}{\partial g_o} = \tfrac{1}{T} \text{Cov}_{p_T}\big(\mathbf{1}\{(i, j) \in Y\}, N_{\text{open}}(Y)\big),$$

the covariance between aligning $(i, j)$ and the number of gaps opened, with the analogous expression for $g_e$ (Table 2). Both are computed by the forward-$U$/backward-$W$ two-pass algorithm of Section 5.2 at the cost of one extra backward sweep. This is the gradient that learns $g_o, g_e$ in Section 7; freezing it collapses training (Table 4). Every algorithm in Table 3 instantiates these same four passes; the other eleven are derived in Sections A to C.

# 7. Experiments

We instantiate the framework on protein structure alignment (Section 7.1) and use it to test the central claim that learning the structure matters, not merely the representations that feed it: a structure-versus-capacity sweep (Section 7.2) and a frozen-versus-learned gap study (Section 7.3). A parsing transfer is in Section H and runtime benchmarks, including the $20{,}000\times$ kernel speedups, in Section I.12.

## 7.1. Protein Structure Alignment

We apply differentiable Smith–Waterman to learn protein structure similarity end-to-end (Figure 2): per-residue embeddings form a pairwise similarity matrix $S$, optimized against TM-align/US-align correspondences (Zhang & Skolnick, 2005; Zhang et al., 2022). From 15,176 SCOPe40 proteins (Chandonia et al., 2022) we form all pairs with

TM-score $> 0.6$ and split *proteins* 2:1 (cross-split pairs removed), yielding 297,601 training and 65,880 validation pairs (about 80/20 over retained pairs); pairs with TM-score $< 0.6$ act as an out-of-distribution test set. Amortization is favorable: all-vs-all TM-align took 3 days on 64 CPU cores, whereas our trained encoders ($O(n)$ embedding, $O(n^2)$ cheap pairwise alignment) score all pairs in $\sim$1 hour on one GPU; end-to-end training times are in Section I.6.

### 7.1.1. SUBSTITUTION MATRIX BASELINES

Learned classical substitution matrices (a full-rank Euclidean parameterization $S = EE^\top$ over the amino-acid and/or Foldseek 3Di (van Kempen et al., 2024) alphabets) form the zero-capacity point of the structure-versus-capacity sweep (Section 7.2). The best, a joint $400 \times 400$ (AA, 3Di) matrix, reaches $F_1 = 0.510$, and 3Di alone (0.468) far outperforms amino acids (0.232), echoing Foldseek; yet all trail the neural network encoders below by more than 0.2 $F_1$. Full per-parameterization metrics, learned gap regimes, and lDDT are in Section I.2.

### 7.1.2. NEURAL NETWORK ENCODERS

We then train three structure encoders, ProteinMPNN (Dauparas et al., 2022), GVP (Jing et al., 2021), and IPA (Jumper et al., 2021), at 64 and 128 dimensions, end-to-end through affine Smith–Waterman (Table 12, appendix). ProteinMPNN-128 is best ($F_1 = 0.748$), reaching 91% of the TM-align reference (lDDT 0.445 vs. 0.489, bounded by the below-40%-identity divergence of SCOPe40) and beating every substitution matrix (best 0.510). Every encoder converges to nearly the same penalties, $g_o \approx -3.2$ and $g_e \approx -0.11$, a $\sim$30:1 open-to-extend ratio matching the biological prior of rare but contiguous gaps; and they scale almost linearly with temperature over the equilibrated $T{=}5 \to 1$ training range ($g_o \approx \kappa_o T + b$, $\kappa_o \approx -2.8$, $R^2 \geq 0.999$), an encoder-agnostic Boltzmann-like constant (Section I.4). They also degrade gracefully with capacity ($128 \to 64$ drops only 0.007–0.009 $F_1$) and track TM-score far better than substitution matrices (Figure 12).

**Qualitative analysis.** Figure 3 shows the soft alignment marginals (structured attention): global algorithms (NW, DTW) concentrate mass on the diagonal, local ones (SW) are sparser (parsing marginals in Figure 4). The learned embeddings recover biochemical structure beyond BLOSUM62 (Henikoff & Henikoff, 1992) from alignment supervision alone (Section G.4).

## 7.2. Structure versus Capacity

We sweep encoder capacity from a fixed learned substitution matrix (no neural network encoder) up to a 128-dimensional ProteinMPNN encoder, crossed against three structural

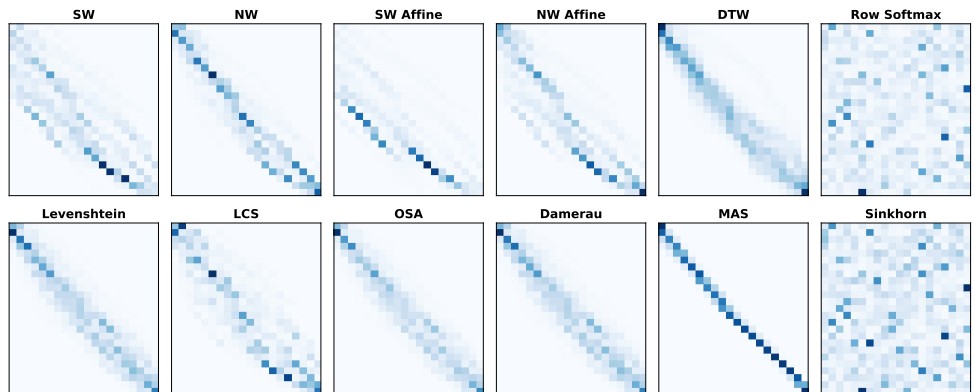

*Figure 3.* **Soft alignment marginals across ten DP algorithms and two attention baselines.** Each heatmap shows the posterior probability $P_{T,ij}$ that positions $i$ (vertical) and $j$ (horizontal) are aligned under the Gibbs distribution at $T = 1.0$ (dark = high). *Top row:* SW, NW, SW-Affine, NW-Affine, DTW, and row-wise softmax attention. *Bottom row:* Levenshtein, LCS, OSA, Damerau, MAS, and Sinkhorn attention. Local algorithms (SW) concentrate mass in high-similarity regions while global algorithms (NW, DTW) spread probability along the full diagonal.

*Table 4.* **Frozen versus learned gap penalties.** Validation $F_1$ with the same ProteinMPNN encoder when the Smith–Waterman gap is frozen at the bioinformatics default ($g = -10$) versus learned through the cross-Jacobian.

| Encoder | Variant | Learned $g$ | Frozen $g=-10$ |
|---|---|---|---|
| MPNN-128 | SW-linear | 0.743 | 0.321 |
| MPNN-64 | SW-linear | 0.736 | **0.128** |
| MPNN-128 | SW-affine | **0.748** | 0.705 |
| MPNN-64 | SW-affine | 0.739 | 0.692 |

choices: the correct DP for alignment (Smith–Waterman), no structural prior (row-softmax cross-attention), and a deliberately wrong prior (DTW, monotone warping without gaps), all trained as in Section 7.1. At full capacity the three reach validation $F_1 = 0.748$, $0.691$, and $0.046$; at zero capacity (the fixed matrix), $0.51$, $< 0.01$, and $0.055$. The reading is stark: the right recursion relation holds across the whole capacity range, the absence of any recursion relation depends entirely on capacity to compensate, and the wrong recursion relation cannot be rescued by capacity at all. The structure also absorbs capacity loss: halving the encoder ($128 \rightarrow 64$) costs the DP only $0.007$–$0.009$ $F_1$, against $0.016$ for softmax and $0.019$ for Sinkhorn.

### 7.3. Frozen versus Learned Gap Penalties

The cross-Jacobian $\partial P_T / \partial \boldsymbol{\eta}$ is what makes the gap penalties learnable; without it they are fixed by hand, at a steep cost. Freezing the single Smith–Waterman gap at the bioinformatics default ($g = -10$) collapses the linear model from $F_1 = 0.743$ to $0.321$ at full encoder capacity, and to $0.128$ at MPNN-64, because $\exp(-10/T)$ grows roughly $3000\times$ harsher as the temperature anneals from 5 to 1 and

the encoder can no longer compensate; learning $g$ through the cross-Jacobian recovers full performance (0.743). The affine variant, with a separate extend penalty to fall back on, pays a far smaller price ($0.748 \rightarrow 0.705$), diagnostic that the learned penalty matters most where the algorithm is least expressive (Table 4). The full regime ablation (Table 7, appendix) confirms learning the gaps is best across all algorithms, with learning temperature jointly indistinguishable and straight-through estimation about 5 percentage points lower in $F_1$.

## 8. Conclusion

Differentiable dynamic programming is structured attention: its temperature-softened marginals are exactly the attention weights, the soft analog of the classical traceback, collapsing onto the hard optimum as $T \rightarrow 0$. Because those marginals are already a first derivative of the log-partition, learning the program's own gap penalties, edit costs, and temperature is inherently a second-order problem. We derive the governing Hessian–vector products and cross-Jacobians in closed form as Gibbs covariances, compute them at the cost of the backward pass, and release them as fused CUDA kernels for twelve algorithms. These quantities are what make the structure trainable: freezing Smith–Waterman's gap penalty collapses $F_1$ from 0.74 to 0.32, while learning it recovers full performance and a biological gap regime, and transfers to constituency parsing, where a CKY CRF supervised by the gold tree's score alone matches dense per-span supervision within $0.003$ $F_1$. Natural next steps are learnable penalty functions $g(i, j) = f_\phi(h_i, h_j)$, multiple sequence alignment, and grammar induction. Where softmax attention replaced recurrence, structured soft attention replaces the hand-tuned discrete algorithms it generalizes.

## Impact Statement

We present d²p, achieving up to 20,000× speedups that make differentiable dynamic programming practical as Py-Torch operators. Following the Flash Attention design philosophy, we provide fused CUDA kernels with a simple API, lowering the barrier to structured deep learning. These tools may benefit scientific and engineering domains where structured constraints are known but labeled alignments or parses are costly. The main risks are overinterpreting learned algorithm parameters as scientific mechanisms without validation, and applying structured priors where their assumptions do not hold; we mitigate these by reporting parameter regimes, ablations, and failure cases (e.g., DTW on gapped alignment).

## Limitations

Several limitations bound the present work. *Numerics:* the fused kernels are stable in float32 only down to $T \approx 0.01$ for the long grid-alignment kernels; below that, log-space accumulation underflows and reduced precision (bf16/fp16) degrades earlier still (the shorter CKY parsing charts stay stable in log space slightly below this, e.g. $T \approx 0.008$ on Scheme). *Parallelism:* the recurrences expose $O(n+m)$ sequential wavefronts for grid algorithms and $O(n)$ span-length stages for chart algorithms, so very long inputs remain latency-bound despite the per-step parallelism. *Modeling:* we learn scalar, position-independent penalties $(g_o, g_e, \lambda, T)$; content- and position-dependent penalty functions $g(i,j) = f_\phi(h_i, h_j)$ are supported in principle by the same cross-Jacobians but unexplored here. *Scope:* our applications are demonstrations rather than tuned, state-of-the-art systems: single-run protein encoders at modest width, and a parsing study that trails scaled autoregressive language models in absolute $F_1$. Multiple sequence alignment, grammar induction, and coupling to larger pretrained encoders are left to future work.

## Acknowledgements

This work was supported by the Regeneron Prize for Creative Innovation.

## Author Contributions

CSM conceived and led the project, implemented the d²p library and all experiments, and wrote the manuscript. CSM and KL developed the mathematical derivations together, and KL verified the derivations and proofs and contributed to the writing.

## Conflicts of Interest

CSM is the founder and CEO of Orikata Bio PBC and holds shares in Orikata Bio PBC. The authors declare no other conflicts of interest.

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

## Overview of Appendix

In Section A, we provide complete derivations for all six alignment algorithms: linear-gap Smith–Waterman (A.1), affine-gap Smith–Waterman (A.2), linear-gap Needleman–Wunsch (A.3), affine-gap Needleman–Wunsch (A.4), Longest Common Subsequence (A.5), and Monotonic Alignment Search (A.6). Each derivation includes the DAG construction, soft recurrence equations, partition function, backward algorithm for computing marginals, parameter gradients (gap penalties), Hessian–vector products, and cross-Jacobians $\partial P_T/\partial g$, $\partial P_T/\partial g_o$, $\partial P_T/\partial g_e$, and $\partial P_T/\partial T$.

Section B covers the four edit distance algorithms using the softmin operator: Dynamic Time Warping (B.1) with step weight cross-Jacobians, Levenshtein distance (B.2) with insertion/deletion cost derivatives, Optimal String Alignment (B.3) with transposition cost derivatives, and Damerau–Levenshtein (B.4) for unrestricted transpositions.

Section C presents the two $O(n^3)$ parsing algorithms: CKY for constituency parsing (C.1) with span marginals $P_T(i,j)$ and split posteriors $P(k \mid i, j)$, and Eisner's algorithm for projective dependency parsing (C.2) with arc marginals $P_T(h \to d)$ and the four-chart decomposition ($I_R, I_L, C_R, C_L$).

Section D provides complete pseudocode for all twelve algorithms in a unified format, covering forward pass, backward pass (marginals), Hessian–vector product, and parameter cross-Jacobians. The pseudocode is organized by family: alignment algorithms (D.2), edit distance algorithms (D.3), and parsing algorithms (D.4).

Section E establishes regularity properties including Lipschitz continuity of the value function (Theorem E.1), Lipschitz continuity of the gradient (Theorem E.2), convexity of $V_T$ for maximization problems (Theorem E.5), and the approximation bound $|V_T(S) - V^*(S)| \le TL_{\max} \log D_{\max}$ (Theorem E.8).

Section F analyzes the zero-temperature limit (Theorem F.1), proving that soft marginals converge to hard alignments: $\lim_{T \to 0^+} P_{ij}^T = \mathbf{1}\{(i, j) \in Y^*\}$ where $Y^*$ is the optimal structure.

Section G contains supplemental figures: training dynamics showing loss, F1, and learned gap penalties over epochs (Figures 5 and 6); learned Euclidean substitution matrices for AA and 3Di alphabets (Figures 7 and 8); PCA comparison of BLOSUM62 vs. learned embeddings (Figure 9); correlation heatmaps across all models, clustered and unclustered (Figures 10 and 11); alignment length vs. TM-score scatter plots (Figure 12); and an example myoglobin–hemoglobin soft alignment (Figure 13).

Section I provides experimental details: SCOPe40 dataset construction and train/val splits; AdamW optimizer settings and temperature annealing schedule; ProteinMPNN, GVP, and IPA encoder architectures with layer counts and hidden dimensions; Euclidean parameterization $S = EE^\top$ for substitution matrices; CUDA kernel benchmark methodology; numerical stability techniques (log-space computation, numerically stable softmax); and anti-diagonal wavefront parallelization strategy for $O(nm/\min(n, m))$ parallel complexity.

Finally, Section J contains formal proofs of all theorems and lemmas stated in the main text, including Theorems 2.2, 3.1, 3.4, E.1 to E.8 and J.2.

## A. Alignment Algorithms

This appendix provides complete derivations for all alignment algorithms.

### A.1. Linear Smith–Waterman

Linear-gap Smith–Waterman is the simplest local alignment algorithm, using a single gap penalty $g < 0$.

**DAG Construction.** The node set is $V = \{(i, j) : 0 \le i \le n, 0 \le j \le m\} \cup \{s, t\}$. Edges:

- **Match (diagonal)**: $(i - 1, j - 1) \to (i, j)$ with weight $S_{ij}$
- **Gap in Y**: $(i - 1, j) \to (i, j)$ with weight $g$
- **Gap in X**: $(i, j - 1) \to (i, j)$ with weight $g$
- **Sky (restart)**: $s \to (i, j)$ with weight $0$
- **Terminal**: $(i, j) \to t$ with weight $0$

**Soft Recurrence.**

$$\alpha_{ij} = \text{LSE}_T\left(\alpha_{i-1,j-1} + S_{ij}, \alpha_{i-1,j} + g, \alpha_{i,j-1} + g, 0\right). \tag{10}$$

The score $S_{ij}$ applies only to the diagonal (match) transition. Initialize $\alpha_{0,0} = 0$, boundaries $\alpha_{i,0} = \alpha_{0,j} = -\infty$ for $i, j > 0$.

**Partition Function.**

$$V_T = \text{LSE}_T\left(\{\alpha_{ij}\}_{1 \le i \le n, 1 \le j \le m}\right). \tag{11}$$

**Backward Algorithm.** Define softmax weights at each $(i, j)$:

$$w^{\text{diag}}, w^{\text{up}}, w^{\text{left}}, w^{\text{sky}} = \text{softmax}\left(\frac{1}{T}\left(\alpha_{i-1,j-1} + S_{ij}, \alpha_{i-1,j} + g, \alpha_{i,j-1} + g, 0\right)\right). \tag{12}$$

Initialize $\beta_{ij} = \exp((\alpha_{ij} - V_T)/T)$ for all $(i, j)$. Propagate backward along anti-diagonals $k = i + j$ from $n + m$ to 2:

$$\beta_{i-1,j-1} \mathrel{+}= \beta_{ij} \cdot w_{ij}^{\text{diag}}, \tag{13}$$
$$\beta_{i-1,j} \mathrel{+}= \beta_{ij} \cdot w_{ij}^{\text{up}}, \tag{14}$$
$$\beta_{i,j-1} \mathrel{+}= \beta_{ij} \cdot w_{ij}^{\text{left}}. \tag{15}$$

The match marginal is $P_{T,ij} = \beta_{ij} \cdot w_{ij}^{\text{diag}}$, the probability that the alignment matches position $i$ with position $j$.

**Gap Gradient.**

$$\frac{\partial V_T}{\partial g} = \sum_{i,j} \beta_{ij} \cdot (w_{ij}^{\text{up}} + w_{ij}^{\text{left}}) = \mathbb{E}_{p_T}[N_{\text{gaps}}]. \tag{16}$$

**HVP.** Forward tangent: $\dot{\alpha}_{ij} = w^{\text{diag}}(\dot{\alpha}_{i-1,j-1} + V_{ij}) + w^{\text{up}}\dot{\alpha}_{i-1,j} + w^{\text{left}}\dot{\alpha}_{i,j-1}$. Note that $V_{ij}$ only appears in the diagonal term since $S_{ij}$ only applies to match transitions.

Backward tangent: Propagate $\dot{\beta}$ with weight tangents $\dot{w}_{ij} = w_{ij}(\dot{\alpha}_{\text{pred}} + V_{ij} - \dot{\alpha}_{ij})/T$.

**Gap Cross-Jacobian.** Forward $U$ pass: $U_{ij} = w^{\text{diag}} \cdot U_{i-1,j-1} + w^{\text{up}}(U_{i-1,j} + 1) + w^{\text{left}}(U_{i,j-1} + 1)$.

Backward $W$ pass: Initialize $W_{ij} = \beta_{ij}^{(0)}(U_{ij} - \partial V_T/\partial g)/T$, propagate with tangent weights.

Output: $[\partial P_T/\partial g]_{ij} = W_{ij}$.

**Temperature Cross-Jacobian.** Forward $U$ pass includes entropy term: $U_{ij} \mathrel{+}= (\alpha_{ij} - \bar{\alpha}_{ij})/T$ where $\bar{\alpha}_{ij} = \sum_k w_k \cdot v_k$ is the weighted average.

## A.2. Affine Smith–Waterman

Affine-gap Smith–Waterman uses separate gap-open ($g_o < 0$) and gap-extend ($g_e < 0$) penalties, requiring three DP states.

**Three-State DAG.** Node set: $V = \{(i, j, \sigma) : 0 \le i \le n, 0 \le j \le m, \sigma \in \{M, I, D\}\} \cup \{s, t\}$.

Edges:

- **Match**: $(\cdot, i-1, j-1, \sigma) \to (i, j, M)$ with weight $S_{ij}$ for $\sigma \in \{M, I, D\}$
- **Sky**: $s \to (i, j, M)$ with weight $S_{ij}$
- **Gap open (I)**: $(i-1, j, M) \to (i, j, I)$ with weight $g_o$
- **Gap extend (I)**: $(i-1, j, I) \to (i, j, I)$ with weight $g_e$
- **Gap open (D)**: $(i, j-1, M) \to (i, j, D)$ with weight $g_o$
- **Gap extend (D)**: $(i, j-1, D) \to (i, j, D)$ with weight $g_e$
- **Terminal**: all $(i, j, \sigma) \to t$ with weight 0

**Soft Recurrences.**

$$\alpha_{ij}^M = \text{LSE}_T(\alpha_{i-1,j-1}^M, \alpha_{i-1,j-1}^I, \alpha_{i-1,j-1}^D, 0) + S_{ij}, \tag{17}$$

$$\alpha_{ij}^I = \text{LSE}_T(\alpha_{i-1,j}^M + g_o, \alpha_{i-1,j}^I + g_e), \tag{18}$$

$$\alpha_{ij}^D = \text{LSE}_T(\alpha_{i,j-1}^M + g_o, \alpha_{i,j-1}^D + g_e). \tag{19}$$

Initialize $\alpha_{0,0}^M = 0$, all others $= -\infty$.

**Partition Function.**

$$V_T = \text{LSE}_T\left(\{\alpha_{ij}^\sigma\}_{i,j,\sigma}\right). \tag{20}$$

**Backward Algorithm.**  Softmax weights at each cell:

$$w^{M \to M}, w^{I \to M}, w^{D \to M}, w^{\text{sky}} = \text{softmax}\left((\alpha_{i-1,j-1}^M, \alpha_{i-1,j-1}^I, \alpha_{i-1,j-1}^D, 0)/T\right), \tag{21}$$

$$w^{M \to I}, w^{I \to I} = \text{softmax}\left((\alpha_{i-1,j}^M + g_o, \alpha_{i-1,j}^I + g_e)/T\right), \tag{22}$$

$$w^{M \to D}, w^{D \to D} = \text{softmax}\left((\alpha_{i,j-1}^M + g_o, \alpha_{i,j-1}^D + g_e)/T\right). \tag{23}$$

Initialize: $\beta_{ij}^\sigma = \exp((\alpha_{ij}^\sigma - V_T)/T)$.

Propagate backward:

$$\beta_{i-1,j-1}^M \mathrel{+}= \beta_{ij}^M \cdot w^{M \to M}, \quad \beta_{i-1,j-1}^I \mathrel{+}= \beta_{ij}^M \cdot w^{I \to M}, \quad \beta_{i-1,j-1}^D \mathrel{+}= \beta_{ij}^M \cdot w^{D \to M}, \tag{24}$$

$$\beta_{i-1,j}^M \mathrel{+}= \beta_{ij}^I \cdot w^{M \to I}, \quad \beta_{i-1,j}^I \mathrel{+}= \beta_{ij}^I \cdot w^{I \to I}, \tag{25}$$

$$\beta_{i,j-1}^M \mathrel{+}= \beta_{ij}^D \cdot w^{M \to D}, \quad \beta_{i,j-1}^D \mathrel{+}= \beta_{ij}^D \cdot w^{D \to D}. \tag{26}$$

Match marginal: $P_{T,ij} = \beta_{ij}^M$.

**Parameter Gradients.**

$$\frac{\partial V_T}{\partial g_o} = \sum_{ij}\left(\beta_{ij}^I w_{ij}^{M \to I} + \beta_{ij}^D w_{ij}^{M \to D}\right) = \mathbb{E}(N_{\text{opens}}), \tag{27}$$

$$\frac{\partial V_T}{\partial g_e} = \sum_{ij}\left(\beta_{ij}^I w_{ij}^{I \to I} + \beta_{ij}^D w_{ij}^{D \to D}\right) = \mathbb{E}(N_{\text{extends}}). \tag{28}$$

**HVP Derivation.**  Forward tangent for the three states with score perturbation $V$:

$$\dot\alpha_{ij}^M = w^{M \to M}(\dot\alpha_{i-1,j-1}^M + V_{ij}) + w^{I \to M}(\dot\alpha_{i-1,j-1}^I + V_{ij}) + w^{D \to M}(\dot\alpha_{i-1,j-1}^D + V_{ij}) + w^{\text{sky}}V_{ij}, \tag{29}$$

$$\dot\alpha_{ij}^I = w^{M \to I}\dot\alpha_{i-1,j}^M + w^{I \to I}\dot\alpha_{i-1,j}^I, \tag{30}$$

$$\dot\alpha_{ij}^D = w^{M \to D}\dot\alpha_{i,j-1}^M + w^{D \to D}\dot\alpha_{i,j-1}^D. \tag{31}$$

Aggregate: $\dot H_T = \sum_\sigma \beta_{ij}^{(0),\sigma}\dot\alpha_{ij}^\sigma$.

Backward tangent: Initialize $\dot\beta_{ij}^\sigma = \beta_{ij}^{(0),\sigma}(\dot\alpha_{ij}^\sigma - \dot H_T)/T$. Weight tangents: $\dot w_{ij}^k = w_{ij}^k(\dot v^k - \dot\alpha_{ij}^\sigma)/T$. Propagate through all state transitions.

Output: $\text{HVP}_{ij} = \dot\beta_{ij}^M$.

**Cross-Jacobians.**  For $S \in \{g_o, g_e\}$:

Forward $U$ pass:

$$U_{ij}^M = \sum_\sigma w_{ij}^{\sigma \to M} \cdot U_{i-1,j-1}^\sigma, \tag{32}$$

$$U_{ij}^I = w^{M \to I}(U_{i-1,j}^M + \mathbf{1}[S = g_o]) + w^{I \to I}(U_{i-1,j}^I + \mathbf{1}[S = g_e]), \tag{33}$$

$$U_{ij}^D = w^{M \to D}(U_{i,j-1}^M + \mathbf{1}[S = g_o]) + w^{D \to D}(U_{i,j-1}^D + \mathbf{1}[S = g_e]). \tag{34}$$

Backward $W$ pass: Initialize $W_{ij}^\sigma = \beta_{ij}^{(0),\sigma}(U_{ij}^\sigma - \partial V_T/\partial S)/T$, propagate.

Output: $[\partial P_T/\partial S]_{ij} = W_{ij}^M$.

**Temperature Cross-Jacobian.** Forward $U^T$ pass includes entropy terms for each state:

$$U_{ij}^{T,M} = \sum_\sigma w^{\sigma\to M} U_{i-1,j-1}^{T,\sigma} + (\alpha_{ij}^M - S_{ij} - \bar\alpha_{ij}^M)/T, \tag{35}$$

$$U_{ij}^{T,I} = w^{M\to I} U_{i-1,j}^{T,M} + w^{I\to I} U_{i-1,j}^{T,I} + (\alpha_{ij}^I - \bar\alpha_{ij}^I)/T, \tag{36}$$

$$U_{ij}^{T,D} = w^{M\to D} U_{i,j-1}^{T,M} + w^{D\to D} U_{i,j-1}^{T,D} + (\alpha_{ij}^D - \bar\alpha_{ij}^D)/T, \tag{37}$$

where $\bar\alpha_{ij}^\sigma$ is the weighted average of predecessor values for state $\sigma$.

Backward $W^T$ pass with entropy-weighted tangents.

Output: $[\partial P_T/\partial T]_{ij} = W_{ij}^{T,M}$.

**Covariance Interpretation.**

$$\frac{\partial P_{T,ij}}{\partial T} = -\frac{1}{T^2}\mathrm{Cov}_{p_T}\big(\mathbf{1}\{(i,j) \in \text{alignment}\}, \text{score(alignment)}\big). \tag{38}$$

## A.3. Linear Needleman–Wunsch

Linear-gap Needleman–Wunsch computes global alignment using a single gap penalty $g < 0$. Unlike Smith–Waterman, alignment must span both complete sequences.

**DAG Construction.** The node set is $V = \{(i,j) : 0 \le i \le n, 0 \le j \le m\}$ with source $(0,0)$ and sink $(n,m)$. Edges:

- **Match (diagonal)**: $(i-1,j-1) \to (i,j)$ with weight $S_{ij}$
- **Gap in Y**: $(i-1,j) \to (i,j)$ with weight $g$
- **Gap in X**: $(i,j-1) \to (i,j)$ with weight $g$

No sky edges (no local restart allowed).

**Soft Recurrence.**

$$\alpha_{ij} = \mathrm{LSE}_T(\alpha_{i-1,j-1} + S_{ij}, \alpha_{i-1,j} + g, \alpha_{i,j-1} + g). \tag{39}$$

The score $S_{ij}$ applies only to the diagonal (match) transition. Boundaries: $\alpha_{0,0} = 0$, $\alpha_{i,0} = i \cdot g$ for $i > 0$, $\alpha_{0,j} = j \cdot g$ for $j > 0$.

**Partition Function.**

$$V_T = \alpha_{n,m}. \tag{40}$$

This is the log-partition for global alignments only.

**Backward Algorithm.** Define softmax weights at each $(i,j)$:

$$w^{\mathrm{diag}}, w^{\mathrm{up}}, w^{\mathrm{left}} = \mathrm{softmax}\left(\frac{1}{T}(\alpha_{i-1,j-1} + S_{ij}, \alpha_{i-1,j} + g, \alpha_{i,j-1} + g)\right). \tag{41}$$

Initialize $\beta_{n,m} = 1$, all others $= 0$. Propagate backward along anti-diagonals $k = i + j$ from $n + m$ to 2:

$$\beta_{i-1,j-1} \mathrel{+}= \beta_{ij} \cdot w_{ij}^{\mathrm{diag}}, \tag{42}$$

$$\beta_{i-1,j} \mathrel{+}= \beta_{ij} \cdot w_{ij}^{\mathrm{up}}, \tag{43}$$

$$\beta_{i,j-1} \mathrel{+}= \beta_{ij} \cdot w_{ij}^{\mathrm{left}}. \tag{44}$$

The match marginal is $P_{T,ij} = \beta_{ij} \cdot w_{ij}^{\mathrm{diag}}$, the probability that the alignment matches position $i$ with position $j$.

**Gap Gradient.**

$$\frac{\partial V_T}{\partial g} = \sum_{i,j} \beta_{ij} \cdot (w_{ij}^{\mathrm{up}} + w_{ij}^{\mathrm{left}}) + \sum_{i=1}^{n} \beta_{i,0} + \sum_{j=1}^{m} \beta_{0,j} = \mathbb{E}_{P_T}[N_{\mathrm{gaps}}]. \tag{45}$$

The boundary terms account for gaps along the edges.

**HVP Derivation.**  Forward tangent with perturbation $V$:

$$\dot{\alpha}_{ij} = w^{\mathrm{diag}}(\dot{\alpha}_{i-1,j-1} + V_{ij}) + w^{\mathrm{up}}\dot{\alpha}_{i-1,j} + w^{\mathrm{left}}\dot{\alpha}_{i,j-1}. \tag{46}$$

The perturbation $V$ of the match scores enters only on the diagonal (match) branch; the up/left branches are gaps and do not use $S$. The boundary cells are pure gaps, so $\dot{\alpha}_{i,0} = \dot{\alpha}_{0,j} = 0$.

Aggregate: $\dot{H}_T = \dot{\alpha}_{n,m}$.

Backward tangent: Initialize $\dot{\beta}_{n,m} = 0$, others $= 0$. Weight tangents: $\dot{w}_{ij}^k = w_{ij}^k(\dot{v}^k + V_{ij} - \dot{\alpha}_{ij})/T$. Propagate:

$$\dot{\beta}_{\mathrm{pred}} \mathrel{+}= \dot{\beta}_{ij}w^k + \beta_{ij}\dot{w}^k. \tag{47}$$

Output: $\mathrm{HVP}_{ij} = \dot{\beta}_{ij}$.

**Gap Cross-Jacobian.**  Forward $U$ pass:

$$U_{ij} = w^{\mathrm{diag}} \cdot U_{i-1,j-1} + w^{\mathrm{up}}(U_{i-1,j} + 1) + w^{\mathrm{left}}(U_{i,j-1} + 1). \tag{48}$$

Boundaries: $U_{i,0} = i$, $U_{0,j} = j$.

Aggregate: $\partial V_T/\partial g = U_{n,m}$.

Backward $W$ pass: Initialize $W_{n,m} = 0$, compute weight tangents from $U$.

$$\dot{w}_{ij}^k = w_{ij}^k(U_{\mathrm{pred}} + \mathbf{1}[k \neq \mathrm{diag}] - U_{ij})/T. \tag{49}$$

Propagate: $W_{\mathrm{pred}} \mathrel{+}= W_{ij}w^k + \beta_{ij}\dot{w}^k$.

Output: $[\partial P_T/\partial g]_{ij} = W_{ij}$.

**Temperature Cross-Jacobian.**  Forward $U^T$ pass includes entropy term:

$$U_{ij}^T = \sum_k w^k(U_{\mathrm{pred}}^T) + (\alpha_{ij} - \bar{\alpha}_{ij})/T, \tag{50}$$

where $\bar{\alpha}_{ij} = \sum_k w^k(\alpha_{\mathrm{pred}} + \mathbf{1}[k \neq \mathrm{diag}] \cdot g)$ is the weighted average of predecessors.

Backward $W^T$ pass follows the same structure with entropy-weighted tangents.

## A.4. Affine Needleman–Wunsch

Affine-gap Needleman–Wunsch uses separate gap-open ($g_o < 0$) and gap-extend ($g_e < 0$) penalties for global alignment with three DP states.

**Three-State DAG.**  Node set: $V = \{(i,j,\sigma) : 0 \leq i \leq n, 0 \leq j \leq m, \sigma \in \{M,I,D\}\}$. States: $M$ = match/mismatch, $I$ = insertion (gap in X), $D$ = deletion (gap in Y).

Edges:

- **Match**: $(\cdot, i-1, j-1, \sigma) \to (i,j,M)$ with weight $S_{ij}$ for $\sigma \in \{M,I,D\}$
- **Gap open (I)**: $(i-1,j,M) \to (i,j,I)$ with weight $g_o$
- **Gap extend (I)**: $(i-1,j,I) \to (i,j,I)$ with weight $g_e$
- **Gap open (D)**: $(i,j-1,M) \to (i,j,D)$ with weight $g_o$
- **Gap extend (D)**: $(i,j-1,D) \to (i,j,D)$ with weight $g_e$

**Soft Recurrences.**

$$\alpha_{ij}^M = \text{LSE}_T(\alpha_{i-1,j-1}^M, \alpha_{i-1,j-1}^I, \alpha_{i-1,j-1}^D) + S_{ij}, \tag{51}$$

$$\alpha_{ij}^I = \text{LSE}_T(\alpha_{i-1,j}^M + g_o, \alpha_{i-1,j}^I + g_e), \tag{52}$$

$$\alpha_{ij}^D = \text{LSE}_T(\alpha_{i,j-1}^M + g_o, \alpha_{i,j-1}^D + g_e). \tag{53}$$

Boundaries:

$$\alpha_{0,0}^M = 0, \quad \alpha_{0,0}^I = \alpha_{0,0}^D = -\infty, \tag{54}$$

$$\alpha_{i,0}^I = g_o + (i-1)g_e, \quad \alpha_{i,0}^M = \alpha_{i,0}^D = -\infty \text{ for } i > 0, \tag{55}$$

$$\alpha_{0,j}^D = g_o + (j-1)g_e, \quad \alpha_{0,j}^M = \alpha_{0,j}^I = -\infty \text{ for } j > 0. \tag{56}$$

**Partition Function.**

$$V_T = \text{LSE}_T(\alpha_{n,m}^M, \alpha_{n,m}^I, \alpha_{n,m}^D). \tag{57}$$

**Backward Algorithm.** Softmax weights at each cell:

$$w^{M\to M}, w^{I\to M}, w^{D\to M} = \text{softmax}\left((\alpha_{i-1,j-1}^M, \alpha_{i-1,j-1}^I, \alpha_{i-1,j-1}^D)/T\right), \tag{58}$$

$$w^{M\to I}, w^{I\to I} = \text{softmax}\left((\alpha_{i-1,j}^M + g_o, \alpha_{i-1,j}^I + g_e)/T\right), \tag{59}$$

$$w^{M\to D}, w^{D\to D} = \text{softmax}\left((\alpha_{i,j-1}^M + g_o, \alpha_{i,j-1}^D + g_e)/T\right). \tag{60}$$

Terminal weights:

$$w_t^M, w_t^I, w_t^D = \text{softmax}\left((\alpha_{n,m}^M, \alpha_{n,m}^I, \alpha_{n,m}^D)/T\right). \tag{61}$$

Initialize: $\beta_{n,m}^M = w_t^M, \beta_{n,m}^I = w_t^I, \beta_{n,m}^D = w_t^D$, all others $= 0$.

Propagate backward:

$$\beta_{i-1,j-1}^M \mathrel{+}= \beta_{ij}^M \cdot w^{M\to M}, \tag{62}$$

$$\beta_{i-1,j-1}^I \mathrel{+}= \beta_{ij}^M \cdot w^{I\to M}, \tag{63}$$

$$\beta_{i-1,j-1}^D \mathrel{+}= \beta_{ij}^M \cdot w^{D\to M}, \tag{64}$$

$$\beta_{i-1,j}^M \mathrel{+}= \beta_{ij}^I \cdot w^{M\to I}, \tag{65}$$

$$\beta_{i-1,j}^I \mathrel{+}= \beta_{ij}^I \cdot w^{I\to I}, \tag{66}$$

$$\beta_{i,j-1}^M \mathrel{+}= \beta_{ij}^D \cdot w^{M\to D}, \tag{67}$$

$$\beta_{i,j-1}^D \mathrel{+}= \beta_{ij}^D \cdot w^{D\to D}. \tag{68}$$

Match marginal: $P_{T,ij} = \beta_{ij}^M$.

**Parameter Gradients.**

$$\frac{\partial V_T}{\partial g_o} = \sum_{ij} \left(\beta_{ij}^I w_{ij}^{M\to I} + \beta_{ij}^D w_{ij}^{M\to D}\right) = \mathbb{E}(N_{\text{opens}}), \tag{69}$$

$$\frac{\partial V_T}{\partial g_e} = \sum_{ij} \left(\beta_{ij}^I w_{ij}^{I\to I} + \beta_{ij}^D w_{ij}^{D\to D}\right) + (\text{boundary terms}) = \mathbb{E}(N_{\text{extends}}). \tag{70}$$

**HVP Derivation.** Forward tangent for the three states:

$$\dot{\alpha}_{ij}^M = w^{M\to M}(\dot{\alpha}_{i-1,j-1}^M + V_{ij}) + w^{I\to M}(\dot{\alpha}_{i-1,j-1}^I + V_{ij}) + w^{D\to M}(\dot{\alpha}_{i-1,j-1}^D + V_{ij}), \tag{71}$$

$$\dot{\alpha}_{ij}^I = w^{M\to I}\dot{\alpha}_{i-1,j}^M + w^{I\to I}\dot{\alpha}_{i-1,j}^I, \tag{72}$$

$$\dot{\alpha}_{ij}^D = w^{M\to D}\dot{\alpha}_{i,j-1}^M + w^{D\to D}\dot{\alpha}_{i,j-1}^D. \tag{73}$$

Aggregate: $\dot{H}_T = w_t^M \dot{\alpha}_{n,m}^M + w_t^I \dot{\alpha}_{n,m}^I + w_t^D \dot{\alpha}_{n,m}^D$.

Backward tangent: Initialize with terminal tangents, propagate through all states with weight tangents.

**Gap-Open Cross-Jacobian.** Forward $U^{g_o}$ pass:

$$U_{ij}^{g_o,M} = \sum_\sigma w_{ij}^{\sigma \to M} \cdot U_{i-1,j-1}^{g_o,\sigma}, \tag{74}$$

$$U_{ij}^{g_o,I} = w^{M \to I}(U_{i-1,j}^{g_o,M} + 1) + w^{I \to I}U_{i-1,j}^{g_o,I}, \tag{75}$$

$$U_{ij}^{g_o,D} = w^{M \to D}(U_{i,j-1}^{g_o,M} + 1) + w^{D \to D}U_{i,j-1}^{g_o,D}. \tag{76}$$

Aggregate: $\partial V_T / \partial g_o = \sum_\sigma w_t^\sigma U_{n,m}^{g_o,\sigma}$.

Backward $W^{g_o}$ pass: Initialize, propagate with weight tangents.

Output: $[\partial P_T / \partial g_o]_{ij} = W_{ij}^{g_o,M}$.

**Gap-Extend Cross-Jacobian.** Forward $U^{g_e}$ pass:

$$U_{ij}^{g_e,M} = \sum_\sigma w_{ij}^{\sigma \to M} \cdot U_{i-1,j-1}^{g_e,\sigma}, \tag{77}$$

$$U_{ij}^{g_e,I} = w^{M \to I}U_{i-1,j}^{g_e,M} + w^{I \to I}(U_{i-1,j}^{g_e,I} + 1), \tag{78}$$

$$U_{ij}^{g_e,D} = w^{M \to D}U_{i,j-1}^{g_e,M} + w^{D \to D}(U_{i,j-1}^{g_e,D} + 1). \tag{79}$$

Boundary contributions: $U_{i,0}^{g_e,I} = i, U_{0,j}^{g_e,D} = j$.

Output: $[\partial P_T / \partial g_e]_{ij} = W_{ij}^{g_e,M}$.

**Temperature Cross-Jacobian.** Forward $U^T$ pass includes entropy terms for each state:

$$U_{ij}^{T,M} = \sum_\sigma w^{\sigma \to M}U_{i-1,j-1}^{T,\sigma} + (\alpha_{ij}^M - S_{ij} - \bar{\alpha}_{ij}^M)/T, \tag{80}$$

$$U_{ij}^{T,I} = w^{M \to I}U_{i-1,j}^{T,M} + w^{I \to I}U_{i-1,j}^{T,I} + (\alpha_{ij}^I - \bar{\alpha}_{ij}^I)/T, \tag{81}$$

$$U_{ij}^{T,D} = w^{M \to D}U_{i,j-1}^{T,M} + w^{D \to D}U_{i,j-1}^{T,D} + (\alpha_{ij}^D - \bar{\alpha}_{ij}^D)/T, \tag{82}$$

where $\bar{\alpha}_{ij}^\sigma$ is the weighted average of predecessor values for state $\sigma$.

Backward $W^T$ pass with entropy-weighted tangents.

Output: $[\partial P_T / \partial T]_{ij} = W_{ij}^{T,M}$.

### A.5. Longest Common Subsequence (LCS)

LCS finds the longest subsequence common to both sequences. It is a special case of alignment with zero gap penalties and binary match scores.

**DAG Construction.** Node set: $V = \{(i,j) : 0 \le i \le n, 0 \le j \le m\}$. Edges:

- **Match**: $(i-1, j-1) \to (i,j)$ with weight $S_{ij}$ where $S_{ij} = r$ if $x_i = y_j$, else 0
- **Skip X**: $(i-1, j) \to (i,j)$ with weight 0
- **Skip Y**: $(i, j-1) \to (i,j)$ with weight 0

**Soft Recurrence.**

$$\alpha_{ij} = \text{LSE}_T(\alpha_{i-1,j-1} + S_{ij}, \alpha_{i-1,j}, \alpha_{i,j-1}). \tag{83}$$

Boundaries: $\alpha_{i,0} = \alpha_{0,j} = 0$ for all $i, j$.

**Partition Function.**

$$V_T = \alpha_{n,m}. \tag{84}$$

**Backward Algorithm.** Softmax weights:

$$w^{\text{match}}, w^{\text{skipX}}, w^{\text{skipY}} = \text{softmax}\left(\frac{1}{T}\left(\alpha_{i-1,j-1} + S_{ij}, \alpha_{i-1,j}, \alpha_{i,j-1}\right)\right). \tag{85}$$

Initialize $\beta_{n,m} = 1$, all others $= 0$. Propagate backward:

$$\beta_{i-1,j-1} \mathrel{+}= \beta_{ij} \cdot w_{ij}^{\text{match}}, \tag{86}$$
$$\beta_{i-1,j} \mathrel{+}= \beta_{ij} \cdot w_{ij}^{\text{skipX}}, \tag{87}$$
$$\beta_{i,j-1} \mathrel{+}= \beta_{ij} \cdot w_{ij}^{\text{skipY}}. \tag{88}$$

Match marginal: $P_{T,ij} = \beta_{ij} \cdot w_{ij}^{\text{match}}$ (probability that position $(i, j)$ is in the LCS).

**Score Gradient.** Since match scores are the only parameters:

$$\frac{\partial V_T}{\partial S_{ij}} = \beta_{ij} \cdot w_{ij}^{\text{match}} = P_{T,ij}. \tag{89}$$

**HVP Derivation.** Forward tangent:

$$\dot{\alpha}_{ij} = w^{\text{match}}(\dot{\alpha}_{i-1,j-1} + V_{ij}) + w^{\text{skipX}}\dot{\alpha}_{i-1,j} + w^{\text{skipY}}\dot{\alpha}_{i,j-1}. \tag{90}$$

Backward tangent with weight tangents $\dot{w} = w(\dot{v} - \dot{\alpha}_{ij})/T$.

**Temperature Cross-Jacobian.** Forward $U^T$ pass:

$$U_{ij}^T = w^{\text{match}}U_{i-1,j-1}^T + w^{\text{skipX}}U_{i-1,j}^T + w^{\text{skipY}}U_{i,j-1}^T + (\alpha_{ij} - \bar{\alpha}_{ij})/T, \tag{91}$$

where $\bar{\alpha}_{ij} = w^{\text{match}}(\alpha_{i-1,j-1} + S_{ij}) + w^{\text{skipX}}\alpha_{i-1,j} + w^{\text{skipY}}\alpha_{i,j-1}$.

Backward $W^T$ pass with entropy-weighted tangents.

Output: $[\partial P_T/\partial T]_{ij} = W_{ij}^T$, representing how each match marginal responds to temperature changes.

### A.6. Monotonic Alignment Search (MAS)

MAS constrains alignments to be monotonic (no crossing paths), used in speech synthesis (Kim et al., 2020). It computes soft alignments between encoder and decoder sequences.

**DAG Construction.** Node set: $V = \{(t,s) : 0 \leq t \leq T_{\max}, 0 \leq s \leq S_{\max}\}$. Here $t$ indexes the decoder (output) sequence and $s$ indexes the encoder (source) sequence.

Edges (monotonic constraint):

- **Stay**: $(t-1,s) \rightarrow (t,s)$ with weight $S_{t,s}$ (same source position)

- **Advance**: $(t-1,s-1) \rightarrow (t,s)$ with weight $S_{t,s}$ (move to next source)

This ensures each decoder position maps to exactly one source position, and source positions are visited in order.

**Soft Recurrence.**

$$\alpha_{t,s} = S_{t,s} + \text{LSE}_T(\alpha_{t-1,s}, \alpha_{t-1,s-1}). \tag{92}$$

Boundaries: $\alpha_{0,0} = S_{0,0}$, $\alpha_{0,s} = -\infty$ for $s > 0$, $\alpha_{t,0} = \alpha_{t-1,0} + S_{t,0}$ for $t > 0$.

**Partition Function.**

$$V_T = \alpha_{T_{\max}, S_{\max}}. \tag{93}$$

**Backward Algorithm.** Softmax weights:

$$w^{\text{stay}}, w^{\text{advance}} = \text{softmax}\left(\frac{1}{T}\big(\alpha_{t-1,s}, \alpha_{t-1,s-1}\big)\right). \tag{94}$$

Initialize $\beta_{T_{\max}, S_{\max}} = 1$, all others $= 0$. Propagate backward:

$$\beta_{t-1,s} \mathrel{+}= \beta_{t,s} \cdot w^{\text{stay}}_{t,s}, \tag{95}$$

$$\beta_{t-1,s-1} \mathrel{+}= \beta_{t,s} \cdot w^{\text{advance}}_{t,s}. \tag{96}$$

Alignment marginal: $P_{t,s} = \beta_{t,s}$ (probability that decoder position $t$ aligns to source position $s$).

**Score Gradient.**

$$\frac{\partial V_T}{\partial S_{t,s}} = \beta_{t,s} = P_{t,s}. \tag{97}$$

**HVP Derivation.** Forward tangent. Since $S_{t,s}$ is added once, outside the LSE, $V_{t,s}$ appears once:

$$\dot{\alpha}_{t,s} = V_{t,s} + w^{\text{stay}}\dot{\alpha}_{t-1,s} + w^{\text{advance}}\dot{\alpha}_{t-1,s-1}. \tag{98}$$

Backward tangent propagates with weight tangents.

**Temperature Cross-Jacobian.** Forward $U^T$ pass:

$$U^T_{t,s} = w^{\text{stay}}U^T_{t-1,s} + w^{\text{advance}}U^T_{t-1,s-1} + (\alpha_{t,s} - S_{t,s} - \bar{\alpha}_{t,s})/T, \tag{99}$$

where $\bar{\alpha}_{t,s} = w^{\text{stay}}\alpha_{t-1,s} + w^{\text{advance}}\alpha_{t-1,s-1}$.

The entropy term $(\alpha_{t,s} - S_{t,s} - \bar{\alpha}_{t,s})/T$ captures uncertainty at each position.

Backward $W^T$ pass with entropy-weighted tangents.

Output: $[\partial P_T/\partial T]_{t,s} = W^T_{t,s}$.

**Covariance Interpretation.**

$$\frac{\partial P_{t,s}}{\partial T} = -\frac{1}{T^2}\text{Cov}_{p_T}\big(\mathbf{1}\{(t,s) \in \text{path}\}, \text{score}(\text{path})\big). \tag{100}$$

Positions where the marginal covaries negatively with total score increase when temperature increases (more probability mass shifts to suboptimal alignments).

## B. Edit Distance Algorithms

Edit distance algorithms *minimize* cost using softmin.

### B.1. Dynamic Time Warping (DTW)

DTW aligns time series by warping the time axis. We extend the standard formulation with learnable step weights $(\omega_d, \omega_h, \omega_v)$ for diagonal, horizontal, and vertical moves.

**Soft Recurrence (Minimization).** Given cost matrix $C \in \mathbb{R}^{n \times m}$ and step weights $(\omega_d, \omega_h, \omega_v)$:

$$\alpha_{ij} = C_{ij} + \text{smin}_T(\alpha_{i-1,j-1} + \omega_d, \alpha_{i-1,j} + \omega_v, \alpha_{i,j-1} + \omega_h). \tag{101}$$

Boundaries: $\alpha_{0,0} = C_{0,0}$ (or 0), $\alpha_{i,0} = \alpha_{i-1,0} + C_{i,0} + \omega_v$, $\alpha_{0,j} = \alpha_{0,j-1} + C_{0,j} + \omega_h$.

**Softmin Weights.**

$$w^{\text{diag}}, w^{\text{up}}, w^{\text{left}} = \text{softmin}\left((\alpha_{i-1,j-1} + \omega_d, \alpha_{i-1,j} + \omega_v, \alpha_{i,j-1} + \omega_h)/T\right). \tag{102}$$

**Backward Algorithm.** Initialize $\beta_{n,m} = 1$. Propagate:

$$\beta_{i-1,j-1} \mathrel{+}= \beta_{ij} w^{\text{diag}}, \quad \beta_{i-1,j} \mathrel{+}= \beta_{ij} w^{\text{up}}, \quad \beta_{i,j-1} \mathrel{+}= \beta_{ij} w^{\text{left}}. \tag{103}$$

Warping posteriors: $P_{T,ij} = \beta_{ij}$.

**Temperature Gradient.**

$$\frac{\partial V_T}{\partial T} = \frac{V_T - \mathbb{E}(\text{cost})}{T} = -H(q_T), \tag{104}$$

where $V_T$ is the soft distance and $H(q_T)$ is the entropy of the path distribution.

**HVP Derivation.** For minimization problems, the Hessian is *negative* semidefinite. The HVP computes $-(1/T)\text{Cov}_{q_T}(\mathbf{1}_a, \mathbf{1}_b) \cdot V$.

Forward tangent with cost perturbation $V$. Because the cell cost $C_{ij}$ is added once, outside the softmin, $V_{ij}$ appears once (not inside each branch):

$$\dot{\alpha}_{ij} = V_{ij} + w^{\text{diag}}\dot{\alpha}_{i-1,j-1} + w^{\text{up}}\dot{\alpha}_{i-1,j} + w^{\text{left}}\dot{\alpha}_{i,j-1}. \tag{105}$$

Aggregate: $\dot{D}_T = \dot{\alpha}_{n,m}$.

Backward tangent: Initialize $\dot{\beta}_{n,m} = 0$. Weight tangents for softmin:

$$\dot{w}_{ij}^k = -w_{ij}^k(\dot{\alpha}_{\text{pred}} - \dot{\alpha}_{ij})/T. \tag{106}$$

Note the negative sign compared to softmax (from the $-1/T$ factor in softmin Hessian).

Propagate:

$$\dot{\beta}_{\text{pred}} \mathrel{+}= \dot{\beta}_{ij} w^k - \beta_{ij} \cdot w^k(\dot{\alpha}_{\text{pred}} - \dot{\alpha}_{ij})/T. \tag{107}$$

Output: $\text{HVP}_{ij} = \dot{\beta}_{ij}$ (the tangent of the cell marginal; the minimization sign is carried by the softmin weight tangents, not an external negation).

**Temperature Cross-Jacobian.** Forward $U^T$ pass:

$$U_{ij}^T = w^{\text{diag}}U_{i-1,j-1}^T + w^{\text{up}}U_{i-1,j}^T + w^{\text{left}}U_{i,j-1}^T - (\alpha_{ij} - C_{ij} - \bar{\alpha}_{ij})/T, \tag{108}$$

where $\bar{\alpha}_{ij} = w^{\text{diag}}\alpha_{i-1,j-1} + w^{\text{up}}\alpha_{i-1,j} + w^{\text{left}}\alpha_{i,j-1}$.

The term $-(\alpha_{ij} - C_{ij} - \bar{\alpha}_{ij})/T$ is the local entropy contribution (negative for softmin).

Backward $W^T$ pass: Initialize $W_{n,m}^T = 0$, propagate with weight tangents.

**Covariance Interpretation.**

$$\frac{\partial P_{T,ij}}{\partial T} = \frac{1}{T^2}\text{Cov}_{q_T}\left(\mathbf{1}\{(i,j) \in \text{path}\}, \text{cost}(\text{path})\right). \tag{109}$$

Note the positive sign (opposite to maximization): positions that covary positively with cost have marginals that *increase* when temperature increases (suboptimal paths become more likely).

**Step Weight Cross-Jacobians.** For learnable step weights $(\omega_d, \omega_h, \omega_v)$, we compute $\partial P_T / \partial \omega_d$, $\partial P_T / \partial \omega_h$, $\partial P_T / \partial \omega_v$.

Forward $U^{\omega_d}$ pass (diagonal step weight):

$$U_{ij}^{\omega_d} = w^{\text{diag}}(U_{i-1,j-1}^{\omega_d} + 1) + w^{\text{up}}U_{i-1,j}^{\omega_d} + w^{\text{left}}U_{i,j-1}^{\omega_d}. \tag{110}$$

The $+1$ term appears only in the diagonal contribution since $\omega_d$ only affects diagonal transitions.

Similarly for horizontal and vertical:

$$U_{ij}^{\omega_h} = w^{\text{diag}}U_{i-1,j-1}^{\omega_h} + w^{\text{up}}U_{i-1,j}^{\omega_h} + w^{\text{left}}(U_{i,j-1}^{\omega_h} + 1), \tag{111}$$

$$U_{ij}^{\omega_v} = w^{\text{diag}}U_{i-1,j-1}^{\omega_v} + w^{\text{up}}(U_{i-1,j}^{\omega_v} + 1) + w^{\text{left}}U_{i,j-1}^{\omega_v}. \tag{112}$$

Aggregate: $\partial V_T / \partial \omega_d = U_{n,m}^{\omega_d} = \mathbb{E}_{q_T}[N_{\text{diag}}]$ (expected number of diagonal steps).

Backward $W$ pass and covariance interpretation follow the standard pattern:

$$\frac{\partial P_{T,ij}}{\partial \omega_d} = -\frac{1}{T}\text{Cov}_{q_T}\big(\mathbf{1}\{(i,j) \in \text{path}\}, N_{\text{diag}}\big). \tag{113}$$

## B.2. Levenshtein Distance

Levenshtein computes the minimum number of edits (insert, delete, substitute) to transform one string into another.

**DAG Construction.** Node set: $V = \{(i,j) : 0 \le i \le n, 0 \le j \le m\}$. Edges:

- **Substitution**: $(i-1, j-1) \to (i,j)$ with cost $C_{ij}$ ($= 0$ if $x_i = y_j$, else substitution cost)
- **Deletion**: $(i-1, j) \to (i,j)$ with cost $c_d$
- **Insertion**: $(i, j-1) \to (i,j)$ with cost $c_i$

**Soft Recurrence.** With insertion cost $c_i$, deletion cost $c_d$, and substitution costs $C_{ij}$:

$$\alpha_{ij} = \text{smin}_T(\alpha_{i-1,j-1} + C_{ij}, \alpha_{i-1,j} + c_d, \alpha_{i,j-1} + c_i). \tag{114}$$

Boundaries: $\alpha_{0,0} = 0$, $\alpha_{i,0} = i \cdot c_d$, $\alpha_{0,j} = j \cdot c_i$.

**Partition Function.**

$$V_T = \alpha_{n,m}. \tag{115}$$

**Softmin Weights.**

$$w^{\text{sub}}, w^{\text{del}}, w^{\text{ins}} = \text{softmin}\big((\alpha_{i-1,j-1} + C_{ij}, \alpha_{i-1,j} + c_d, \alpha_{i,j-1} + c_i)/T\big). \tag{116}$$

**Backward Algorithm.** Initialize $\beta_{n,m} = 1$, all others $= 0$. Propagate backward along anti-diagonals:

$$\beta_{i-1,j-1} \mathrel{+}= \beta_{ij} \cdot w_{ij}^{\text{sub}}, \tag{117}$$

$$\beta_{i-1,j} \mathrel{+}= \beta_{ij} \cdot w_{ij}^{\text{del}}, \tag{118}$$

$$\beta_{i,j-1} \mathrel{+}= \beta_{ij} \cdot w_{ij}^{\text{ins}}. \tag{119}$$

Edit marginal: $P_{ij}^{\text{sub}} = \beta_{ij} \cdot w_{ij}^{\text{sub}}$ (probability of substitution at $(i,j)$).

**Parameter Gradients.**

$$\frac{\partial V_T}{\partial c_i} = \sum_{ij} \beta_{ij} w_{ij}^{\text{ins}} + \sum_{j=1}^{m} \beta_{0,j} = \mathbb{E}(N_{\text{insertions}}), \tag{120}$$

$$\frac{\partial V_T}{\partial c_d} = \sum_{ij} \beta_{ij} w_{ij}^{\text{del}} + \sum_{i=1}^{n} \beta_{i,0} = \mathbb{E}(N_{\text{deletions}}). \tag{121}$$

The boundary edges (top row, left column) carry a single edit each; weighting them by the outside value $\beta_{0,j}$ (resp. $\beta_{i,0}$) gives their marginal probability, matching the Needleman–Wunsch gap gradient.

**HVP Derivation.**   Forward tangent with substitution cost perturbation $V$:

$$\dot{\alpha}_{ij} = w^{\text{sub}}(\dot{\alpha}_{i-1,j-1} + V_{ij}) + w^{\text{del}}\dot{\alpha}_{i-1,j} + w^{\text{ins}}\dot{\alpha}_{i,j-1}. \tag{122}$$

Boundary tangents: $\dot{\alpha}_{i,0} = 0$, $\dot{\alpha}_{0,j} = 0$.

Aggregate: $\dot{D}_T = \dot{\alpha}_{n,m}$.

Backward tangent: Initialize $\dot{\beta}_{n,m} = 0$. Weight tangents (softmin):

$$\dot{w}_{ij}^k = -w_{ij}^k(\dot{v}^k - \dot{\alpha}_{ij})/T, \tag{123}$$

where $\dot{v}^{\text{sub}} = \dot{\alpha}_{i-1,j-1} + V_{ij}$, $\dot{v}^{\text{del}} = \dot{\alpha}_{i-1,j}$, $\dot{v}^{\text{ins}} = \dot{\alpha}_{i,j-1}$.

Propagate:

$$\dot{\beta}_{\text{pred}} \mathrel{+}= \dot{\beta}_{ij} w^k + \beta_{ij} \dot{w}^k. \tag{124}$$

Output: $\text{HVP}_{ij} = \dot{\beta}_{ij} w_{ij}^{\text{sub}} + \beta_{ij} \dot{w}_{ij}^{\text{sub}}$ for substitution positions (transition-scored; the softmin sign is carried by $\dot{w}^{\text{sub}}$).

**Cross-Jacobians.**   For $S \in \{c_i, c_d\}$:

Forward $U$ pass:
$$U_{ij} = w^{\text{sub}}U_{i-1,j-1} + w^{\text{del}}(U_{i-1,j} + \mathbf{1}[S = c_d]) + w^{\text{ins}}(U_{i,j-1} + \mathbf{1}[S = c_i]). \tag{125}$$

Boundaries: $U_{i,0} = i \cdot \mathbf{1}[S = c_d]$, $U_{0,j} = j \cdot \mathbf{1}[S = c_i]$.

Aggregate: $\partial V_T/\partial S = U_{n,m}$.

Backward $W$ pass: Initialize $W_{n,m} = 0$. Weight tangents from $U$:

$$\dot{w}_{ij}^k = -w_{ij}^k(U_{\text{pred}} + \mathbf{1}[S \text{ affects } k] - U_{ij})/T. \tag{126}$$

Propagate: $W_{\text{pred}} \mathrel{+}= W_{ij} w^k + \beta_{ij} \dot{w}^k$.

Output: $[\partial P_T/\partial S]_{ij} = W_{ij}$.

**Covariance Interpretation.**

$$\frac{\partial P_{T,ij}^{\text{sub}}}{\partial c_i} = -\frac{1}{T}\text{Cov}_{q_T}(\mathbf{1}\{\text{sub at } (i,j)\}, N_{\text{ins}}). \tag{127}$$

For minimization: positive covariance means the marginal *decreases* when cost increases (higher insertion cost makes paths with more insertions less likely, reducing competition at this position).

**Temperature Cross-Jacobian.**   Forward $U^T$ pass:

$$U_{ij}^T = w^{\text{sub}}U_{i-1,j-1}^T + w^{\text{del}}U_{i-1,j}^T + w^{\text{ins}}U_{i,j-1}^T - (\alpha_{ij} - \bar{\alpha}_{ij})/T, \tag{128}$$

where $\bar{\alpha}_{ij} = w^{\text{sub}}(\alpha_{i-1,j-1} + C_{ij}) + w^{\text{del}}(\alpha_{i-1,j} + c_d) + w^{\text{ins}}(\alpha_{i,j-1} + c_i)$.

The negative sign arises from softmin (minimization).

Backward $W^T$ pass with entropy-weighted tangents.

$$\frac{\partial P_{T,ij}}{\partial T} = \frac{1}{T^2} \text{Cov}_{q_T}\big(\mathbf{1}\{(i,j) \in \text{edit path}\}, \text{cost}(\text{path})\big). \tag{129}$$

### B.3. Optimal String Alignment (OSA)

OSA extends Levenshtein with adjacent transposition. Unlike Damerau–Levenshtein, transposed characters cannot be individually edited afterward.

**DAG Construction.** Node set: $V = \{(i,j) : 0 \le i \le n, 0 \le j \le m\}$. Edges:

- **Substitution**: $(i-1, j-1) \to (i,j)$ with cost $C_{ij}$
- **Deletion**: $(i-1, j) \to (i,j)$ with cost $c_d$
- **Insertion**: $(i, j-1) \to (i,j)$ with cost $c_i$
- **Transposition**: $(i-2, j-2) \to (i,j)$ with cost $c_t$ if $x_i = y_{j-1}$ and $x_{i-1} = y_j$

**Transposition Mask.**

$$T_m[i,j] = \begin{cases} 1 & \text{if } i \ge 2, j \ge 2, x_i = y_{j-1}, x_{i-1} = y_j \\ 0 & \text{otherwise} \end{cases} \tag{130}$$

**Soft Recurrence.**

$$\alpha_{ij} = \text{smin}_T\big(v^{\text{sub}}, v^{\text{del}}, v^{\text{ins}}, v^{\text{trans}}\big), \tag{131}$$

where:

$$v^{\text{sub}} = \alpha_{i-1,j-1} + C_{ij}, \tag{132}$$
$$v^{\text{del}} = \alpha_{i-1,j} + c_d, \tag{133}$$
$$v^{\text{ins}} = \alpha_{i,j-1} + c_i, \tag{134}$$
$$v^{\text{trans}} = \begin{cases} \alpha_{i-2,j-2} + c_t & \text{if } T_m[i,j] = 1 \\ +\infty & \text{otherwise} \end{cases} \tag{135}$$

Boundaries: $\alpha_{0,0} = 0$, $\alpha_{i,0} = i \cdot c_d$, $\alpha_{0,j} = j \cdot c_i$.

**Partition Function.**

$$V_T = \alpha_{n,m}. \tag{136}$$

**Softmin Weights.**

$$w^{\text{sub}}, w^{\text{del}}, w^{\text{ins}}, w^{\text{trans}} = \text{softmin}\big((v^{\text{sub}}, v^{\text{del}}, v^{\text{ins}}, v^{\text{trans}})/T\big). \tag{137}$$

Note: $w^{\text{trans}} = 0$ when transposition is invalid.

**Backward Algorithm.** Initialize $\beta_{n,m} = 1$, all others $= 0$. Propagate backward:

$$\beta_{i-1,j-1} \mathrel{+}= \beta_{ij} \cdot w_{ij}^{\text{sub}}, \tag{138}$$
$$\beta_{i-1,j} \mathrel{+}= \beta_{ij} \cdot w_{ij}^{\text{del}}, \tag{139}$$
$$\beta_{i,j-1} \mathrel{+}= \beta_{ij} \cdot w_{ij}^{\text{ins}}, \tag{140}$$
$$\beta_{i-2,j-2} \mathrel{+}= \beta_{ij} \cdot w_{ij}^{\text{trans}} \quad \text{if } T_m[i,j] = 1. \tag{141}$$

Edit marginals: $P_{ij}^{\text{sub}} = \beta_{ij} w_{ij}^{\text{sub}}$, $P_{ij}^{\text{trans}} = \beta_{ij} w_{ij}^{\text{trans}}$.

**Parameter Gradients.**

$$\frac{\partial V_T}{\partial c_i} = \sum_{ij} \beta_{ij} w_{ij}^{\text{ins}} + (\text{boundary}) = \mathbb{E}(N_{\text{ins}}), \tag{142}$$

$$\frac{\partial V_T}{\partial c_d} = \sum_{ij} \beta_{ij} w_{ij}^{\text{del}} + (\text{boundary}) = \mathbb{E}(N_{\text{del}}), \tag{143}$$

$$\frac{\partial V_T}{\partial c_t} = \sum_{ij:T_m[i,j]=1} \beta_{ij} w_{ij}^{\text{trans}} = \mathbb{E}(N_{\text{trans}}). \tag{144}$$

**HVP Derivation.** Forward tangent:

$$\dot{\alpha}_{ij} = w^{\text{sub}}(\dot{\alpha}_{i-1,j-1} + V_{ij}) + w^{\text{del}}\dot{\alpha}_{i-1,j} + w^{\text{ins}}\dot{\alpha}_{i,j-1} + w^{\text{trans}}\dot{\alpha}_{i-2,j-2}. \tag{145}$$

Backward tangent with weight tangents $\dot{w}^k = -w^k(\dot{v}^k - \dot{\alpha}_{ij})/T$.

**Insertion Cross-Jacobian.** Forward $U^{c_i}$ pass:

$$U_{ij}^{c_i} = w^{\text{sub}}U_{i-1,j-1}^{c_i} + w^{\text{del}}U_{i-1,j}^{c_i} + w^{\text{ins}}(U_{i,j-1}^{c_i} + 1) + w^{\text{trans}}U_{i-2,j-2}^{c_i}. \tag{146}$$

Boundaries: $U_{0,j}^{c_i} = j$.

**Deletion Cross-Jacobian.** Forward $U^{c_d}$ pass:

$$U_{ij}^{c_d} = w^{\text{sub}}U_{i-1,j-1}^{c_d} + w^{\text{del}}(U_{i-1,j}^{c_d} + 1) + w^{\text{ins}}U_{i,j-1}^{c_d} + w^{\text{trans}}U_{i-2,j-2}^{c_d}. \tag{147}$$

Boundaries: $U_{i,0}^{c_d} = i$.

**Transposition Cross-Jacobian.** Forward $U^{c_t}$ pass:

$$U_{ij}^{c_t} = w^{\text{sub}}U_{i-1,j-1}^{c_t} + w^{\text{del}}U_{i-1,j}^{c_t} + w^{\text{ins}}U_{i,j-1}^{c_t} + w^{\text{trans}}(U_{i-2,j-2}^{c_t} + 1). \tag{148}$$

**Backward $W$ Pass (all parameters).** For each $S$, initialize $W_{n,m}^S = 0$. Weight tangents: $\dot{w}^k = -w^k(U_{\text{pred}}^S + \mathbf{1}[S \text{ affects } k] - U_{ij}^S)/T$. Propagate: $W_{\text{pred}}^S \mathrel{+}= W_{ij}^S w^k + \beta_{ij}\dot{w}^k$.

Output: $[\partial P_T/\partial S]_{ij} = W_{ij}^S$.

**Temperature Cross-Jacobian.** Forward $U^T$ pass:

$$U_{ij}^T = w^{\text{sub}}U_{i-1,j-1}^T + w^{\text{del}}U_{i-1,j}^T + w^{\text{ins}}U_{i,j-1}^T + w^{\text{trans}}U_{i-2,j-2}^T - (\alpha_{ij} - \bar{\alpha}_{ij})/T, \tag{149}$$

where $\bar{\alpha}_{ij}$ is the weighted average of predecessor values.

Backward $W^T$ pass with entropy-weighted tangents.

$$\frac{\partial P_{T,ij}}{\partial T} = \frac{1}{T^2}\text{Cov}_{q_T}\big(\mathbf{1}\{(i,j) \in \text{edit path}\}, \text{cost}(\text{path})\big). \tag{150}$$

## B.4. Damerau–Levenshtein

True Damerau–Levenshtein allows unrestricted transpositions (can transpose across intervening insertions/deletions).

**DAG Construction.** The node set is $V = \{(i,j) : 0 \le i \le n, 0 \le j \le m\}$. In addition to standard Levenshtein edges, transposition edges connect $(k,l) \to (i,j)$ for variable $(k,l)$ where the intervening characters form a valid transposition.

**Auxiliary Data Structure.** For each character $c$, maintain $\text{DA}[c] = $ last row where $c$ appeared. Also track $\text{DB} = $ last column where a match occurred in current row.

**Soft Recurrence.** Let $k = \text{DA}[y_j]$ and $l = \text{DB}$ from before position $j$.

$$\alpha_{ij} = \text{smin}_T\left(\alpha_{i-1,j-1} + C_{ij}, \alpha_{i-1,j} + c_d, \alpha_{i,j-1} + c_i,\right.$$
$$\left.\alpha_{k-1,l-1} + (i - k - 1)\cdot c_d + c_t + (j - l - 1)\cdot c_i\right). \tag{151}$$

The transposition term accounts for: deleting characters $k+1, \ldots, i-1$, the transposition cost, and inserting characters $l+1, \ldots, j-1$.

Boundaries: $\alpha_{i,0} = i \cdot c_d$, $\alpha_{0,j} = j \cdot c_i$.

**Softmin Weights.**
$$w^{\text{sub}}, w^{\text{del}}, w^{\text{ins}}, w^{\text{trans}} = \text{softmin}\left((v^{\text{sub}}, v^{\text{del}}, v^{\text{ins}}, v^{\text{trans}})/T\right), \tag{152}$$

where the $v$ values are the four options in the recurrence.

**Backward Algorithm.** Initialize $\beta_{n,m} = 1$. Propagate backward:

$$\beta_{i-1,j-1} \mathrel{+}= \beta_{ij}w^{\text{sub}}, \tag{153}$$
$$\beta_{i-1,j} \mathrel{+}= \beta_{ij}w^{\text{del}}, \tag{154}$$
$$\beta_{i,j-1} \mathrel{+}= \beta_{ij}w^{\text{ins}}, \tag{155}$$
$$\beta_{k-1,l-1} \mathrel{+}= \beta_{ij}w^{\text{trans}}. \tag{156}$$

Alignment marginal: $P_{T,ij} = \beta_{ij}$.

**Parameter Gradients.**

$$\frac{\partial V_T}{\partial c_i} = \sum_{ij} \beta_{ij}\left(w_{ij}^{\text{ins}} + w_{ij}^{\text{trans}} \cdot (j - l_{ij} - 1)\right), \tag{157}$$

$$\frac{\partial V_T}{\partial c_d} = \sum_{ij} \beta_{ij}\left(w_{ij}^{\text{del}} + w_{ij}^{\text{trans}} \cdot (i - k_{ij} - 1)\right), \tag{158}$$

$$\frac{\partial V_T}{\partial c_t} = \sum_{ij} \beta_{ij}w_{ij}^{\text{trans}}. \tag{159}$$

**HVP Derivation.** Forward tangent with cost perturbation $V$:

$$\dot{\alpha}_{ij} = w^{\text{sub}}(\dot{\alpha}_{i-1,j-1} + V_{ij}) + w^{\text{del}}\dot{\alpha}_{i-1,j} + w^{\text{ins}}\dot{\alpha}_{i,j-1} + w^{\text{trans}}\dot{\alpha}_{k-1,l-1}. \tag{160}$$

Backward tangent: Propagate $\dot{\beta}$ with weight tangents $\dot{w} = -w(\dot{v} - \dot{\alpha}_{ij})/T$ (negative sign for softmin) where $\dot{v}$ includes the appropriate tangent values.

**Cross-Jacobians.** For $S \in \{c_i, c_d, c_t\}$:

Forward $U$ pass:

$$U_{ij} = w^{\text{sub}}U_{i-1,j-1} + w^{\text{del}}(U_{i-1,j} + \mathbf{1}[S = c_d]) \tag{161}$$
$$+ w^{\text{ins}}(U_{i,j-1} + \mathbf{1}[S = c_i])$$
$$+ w^{\text{trans}}\left(U_{k-1,l-1} + (i - k - 1)\mathbf{1}[S = c_d] + \mathbf{1}[S = c_t] + (j - l - 1)\mathbf{1}[S = c_i]\right).$$

Backward $W$ pass: Initialize $W_{ij} = \beta_{ij}^{(0)}(U_{ij} - \partial V_T/\partial S)/T$, propagate.

Output: $[\partial P_T/\partial S]_{ij} = W_{ij}$.

**Temperature Cross-Jacobian.** Forward $U^T$ pass:

$$U_{ij}^T = w^{\text{sub}} U_{i-1,j-1}^T + w^{\text{del}} U_{i-1,j}^T + w^{\text{ins}} U_{i,j-1}^T + w^{\text{trans}} U_{k-1,l-1}^T - (\alpha_{ij} - \bar{\alpha}_{ij})/T, \tag{162}$$

where $\bar{\alpha}_{ij}$ is the weighted average of predecessor values including the variable-distance transposition.

Backward $W^T$ pass with entropy-weighted tangents.

$$\frac{\partial P_{T,ij}}{\partial T} = \frac{1}{T^2} \text{Cov}_{q_T} \big( \mathbf{1}\{(i,j) \in \text{edit path}\}, \text{cost(path)} \big). \tag{163}$$

## C. Parsing Algorithms

Parsing algorithms have $O(n^3)$ complexity and operate over chart structures. Their soft marginals are visualized in Figure 4 (Section G).

### C.1. CKY (Cocke–Kasami–Younger)

CKY parses context-free grammars in Chomsky normal form. We extend the formulation with a learnable span length penalty $\lambda$ that adds $\lambda \cdot (j - i)$ to each span score, enabling learning of tree depth preferences.

**Chart Structure.** The chart $Z[i, j]$ represents the inside score for span $[i, j]$. Merge scores $M[i, k, j]$ give the score for combining spans $[i, k]$ and $[k + 1, j]$.

**Soft Recurrence.** Base case: $Z[i, i] = \text{leaf}_i$ (terminal score).

Induction (increasing span length):

$$Z[i, j] = \text{LSE}_T \big( \{ Z[i, k] + Z[k + 1, j] + M[i, k, j] + \lambda \cdot (j - i) \}_{k=i}^{j-1} \big). \tag{164}$$

**Partition Function.** $V_T = Z[0, n - 1]$ (root span).

**Backward Algorithm (Outside).** Initialize $\beta[0, n - 1] = 1$. Top-down by decreasing span length:

$$w_k = \text{softmax} \big( (Z[i, k] + Z[k + 1, j] + M[i, k, j])_{k=i}^{j-1} / T \big)_k. \tag{165}$$

Propagate:

$$\beta[i, k] \mathrel{+}= \beta[i, j] \cdot w_k, \quad \beta[k + 1, j] \mathrel{+}= \beta[i, j] \cdot w_k. \tag{166}$$

**Marginals.**

- Span marginal: $P(\text{span } [i, j]) = \beta[i, j]$

- Split marginal: $P(k \mid i, j) = w_k$ (conditional on span)

- Joint split: $P(\text{split at } k \text{ in } [i, j]) = \beta[i, j] \cdot w_k$

**Gradients.**

$$\frac{\partial V_T}{\partial M[i, k, j]} = \beta[i, j] \cdot w_k. \tag{167}$$

**HVP.** Forward tangent (increasing span):

$$\dot{Z}[i, j] = \sum_{k=i}^{j-1} w_k \cdot (\dot{Z}[i, k] + \dot{Z}[k + 1, j] + V[i, k, j]). \tag{168}$$

Backward tangent (decreasing span):

$$\dot{\beta}[i, k] \mathrel{+}= \dot{\beta}[i, j] \cdot w_k + \beta[i, j] \cdot \dot{w}_k. \tag{169}$$

**Temperature Cross-Jacobian.** Forward $U^T$ pass includes entropy: $U^T[i,j] \mathrel{+}= (Z[i,j] - \bar{Z}[i,j])/T$.

**Span Length Penalty Cross-Jacobian.** For the span length penalty $\lambda$, we compute $\partial P_T / \partial \lambda$.

Forward $U^\lambda$ pass accumulates span lengths:

$$U^\lambda[i,j] = \sum_{k=i}^{j-1} w_k \cdot (U^\lambda[i,k] + U^\lambda[k+1,j] + (j-i)). \tag{170}$$

Aggregate: $\partial V_T / \partial \lambda = U^\lambda[0, n-1] = \mathbb{E}_{p_T}[\sum_{(i,j) \in Y}(j-i)]$ (expected total span length).

Covariance interpretation:

$$\frac{\partial P_T[i,j]}{\partial \lambda} = \frac{1}{T} \mathrm{Cov}_{p_T}\big(\mathbf{1}\{[i,j] \in Y\}, \textstyle\sum_{(i',j') \in Y}(j'-i')\big). \tag{171}$$

### C.2. Eisner's Algorithm

Eisner's algorithm (Eisner, 1996) computes the partition function over projective dependency trees. We extend the formulation with learnable arc length penalty $\lambda$ (adds $\lambda \cdot |i-j|$ to arc scores, penalizing long-distance dependencies) and direction bias $\delta$ (adds $\delta$ to rightward arcs).

**Four-State Decomposition.**

- $C_R[i,j]$: Complete right span (head at $i$, all dependents attached)

- $C_L[i,j]$: Complete left span (head at $j$, all dependents attached)

- $I_R[i,j]$: Incomplete right span (arc $i \to j$, right children of $j$ pending)

- $I_L[i,j]$: Incomplete left span (arc $j \to i$, left children of $i$ pending)

**Soft Recurrences.** Base: $C_R[i,i] = C_L[i,i] = 0$.

Incomplete spans (attach arc with length penalty $\lambda$ and direction bias $\delta$):

$$I_R[i,j] = \mathrm{arc}[i,j] + \lambda|j-i| + \delta + \mathrm{LSE}_T\big(\{C_R[i,k] + C_L[k+1,j]\}_{k=i}^{j-1}\big), \tag{172}$$

$$I_L[i,j] = \mathrm{arc}[j,i] + \lambda|j-i| + \mathrm{LSE}_T\big(\{C_R[i,k] + C_L[k+1,j]\}_{k=i}^{j-1}\big). \tag{173}$$

Note: $\delta$ is added only to rightward arcs ($I_R$); leftward arcs ($I_L$) do not include it.

Complete spans (combine):

$$C_R[i,j] = \mathrm{LSE}_T\big(\{C_R[i,k] + I_R[k,j]\}_{k=i}^{j-1}\big), \tag{174}$$

$$C_L[i,j] = \mathrm{LSE}_T\big(\{I_L[i,k] + C_L[k,j]\}_{k=i+1}^{j}\big). \tag{175}$$

**Partition Function.** $V_T = C_R[0, n-1]$ (full sentence, root at position 0).

**Arc Marginals.**

$$P(\mathrm{arc}\ i \to j) = \beta_{I_R}[i,j], \quad P(\mathrm{arc}\ j \to i) = \beta_{I_L}[i,j]. \tag{176}$$

**Backward Algorithm (Complete).** Initialize: $\beta_{C_R}[0, n-1] = 1$, all others $= 0$.

For incomplete span formation, define softmax weights over split points:

$$w_k^I[i,j] = \mathrm{softmax}\big(\{C_R[i,k'] + C_L[k'+1,j]\}_{k'=i}^{j-1}/T\big)_k. \tag{177}$$

For complete span formation:

$$w_k^{C_R}[i,j] = \text{softmax}\left(\{C_R[i,k'] + I_R[k',j]\}_{k'=i}^{j-1}/T\right)_k, \tag{178}$$

$$w_k^{C_L}[i,j] = \text{softmax}\left(\{I_L[i,k'] + C_L[k',j]\}_{k'=i+1}^{j}/T\right)_k. \tag{179}$$

Propagate top-down by decreasing span length $\ell = j - i$:

**Step 1: Complete to Incomplete.**

$$\beta_{I_R}[k,j] \mathrel{+}= \beta_{C_R}[i,j] \cdot w_k^{C_R}[i,j] \quad \text{for } k = i, \ldots, j-1, \tag{180}$$

$$\beta_{C_R}[i,k] \mathrel{+}= \beta_{C_R}[i,j] \cdot w_k^{C_R}[i,j], \tag{181}$$

$$\beta_{I_L}[i,k] \mathrel{+}= \beta_{C_L}[i,j] \cdot w_k^{C_L}[i,j] \quad \text{for } k = i+1, \ldots, j, \tag{182}$$

$$\beta_{C_L}[k,j] \mathrel{+}= \beta_{C_L}[i,j] \cdot w_k^{C_L}[i,j]. \tag{183}$$

**Step 2: Incomplete to Complete.**

$$\beta_{C_R}[i,k] \mathrel{+}= \beta_{I_R}[i,j] \cdot w_k^{I}[i,j] \quad \text{for } k = i, \ldots, j-1, \tag{184}$$

$$\beta_{C_L}[k+1,j] \mathrel{+}= \beta_{I_R}[i,j] \cdot w_k^{I}[i,j], \tag{185}$$

$$\beta_{C_R}[i,k] \mathrel{+}= \beta_{I_L}[i,j] \cdot w_k^{I}[i,j], \tag{186}$$

$$\beta_{C_L}[k+1,j] \mathrel{+}= \beta_{I_L}[i,j] \cdot w_k^{I}[i,j]. \tag{187}$$

**Arc Score Gradients.**

$$\frac{\partial V_T}{\partial A[i,j]} = \beta_{I_R}[i,j] = P(\text{arc } i \to j), \tag{188}$$

$$\frac{\partial V_T}{\partial A[j,i]} = \beta_{I_L}[i,j] = P(\text{arc } j \to i). \tag{189}$$

**HVP Derivation.**    Forward tangent for incomplete spans:

$$\dot{I}_R[i,j] = V[i,j] + \sum_{k=i}^{j-1} w_k^{I}[i,j] \cdot (\dot{C}_R[i,k] + \dot{C}_L[k+1,j]), \tag{190}$$

and similarly for $\dot{I}_L$.

Forward tangent for complete spans:

$$\dot{C}_R[i,j] = \sum_{k=i}^{j-1} w_k^{C_R}[i,j] \cdot (\dot{C}_R[i,k] + \dot{I}_R[k,j]), \tag{191}$$

and similarly for $\dot{C}_L$.

Backward tangent propagates through all four states with weight tangents:

$$\dot{w}_k^{I} = w_k^{I}(\dot{C}_R[i,k] + \dot{C}_L[k+1,j] - \dot{I})/T, \tag{192}$$

where $\dot{I} = \sum_{k'} w_{k'}^{I}(\dot{C}_R[i,k'] + \dot{C}_L[k'+1,j])$.

**Temperature Cross-Jacobian.**    Forward $U^T$ pass includes entropy terms at each combination step:

$$U_{I_R}^T[i,j] = \sum_k w_k^{I}(U_{C_R}^T[i,k] + U_{C_L}^T[k+1,j]) + (I_R[i,j] - A[i,j] - \bar{I}[i,j])/T, \tag{193}$$

where $\bar{I}[i,j] = \sum_k w_k^{I}(C_R[i,k] + C_L[k+1,j])$ is the weighted average.

Similar entropy terms for $U_{C_R}^T$ and $U_{C_L}^T$.

Backward $W^T$ pass with entropy-weighted tangents.

**Covariance Interpretation.**

$$\frac{\partial P_T(\text{arc } i \to j)}{\partial T} = -\frac{1}{T^2}\text{Cov}_{p_T}\big(\mathbf{1}\{i \to j \in \text{tree}\}, \text{score}(\text{tree})\big). \tag{194}$$

Arcs whose inclusion covaries positively with tree score have marginals that decrease as temperature increases.

**Arc Length Penalty Cross-Jacobian.** For the arc length penalty $\lambda$, we compute $\partial P_T/\partial\lambda$.

Forward $U^\lambda$ pass accumulates arc lengths through the four-state decomposition:

$$U^\lambda_{I_R}[i,j] = |j - i| + \sum_k w^I_k(U^\lambda_{C_R}[i,k] + U^\lambda_{C_L}[k+1,j]), \tag{195}$$

$$U^\lambda_{I_L}[i,j] = |j - i| + \sum_k w^I_k(U^\lambda_{C_R}[i,k] + U^\lambda_{C_L}[k+1,j]), \tag{196}$$

$$U^\lambda_{C_R}[i,j] = \sum_k w^{C_R}_k(U^\lambda_{C_R}[i,k] + U^\lambda_{I_R}[k,j]), \tag{197}$$

$$U^\lambda_{C_L}[i,j] = \sum_k w^{C_L}_k(U^\lambda_{I_L}[i,k] + U^\lambda_{C_L}[k,j]). \tag{198}$$

Aggregate: $\partial V_T/\partial\lambda = U^\lambda_{C_R}[0, n-1] = \mathbb{E}_{p_T}[\sum_{(h,d)\in\text{tree}} |h - d|]$ (expected total arc length).

Covariance interpretation:

$$\frac{\partial P_T(\text{arc } i \to j)}{\partial\lambda} = \frac{1}{T}\text{Cov}_{p_T}\big(\mathbf{1}\{i \to j \in \text{tree}\}, \sum_{(h,d)\in\text{tree}} |h - d|\big). \tag{199}$$

**Direction Bias Cross-Jacobian.** For the direction bias $\delta$ (added only to rightward arcs), we compute $\partial P_T/\partial\delta$.

Forward $U^\delta$ pass counts rightward arcs:

$$U^\delta_{I_R}[i,j] = 1 + \sum_k w^I_k(U^\delta_{C_R}[i,k] + U^\delta_{C_L}[k+1,j]), \tag{200}$$

$$U^\delta_{I_L}[i,j] = \sum_k w^I_k(U^\delta_{C_R}[i,k] + U^\delta_{C_L}[k+1,j]), \tag{201}$$

$$U^\delta_{C_R}[i,j] = \sum_k w^{C_R}_k(U^\delta_{C_R}[i,k] + U^\delta_{I_R}[k,j]), \tag{202}$$

$$U^\delta_{C_L}[i,j] = \sum_k w^{C_L}_k(U^\delta_{I_L}[i,k] + U^\delta_{C_L}[k,j]). \tag{203}$$

Aggregate: $\partial V_T/\partial\delta = U^\delta_{C_R}[0, n-1] = \mathbb{E}_{p_T}[N_{\text{right}}]$ (expected number of rightward arcs).

Covariance interpretation:

$$\frac{\partial P_T(\text{arc } i \to j)}{\partial\delta} = \frac{1}{T}\text{Cov}_{p_T}\big(\mathbf{1}\{i \to j \in \text{tree}\}, N_{\text{right}}\big). \tag{204}$$

# D. Pseudocode

We provide systematic pseudocode for each algorithm family.

## D.1. Template Structure

Each algorithm has four computational passes:

1. **Forward**: Compute $\alpha$ tables and partition $V_T$

2. **Backward**: Compute $\beta$ tables, marginals, parameter gradients

3. **HVP**: Forward tangent $\dot{\alpha}$, backward tangent $\dot{\beta}$

4. **Cross-Jacobian**: Forward $U$, backward $W$

## D.2. Alignment Pseudocode

---
**Algorithm 1** Linear Smith–Waterman: Forward Pass
---

**Input:** Scores $S \in \mathbb{R}^{n \times m}$, gap $g$, temperature $T$
**Output:** Forward table $\alpha \in \mathbb{R}^{(n+1) \times (m+1)}$, partition $V_T$
Initialize: $\alpha_{0,0} \leftarrow 0$; $\alpha_{i,0}, \alpha_{0,j} \leftarrow -\infty$ for $i, j > 0$
**for** $k = 2$ **to** $n + m$ **do**
    **for** $(i, j)$ on anti-diagonal $k$ **in parallel do**
        $\alpha_{ij} \leftarrow \mathrm{LSE}_T(\alpha_{i-1,j-1} + S_{ij}, \alpha_{i-1,j} + g, \alpha_{i,j-1} + g, 0)$
    **end for**
**end for**
$V_T \leftarrow \mathrm{LSE}_T(\{\alpha_{ij}\}_{i \geq 1, j \geq 1})$
**return** $\alpha, V_T$

---

---
**Algorithm 2** Linear Smith–Waterman: Backward Pass
---

**Input:** Forward $\alpha$, partition $V_T$, scores $S$, gap $g$, temperature $T$
**Output:** Marginals $P \in \mathbb{R}^{n \times m}$, gap gradient $\partial V_T / \partial g$
Initialize: $\beta_{ij} \leftarrow \exp((\alpha_{ij} - V_T)/T)$ for all $(i, j)$
$G \leftarrow 0$
**for** $k = n + m$ **down to** $2$ **do**
    **for** $(i, j)$ on anti-diagonal $k$ **in parallel do**
        $w \leftarrow \mathrm{softmax}((\alpha_{i-1,j-1} + S_{ij}, \alpha_{i-1,j} + g, \alpha_{i,j-1} + g, 0)/T)$
        $P_{T,ij} \leftarrow \beta_{ij} \cdot w_1$ {Match marginal}
        $\beta_{i-1,j-1} \mathrel{+}= \beta_{ij} \cdot w_1$
        $\beta_{i-1,j} \mathrel{+}= \beta_{ij} \cdot w_2;$    $G \mathrel{+}= \beta_{ij} \cdot w_2$
        $\beta_{i,j-1} \mathrel{+}= \beta_{ij} \cdot w_3;$    $G \mathrel{+}= \beta_{ij} \cdot w_3$
    **end for**
**end for**
**return** $P, G$

---

---

**Algorithm 3** Linear Smith–Waterman: Score HVP

---

**Input:** Forward $\alpha$, partition $V_T$, vector $V \in \mathbb{R}^{n \times m}$, params
**Output:** HVP $\nabla_S^2 V_T \cdot V \in \mathbb{R}^{n \times m}$
*// Forward tangent*
Initialize: $\dot{\alpha}_{ij} \leftarrow 0$ for all $(i, j)$
**for** $k = 2$ **to** $n + m$ **do**
   **for** $(i, j)$ on anti-diagonal $k$ **do**
      $w \leftarrow \mathrm{softmax}((\alpha_{i-1,j-1} + S_{ij}, \alpha_{i-1,j} + g, \alpha_{i,j-1} + g, 0)/T)$
      $\dot{\alpha}_{ij} \leftarrow w_1(\dot{\alpha}_{i-1,j-1} + V_{ij}) + w_2 \dot{\alpha}_{i-1,j} + w_3 \dot{\alpha}_{i,j-1}$
   **end for**
**end for**
$\dot{H}_T \leftarrow \sum_{ij} \beta_{ij}^{(0)} \cdot \dot{\alpha}_{ij}$
*// Backward tangent*
Initialize: $\dot{\beta}_{ij} \leftarrow \beta_{ij}^{(0)} (\dot{\alpha}_{ij} - \dot{H}_T)/T$
**for** $k = n + m$ **down to** 2 **do**
   **for** $(i, j)$ on anti-diagonal $k$ **do**
      Compute $\dot{w}_k = w_k(\dot{v}_k - \dot{\alpha}_{ij})/T$ for each transition
      $\mathrm{HVP}_{ij} \leftarrow \dot{\beta}_{ij} w_1 + \beta_{ij} \dot{w}_1$ {transition-scored: $P_{ij} = \beta_{ij} w_1$}
      Propagate: $\dot{\beta}_{\mathrm{pred}} \mathrel{+}= \dot{\beta}_{ij} w_k + \beta_{ij} \dot{w}_k$
   **end for**
**end for**
**return** HVP

---

**Algorithm 4** Linear Smith–Waterman: Gap Cross-Jacobian

---

**Input:** Forward $\alpha$, partition $V_T$, params
**Output:** Cross-Jacobian $\partial P_T / \partial g \in \mathbb{R}^{n \times m}$
*// Forward U pass*
Initialize: $U_{ij} \leftarrow 0$ for all $(i, j)$
**for** $k = 2$ **to** $n + m$ **do**
   **for** $(i, j)$ on anti-diagonal $k$ **do**
      $w \leftarrow \mathrm{softmax}((\alpha_{i-1,j-1} + S_{ij}, \alpha_{i-1,j} + g, \alpha_{i,j-1} + g, 0)/T)$
      $U_{ij} \leftarrow w_1 U_{i-1,j-1} + w_2(U_{i-1,j} + 1) + w_3(U_{i,j-1} + 1)$
   **end for**
**end for**
$\partial V_T / \partial g \leftarrow \sum_{ij} \beta_{ij}^{(0)} U_{ij}$
*// Backward W pass*
Initialize: $W_{ij} \leftarrow \beta_{ij}^{(0)} (U_{ij} - \partial V_T / \partial g)/T$
**for** $k = n + m$ **down to** 2 **do**
   **for** $(i, j)$ on anti-diagonal $k$ **do**
      Compute weight tangents $\dot{w}_k$ from $U$ values
      $J_{ij} \leftarrow W_{ij}$
      Propagate: $W_{\mathrm{pred}} \mathrel{+}= W_{ij} w_k + \beta_{ij} \dot{w}_k$
   **end for**
**end for**
**return** $J = \partial P_T / \partial g$

---

---

**Algorithm 5** Affine Smith–Waterman: Forward Pass

---

**Input:** Scores $S$, gap penalties $g_o, g_e$, temperature $T$
**Output:** Forward tables $\alpha^M, \alpha^I, \alpha^D$, partition $V_T$
Initialize: $\alpha^M_{0,0} \leftarrow 0$; all other $\alpha^\sigma_{ij} \leftarrow -\infty$
**for** $k = 2$ **to** $n + m$ **do**
    **for** $(i, j)$ on anti-diagonal $k$ **in parallel do**
        $\alpha^M_{ij} \leftarrow \mathrm{LSE}_T(\alpha^M_{i-1,j-1}, \alpha^I_{i-1,j-1}, \alpha^D_{i-1,j-1}, 0) + S_{ij}$
        $\alpha^I_{ij} \leftarrow \mathrm{LSE}_T(\alpha^M_{i-1,j} + g_o, \alpha^I_{i-1,j} + g_e)$
        $\alpha^D_{ij} \leftarrow \mathrm{LSE}_T(\alpha^M_{i,j-1} + g_o, \alpha^D_{i,j-1} + g_e)$
    **end for**
**end for**
$V_T \leftarrow \mathrm{LSE}_T(\{\alpha^\sigma_{ij}\}_{i,j,\sigma})$
**return** $\alpha^M, \alpha^I, \alpha^D, V_T$

---

**Algorithm 6** Affine Smith–Waterman: Backward Pass

---

**Input:** Forward tables $\alpha^M, \alpha^I, \alpha^D$, partition $V_T$, params
**Output:** Marginals $P$, gradients $\partial V_T / \partial g_o$, $\partial V_T / \partial g_e$
Initialize: $\beta^\sigma_{ij} \leftarrow \exp((\alpha^\sigma_{ij} - V_T)/T)$ for all $(i, j, \sigma)$
$G_o, G_e \leftarrow 0$
**for** $k = n + m$ **down to** 2 **do**
    **for** $(i, j)$ on anti-diagonal $k$ **in parallel do**
        $w^M \leftarrow \mathrm{softmax}((\alpha^M_{i-1,j-1}, \alpha^I_{i-1,j-1}, \alpha^D_{i-1,j-1}, 0)/T)$
        $w^I \leftarrow \mathrm{softmax}((\alpha^M_{i-1,j} + g_o, \alpha^I_{i-1,j} + g_e)/T)$
        $w^D \leftarrow \mathrm{softmax}((\alpha^M_{i,j-1} + g_o, \alpha^D_{i,j-1} + g_e)/T)$
        $P_{T,ij} \leftarrow \beta^M_{ij}$
        $\beta^M_{i-1,j-1} \mathrel{+}= \beta^M_{ij} w^M_1; \beta^I_{i-1,j-1} \mathrel{+}= \beta^M_{ij} w^M_2; \beta^D_{i-1,j-1} \mathrel{+}= \beta^M_{ij} w^M_3$
        $\beta^M_{i-1,j} \mathrel{+}= \beta^I_{ij} w^I_1; \beta^I_{i-1,j} \mathrel{+}= \beta^I_{ij} w^I_2$
        $\beta^M_{i,j-1} \mathrel{+}= \beta^D_{ij} w^D_1; \beta^D_{i,j-1} \mathrel{+}= \beta^D_{ij} w^D_2$
        $G_o \mathrel{+}= \beta^I_{ij} w^I_1 + \beta^D_{ij} w^D_1$
        $G_e \mathrel{+}= \beta^I_{ij} w^I_2 + \beta^D_{ij} w^D_2$
    **end for**
**end for**
**return** $P, G_o, G_e$

---

---

**Algorithm 7** Affine Smith–Waterman: Score HVP

---

**Input:** Forward tables, partition, vector $V$, params
**Output:** HVP $\nabla_S^2 V_T \cdot V$
*// Forward tangent (3-state)*
**for** $k = 2$ **to** $n + m$ **do**
    **for** $(i, j)$ on anti-diagonal $k$ **do**
        $\dot{\alpha}_{ij}^M \leftarrow \sum_\sigma w^{\sigma \to M}(\dot{\alpha}_{i-1,j-1}^\sigma + V_{ij}) + w^{\text{sky}} V_{ij}$
        $\dot{\alpha}_{ij}^I \leftarrow w^{M \to I} \dot{\alpha}_{i-1,j}^M + w^{I \to I} \dot{\alpha}_{i-1,j}^I$
        $\dot{\alpha}_{ij}^D \leftarrow w^{M \to D} \dot{\alpha}_{i,j-1}^M + w^{D \to D} \dot{\alpha}_{i,j-1}^D$
    **end for**
**end for**
Aggregate: $\dot{H}_T \leftarrow \sum_\sigma w_t^\sigma \dot{\alpha}_{n,m}^\sigma$
*// Backward tangent*
Initialize: $\dot{\beta}_{ij}^\sigma \leftarrow \beta_{ij}^{(0),\sigma}(\dot{\alpha}_{ij}^\sigma - \dot{H}_T)/T$
**for** $k = n + m$ **down to** $2$ **do**
    **for** $(i, j)$ on anti-diagonal $k$ **do**
        Compute weight tangents $\dot{w}$ from $\dot{\alpha}$ values
        $\text{HVP}_{ij} \leftarrow \dot{\beta}_{ij}^M$
        Propagate: $\dot{\beta}_{\text{pred}}^\sigma \mathrel{+}= \dot{\beta}_{ij} w + \beta_{ij} \dot{w}$
    **end for**
**end for**
**return** HVP

---

**Algorithm 8** Affine Smith–Waterman: Gap-Open Cross-Jacobian

---

**Input:** Forward tables, partition, params
**Output:** Cross-Jacobian $\partial P_T / \partial g_o$
*// Forward U pass*
**for** $k = 2$ **to** $n + m$ **do**
    **for** $(i, j)$ on anti-diagonal $k$ **do**
        $U_{ij}^M \leftarrow \sum_\sigma w^{\sigma \to M} U_{i-1,j-1}^\sigma$
        $U_{ij}^I \leftarrow w^{M \to I}(U_{i-1,j}^M + 1) + w^{I \to I} U_{i-1,j}^I$
        $U_{ij}^D \leftarrow w^{M \to D}(U_{i,j-1}^M + 1) + w^{D \to D} U_{i,j-1}^D$
    **end for**
**end for**
$\partial V_T / \partial g_o \leftarrow \sum_\sigma w_t^\sigma U_{n,m}^\sigma$
*// Backward W pass*
Initialize: $W_{ij}^\sigma \leftarrow \beta_{ij}^{(0),\sigma}(U_{ij}^\sigma - \partial V_T / \partial g_o)/T$
**for** $k = n + m$ **down to** $2$ **do**
    **for** $(i, j)$ on anti-diagonal $k$ **do**
        Compute weight tangents from $U$ values
        $J_{ij} \leftarrow W_{ij}^M$
        Propagate: $W_{\text{pred}}^\sigma \mathrel{+}= W_{ij} w + \beta_{ij} \dot{w}$
    **end for**
**end for**
**return** $J = \partial P_T / \partial g_o$

---

---

**Algorithm 9** Linear Needleman–Wunsch: Forward Pass

---

**Input:** Scores $S \in \mathbb{R}^{n \times m}$, gap $g$, temperature $T$
**Output:** Forward table $\alpha$, partition $V_T$
Initialize: $\alpha_{0,0} \leftarrow 0$; $\alpha_{i,0} \leftarrow i \cdot g$; $\alpha_{0,j} \leftarrow j \cdot g$
**for** $k = 2$ **to** $n + m$ **do**
   **for** $(i, j)$ on anti-diagonal $k$ **in parallel do**
      $\alpha_{ij} \leftarrow \text{LSE}_T(\alpha_{i-1,j-1} + S_{ij}, \alpha_{i-1,j} + g, \alpha_{i,j-1} + g)$
   **end for**
**end for**
$V_T \leftarrow \alpha_{n,m}$
**return** $\alpha, V_T$

---

---

**Algorithm 10** Linear Needleman–Wunsch: Backward Pass

---

**Input:** Forward $\alpha$, partition $V_T$, params
**Output:** Marginals $P$, gap gradient $\partial V_T / \partial g$
Initialize: $\beta_{n,m} \leftarrow 1$; all others $\leftarrow 0$
$G \leftarrow 0$
**for** $k = n + m$ **down to** $2$ **do**
   **for** $(i, j)$ on anti-diagonal $k$ **in parallel do**
      $w \leftarrow \text{softmax}((\alpha_{i-1,j-1} + S_{ij}, \alpha_{i-1,j} + g, \alpha_{i,j-1} + g)/T)$
      $P_{T,ij} \leftarrow \beta_{ij} \cdot w_1$ {Match marginal}
      $\beta_{i-1,j-1} \mathrel{+}= \beta_{ij} w_1$
      $\beta_{i-1,j} \mathrel{+}= \beta_{ij} w_2$; $G \mathrel{+}= \beta_{ij} w_2$
      $\beta_{i,j-1} \mathrel{+}= \beta_{ij} w_3$; $G \mathrel{+}= \beta_{ij} w_3$
   **end for**
**end for**
Add boundary contributions to $G$
**return** $P, G$

---

---

**Algorithm 11** Linear Needleman–Wunsch: Score HVP

---

**Input:** Forward $\alpha$, partition $V_T$, vector $V$, gap $g$, temperature $T$
**Output:** HVP $\nabla_S^2 V_T \cdot V$
*// Forward tangent*
Initialize: $\dot{\alpha}_{ij} \leftarrow 0$; $\dot{\alpha}_{i,0} = \dot{\alpha}_{0,j} = 0$ {boundary is pure gaps; match-score perturbation does not reach it}
**for** $k = 2$ **to** $n + m$ **do**
    **for** $(i, j)$ on anti-diagonal $k$ **do**
        $w \leftarrow \text{softmax}((\alpha_{i-1,j-1} + S_{ij}, \alpha_{i-1,j} + g, \alpha_{i,j-1} + g)/T)$
        $\dot{\alpha}_{ij} \leftarrow w_1(\dot{\alpha}_{i-1,j-1} + V_{ij}) + w_2\dot{\alpha}_{i-1,j} + w_3\dot{\alpha}_{i,j-1}$
    **end for**
**end for**
$\dot{H}_T \leftarrow \dot{\alpha}_{n,m}$
*// Backward tangent*
Initialize: $\dot{\beta}_{n,m} \leftarrow 0$
**for** $k = n + m$ **down to** $2$ **do**
    **for** $(i, j)$ on anti-diagonal $k$ **do**
        Compute $\dot{w}_k = w_k(\dot{v}_k - \dot{\alpha}_{ij})/T$
        $\text{HVP}_{ij} \leftarrow \dot{\beta}_{ij}w_1 + \beta_{ij}\dot{w}_1$ {transition-scored: $P_{ij} = \beta_{ij}w_1$}
        Propagate: $\dot{\beta}_{\text{pred}} \mathrel{+}= \dot{\beta}_{ij}w_k + \beta_{ij}\dot{w}_k$
    **end for**
**end for**
**return** HVP

---

**Algorithm 12** Linear Needleman–Wunsch: Gap Cross-Jacobian

---

**Input:** Forward $\alpha$, backward $\beta$, gap $g$, temperature $T$
**Output:** Cross-Jacobian $\partial P_T/\partial g$
*// Forward U pass*
Initialize: $U_{i,0} \leftarrow i$; $U_{0,j} \leftarrow j$
**for** $k = 2$ **to** $n + m$ **do**
    **for** $(i, j)$ on anti-diagonal $k$ **do**
        $w \leftarrow \text{softmax}((\alpha_{i-1,j-1} + S_{ij}, \alpha_{i-1,j} + g, \alpha_{i,j-1} + g)/T)$
        $U_{ij} \leftarrow w_1 U_{i-1,j-1} + w_2(U_{i-1,j} + 1) + w_3(U_{i,j-1} + 1)$
    **end for**
**end for**
$\partial V_T/\partial g \leftarrow U_{n,m}$
*// Backward W pass*
Initialize: $W_{n,m} \leftarrow 0$
**for** $k = n + m$ **down to** $2$ **do**
    **for** $(i, j)$ on anti-diagonal $k$ **do**
        Compute weight tangents from $U$
        $J_{ij} \leftarrow W_{ij}$
        Propagate: $W_{\text{pred}} \mathrel{+}= W_{ij}w_k + \beta_{ij}\dot{w}_k$
    **end for**
**end for**
**return** $J = \partial P_T/\partial g$

---

---

**Algorithm 13** Affine Needleman–Wunsch: Forward Pass

---

**Input:** Scores $S$, gap penalties $g_o, g_e$, temperature $T$
**Output:** Forward tables $\alpha^M, \alpha^I, \alpha^D$, partition $V_T$
Initialize: $\alpha^M_{0,0} \leftarrow 0$; $\alpha^I_{i,0} \leftarrow g_o + (i-1)g_e$; $\alpha^D_{0,j} \leftarrow g_o + (j-1)g_e$
All other $\alpha^\sigma_{ij} \leftarrow -\infty$
**for** $k = 2$ **to** $n + m$ **do**
    **for** $(i, j)$ on anti-diagonal $k$ **in parallel do**
        $\alpha^M_{ij} \leftarrow \mathrm{LSE}_T(\alpha^M_{i-1,j-1}, \alpha^I_{i-1,j-1}, \alpha^D_{i-1,j-1}) + S_{ij}$
        $\alpha^I_{ij} \leftarrow \mathrm{LSE}_T(\alpha^M_{i-1,j} + g_o, \alpha^I_{i-1,j} + g_e)$
        $\alpha^D_{ij} \leftarrow \mathrm{LSE}_T(\alpha^M_{i,j-1} + g_o, \alpha^D_{i,j-1} + g_e)$
    **end for**
**end for**
$V_T \leftarrow \mathrm{LSE}_T(\alpha^M_{n,m}, \alpha^I_{n,m}, \alpha^D_{n,m})$
**return** $\alpha^M, \alpha^I, \alpha^D, V_T$

---

---

**Algorithm 14** Affine Needleman–Wunsch: Backward Pass

---

**Input:** Forward tables $\alpha^M, \alpha^I, \alpha^D$, partition $V_T$, params
**Output:** Marginals $P$, gradients $\partial V_T / \partial g_o$, $\partial V_T / \partial g_e$
Initialize terminal: $w^M_t, w^I_t, w^D_t \leftarrow \mathrm{softmax}((\alpha^M_{n,m}, \alpha^I_{n,m}, \alpha^D_{n,m})/T)$
$\beta^M_{n,m} \leftarrow w^M_t$; $\beta^I_{n,m} \leftarrow w^I_t$; $\beta^D_{n,m} \leftarrow w^D_t$; all others $\leftarrow 0$
$G_o, G_e \leftarrow 0$
**for** $k = n + m$ **down to** $2$ **do**
    **for** $(i, j)$ on anti-diagonal $k$ **in parallel do**
        $w^M \leftarrow \mathrm{softmax}((\alpha^M_{i-1,j-1}, \alpha^I_{i-1,j-1}, \alpha^D_{i-1,j-1})/T)$
        $w^I \leftarrow \mathrm{softmax}((\alpha^M_{i-1,j} + g_o, \alpha^I_{i-1,j} + g_e)/T)$
        $w^D \leftarrow \mathrm{softmax}((\alpha^M_{i,j-1} + g_o, \alpha^D_{i,j-1} + g_e)/T)$
        $P_{T,ij} \leftarrow \beta^M_{ij}$
        Propagate $\beta$ through all state transitions
        $G_o \mathrel{+}= \beta^I_{ij} w^I_1 + \beta^D_{ij} w^D_1$
        $G_e \mathrel{+}= \beta^I_{ij} w^I_2 + \beta^D_{ij} w^D_2$
    **end for**
**end for**
Add boundary contributions to $G_e$
**return** $P, G_o, G_e$

---

**Algorithm 15** Affine Needleman–Wunsch: Score HVP

**Input:** Forward tables, partition, vector $V$, params
**Output:** HVP $\nabla_S^2 V_T \cdot V$
*// Forward tangent (3-state)*
**for** $k = 2$ **to** $n + m$ **do**
    **for** $(i, j)$ on anti-diagonal $k$ **do**
        $\dot{\alpha}_{ij}^M \leftarrow \sum_\sigma w^{\sigma \to M}(\dot{\alpha}_{i-1,j-1}^\sigma + V_{ij})$
        $\dot{\alpha}_{ij}^I \leftarrow w^{M \to I}\dot{\alpha}_{i-1,j}^M + w^{I \to I}\dot{\alpha}_{i-1,j}^I$
        $\dot{\alpha}_{ij}^D \leftarrow w^{M \to D}\dot{\alpha}_{i,j-1}^M + w^{D \to D}\dot{\alpha}_{i,j-1}^D$
    **end for**
**end for**
Aggregate: $\dot{H}_T \leftarrow \sum_\sigma w_t^\sigma \dot{\alpha}_{n,m}^\sigma$
*// Backward tangent*
Initialize with terminal tangents, propagate through all states
**return** HVP

**Algorithm 16** Affine Needleman–Wunsch: Gap-Open Cross-Jacobian

**Input:** Forward tables, backward $\beta$, params
**Output:** Cross-Jacobian $\partial P_T / \partial g_o$
*// Forward U pass*
Initialize: $U_{ij}^{g_o,\sigma} \leftarrow 0$; $U_{i,0}^{g_o,I} \leftarrow 1, U_{0,j}^{g_o,D} \leftarrow 1$ {each boundary gap opens once; for $g_e$, $U_{i,0}^{g_e,I} = i - 1, U_{0,j}^{g_e,D} = j - 1$}
**for** $k = 2$ **to** $n + m$ **do**
    **for** $(i, j)$ on anti-diagonal $k$ **do**
        $U_{ij}^{g_o,M} \leftarrow \sum_\sigma w^{\sigma \to M}U_{i-1,j-1}^{g_o,\sigma}$
        $U_{ij}^{g_o,I} \leftarrow w^{M \to I}(U_{i-1,j}^{g_o,M} + 1) + w^{I \to I}U_{i-1,j}^{g_o,I}$
        $U_{ij}^{g_o,D} \leftarrow w^{M \to D}(U_{i,j-1}^{g_o,M} + 1) + w^{D \to D}U_{i,j-1}^{g_o,D}$
    **end for**
**end for**
$\partial V_T / \partial g_o \leftarrow \sum_\sigma w_t^\sigma U_{n,m}^{g_o,\sigma}$
*// Backward W pass*
Initialize, propagate with weight tangents
**return** $J = \partial P_T / \partial g_o$

**Algorithm 17** LCS: Soft Forward Pass

**Input:** Match scores $S$ (where $S_{ij} = r$ if $x_i = y_j$, else 0), temperature $T$
**Output:** Forward table $\alpha$, partition $V_T$
Initialize: $\alpha_{i,0} \leftarrow 0$; $\alpha_{0,j} \leftarrow 0$ for all $i, j$
**for** $k = 2$ **to** $n + m$ **do**
    **for** $(i, j)$ on anti-diagonal $k$ **in parallel do**
        $\alpha_{ij} \leftarrow \text{LSE}_T(\alpha_{i-1,j-1} + S_{ij}, \alpha_{i-1,j}, \alpha_{i,j-1})$
    **end for**
**end for**
$V_T \leftarrow \alpha_{n,m}$
**return** $\alpha, V_T$

---

**Algorithm 18** LCS: Backward Pass

---

**Input:** Forward $\alpha$, partition $V_T$, scores $S$, temperature $T$
**Output:** Match marginals $P$
Initialize: $\beta_{n,m} \leftarrow 1$; all others $\leftarrow 0$
**for** $k = n + m$ **down to** 2 **do**
   **for** $(i, j)$ on anti-diagonal $k$ **in parallel do**
      $w \leftarrow \text{softmax}((\alpha_{i-1,j-1} + S_{ij}, \alpha_{i-1,j}, \alpha_{i,j-1})/T)$
      $P_{T,ij} \leftarrow \beta_{ij} \cdot w_1$ {Match probability}
      $\beta_{i-1,j-1} \mathrel{+}= \beta_{ij} w_1$
      $\beta_{i-1,j} \mathrel{+}= \beta_{ij} w_2$
      $\beta_{i,j-1} \mathrel{+}= \beta_{ij} w_3$
   **end for**
**end for**
**return** $P$

---

**Algorithm 19** LCS: Score HVP

---

**Input:** Forward $\alpha$, partition $V_T$, vector $V$, scores $S$, temperature $T$
**Output:** HVP $\nabla_S^2 V_T \cdot V$
*// Forward tangent*
Initialize: $\dot{\alpha}_{ij} \leftarrow 0$
**for** $k = 2$ **to** $n + m$ **do**
   **for** $(i, j)$ on anti-diagonal $k$ **do**
      $w \leftarrow \text{softmax}((\alpha_{i-1,j-1} + S_{ij}, \alpha_{i-1,j}, \alpha_{i,j-1})/T)$
      $\dot{\alpha}_{ij} \leftarrow w_1(\dot{\alpha}_{i-1,j-1} + V_{ij}) + w_2 \dot{\alpha}_{i-1,j} + w_3 \dot{\alpha}_{i,j-1}$
   **end for**
**end for**
$\dot{H}_T \leftarrow \dot{\alpha}_{n,m}$
*// Backward tangent*
Initialize: $\dot{\beta}_{n,m} \leftarrow 0$
**for** $k = n + m$ **down to** 2 **do**
   **for** $(i, j)$ on anti-diagonal $k$ **do**
      Compute $\dot{w}_k = w_k(\dot{v}_k - \dot{\alpha}_{ij})/T$
      $\text{HVP}_{ij} \leftarrow \dot{\beta}_{ij} \cdot w_1$
      Propagate: $\dot{\beta}_{\text{pred}} \mathrel{+}= \dot{\beta}_{ij} w_k + \beta_{ij} \dot{w}_k$
   **end for**
**end for**
**return** HVP

---

**Algorithm 20** MAS: Soft Forward Pass

---

**Input:** Scores $S \in \mathbb{R}^{T_{\max} \times S_{\max}}$, temperature $T$
**Output:** Forward table $\alpha$, partition $V_T$
Initialize: $\alpha_{0,0} \leftarrow S_{0,0}$; $\alpha_{0,s} \leftarrow -\infty$ for $s > 0$
$\alpha_{t,0} \leftarrow \alpha_{t-1,0} + S_{t,0}$ for $t > 0$
**for** $t = 1$ **to** $T_{\max}$ **do**
   **for** $s = 1$ **to** $S_{\max}$ **in parallel do**
      $\alpha_{t,s} \leftarrow S_{t,s} + \text{LSE}_T(\alpha_{t-1,s}, \alpha_{t-1,s-1})$
   **end for**
**end for**
$V_T \leftarrow \alpha_{T_{\max}, S_{\max}}$
**return** $\alpha, V_T$

---

---

**Algorithm 21** MAS: Backward Pass

---

**Input:** Forward $\alpha$, partition $V_T$, temperature $T$
**Output:** Alignment marginals $P$
Initialize: $\beta_{T_{\max}, S_{\max}} \leftarrow 1$; all others $\leftarrow 0$
**for** $t = T_{\max}$ **down to** 1 **do**
   **for** $s = S_{\max}$ **down to** 0 **in parallel do**
      $w \leftarrow \text{softmax}((\alpha_{t-1,s}, \alpha_{t-1,s-1})/T)$
      $P_{t,s} \leftarrow \beta_{t,s}$
      $\beta_{t-1,s} \mathrel{+}= \beta_{t,s} w_1$
      $\beta_{t-1,s-1} \mathrel{+}= \beta_{t,s} w_2$
   **end for**
**end for**
**return** $P$

---

---

**Algorithm 22** MAS: Score HVP

---

**Input:** Forward $\alpha$, partition $V_T$, vector $V$, temperature $T$
**Output:** HVP $\nabla_S^2 V_T \cdot V$
*// Forward tangent*
Initialize: $\dot{\alpha}_{t,s} \leftarrow 0$
**for** $t = 1$ **to** $T_{\max}$ **do**
   **for** $s = 1$ **to** $S_{\max}$ **do**
      $w \leftarrow \text{softmax}((\alpha_{t-1,s}, \alpha_{t-1,s-1})/T)$
      $\dot{\alpha}_{t,s} \leftarrow V_{t,s} + w_1 \dot{\alpha}_{t-1,s} + w_2 \dot{\alpha}_{t-1,s-1}$
   **end for**
**end for**
$\dot{H}_T \leftarrow \dot{\alpha}_{T_{\max}, S_{\max}}$
*// Backward tangent*
Initialize: $\dot{\beta}_{T_{\max}, S_{\max}} \leftarrow 0$
**for** $t = T_{\max}$ **down to** 1 **do**
   **for** $s = S_{\max}$ **down to** 0 **do**
      Compute $\dot{w}_k = w_k(\dot{\alpha}_{\text{pred}} - \dot{\alpha}_{t,s})/T$
      $\text{HVP}_{t,s} \leftarrow \dot{\beta}_{t,s}$
      Propagate: $\dot{\beta}_{\text{pred}} \mathrel{+}= \dot{\beta}_{t,s} w_k + \beta_{t,s} \dot{w}_k$
   **end for**
**end for**
**return** HVP

---

## D.3. Edit Distance Pseudocode

---
**Algorithm 23** DTW: Soft Forward Pass
---

**Input:** Cost matrix $C \in \mathbb{R}^{n \times m}$, step weights $\omega_d, \omega_h, \omega_v$, temperature $T$
**Output:** Forward table $\alpha$, distance $V_T$
Initialize: $\alpha_{0,0} \leftarrow C_{0,0}$
$\alpha_{i,0} \leftarrow \alpha_{i-1,0} + C_{i,0} + \omega_v$ for $i > 0$
$\alpha_{0,j} \leftarrow \alpha_{0,j-1} + C_{0,j} + \omega_h$ for $j > 0$
**for** $k = 2$ **to** $n + m$ **do**
  **for** $(i, j)$ on anti-diagonal $k$ **in parallel do**
    $\alpha_{ij} \leftarrow C_{ij} + \mathrm{smin}_T(\alpha_{i-1,j-1} + \omega_d, \alpha_{i-1,j} + \omega_v, \alpha_{i,j-1} + \omega_h)$
  **end for**
**end for**
$V_T \leftarrow \alpha_{n,m}$
**return** $\alpha, V_T$

---
**Algorithm 24** DTW: Backward Pass
---

**Input:** Forward $\alpha$, distance $V_T$, step weights $\omega_d, \omega_h, \omega_v$, temperature $T$
**Output:** Warping marginals $P$, step weight gradients $G_d, G_h, G_v$
Initialize: $\beta_{n,m} \leftarrow 1$; all others $\leftarrow 0$; $G_d, G_h, G_v \leftarrow 0$
**for** $k = n + m$ **down to** $2$ **do**
  **for** $(i, j)$ on anti-diagonal $k$ **in parallel do**
    $w \leftarrow \mathrm{softmin}((\alpha_{i-1,j-1} + \omega_d, \alpha_{i-1,j} + \omega_v, \alpha_{i,j-1} + \omega_h)/T)$
    $P_{T,ij} \leftarrow \beta_{ij}$
    $\beta_{i-1,j-1} \mathrel{+}= \beta_{ij} w_1; \quad G_d \mathrel{+}= \beta_{ij} w_1$
    $\beta_{i-1,j} \mathrel{+}= \beta_{ij} w_2; \quad G_v \mathrel{+}= \beta_{ij} w_2$
    $\beta_{i,j-1} \mathrel{+}= \beta_{ij} w_3; \quad G_h \mathrel{+}= \beta_{ij} w_3$
  **end for**
**end for**
**return** $P, G_d, G_h, G_v$

---

**Algorithm 25** DTW: Cost HVP

---

**Input:** Forward $\alpha$, distance $V_T$, step weights $\omega_d, \omega_h, \omega_v$, vector $V$, temperature $T$
**Output:** HVP $\nabla_C^2 V_T \cdot V$
*// Forward tangent*
Initialize: $\dot{\alpha}_{ij} \leftarrow 0$
**for** $k = 2$ **to** $n + m$ **do**
   **for** $(i, j)$ on anti-diagonal $k$ **do**
      $w \leftarrow \text{softmin}((\alpha_{i-1,j-1} + \omega_d, \alpha_{i-1,j} + \omega_v, \alpha_{i,j-1} + \omega_h)/T)$
      $\dot{\alpha}_{ij} \leftarrow V_{ij} + w_1 \dot{\alpha}_{i-1,j-1} + w_2 \dot{\alpha}_{i-1,j} + w_3 \dot{\alpha}_{i,j-1}$
   **end for**
**end for**
$\dot{D}_T \leftarrow \dot{\alpha}_{n,m}$
*// Backward tangent (softmin: negative weight tangents)*
Initialize: $\dot{\beta}_{n,m} \leftarrow 0$
**for** $k = n + m$ **down to** $2$ **do**
   **for** $(i, j)$ on anti-diagonal $k$ **do**
      Compute $\dot{w}_k = -w_k(\dot{\alpha}_{\text{pred}} - \dot{\alpha}_{ij})/T$
      $\text{HVP}_{ij} \leftarrow \dot{\beta}_{ij}$ {cell-scored: HVP is the tangent of the cell marginal}
      Propagate: $\dot{\beta}_{\text{pred}} \mathrel{+}= \dot{\beta}_{ij} w_k + \beta_{ij} \dot{w}_k$
   **end for**
**end for**
**return** HVP

---

**Algorithm 26** Levenshtein: Soft Forward Pass

---

**Input:** Substitution costs $C$, insertion $c_i$, deletion $c_d$, temperature $T$
**Output:** Forward table $\alpha$, distance $V_T$
Initialize: $\alpha_{0,0} \leftarrow 0$
$\alpha_{i,0} \leftarrow i \cdot c_d$ for $i > 0$
$\alpha_{0,j} \leftarrow j \cdot c_i$ for $j > 0$
**for** $k = 2$ **to** $n + m$ **do**
   **for** $(i, j)$ on anti-diagonal $k$ **in parallel do**
      $\alpha_{ij} \leftarrow \text{smin}_T(\alpha_{i-1,j-1} + C_{ij}, \alpha_{i-1,j} + c_d, \alpha_{i,j-1} + c_i)$
   **end for**
**end for**
$V_T \leftarrow \alpha_{n,m}$
**return** $\alpha, V_T$

---

---

**Algorithm 27** Levenshtein: Backward Pass

---

**Input:** Forward $\alpha$, costs $c_i, c_d$, temperature $T$
**Output:** Edit marginals, gradients $\partial V_T/\partial c_i$, $\partial V_T/\partial c_d$
Initialize: $\beta_{n,m} \leftarrow 1$; all others $\leftarrow 0$
$G_i, G_d \leftarrow 0$
**for** $k = n + m$ **down to** 2 **do**
   **for** $(i, j)$ on anti-diagonal $k$ **in parallel do**
      $w \leftarrow \text{softmin}((\alpha_{i-1,j-1} + C_{ij}, \alpha_{i-1,j} + c_d, \alpha_{i,j-1} + c_i)/T)$
      $P_{ij}^{\text{sub}} \leftarrow \beta_{ij} w_1$
      $\beta_{i-1,j-1} \mathrel{+}= \beta_{ij} w_1$
      $\beta_{i-1,j} \mathrel{+}= \beta_{ij} w_2$; $G_d \mathrel{+}= \beta_{ij} w_2$
      $\beta_{i,j-1} \mathrel{+}= \beta_{ij} w_3$; $G_i \mathrel{+}= \beta_{ij} w_3$
   **end for**
**end for**
Add boundary contributions to $G_i, G_d$
**return** $P^{\text{sub}}, G_i, G_d$

---

**Algorithm 28** Levenshtein: Substitution Cost HVP

---

**Input:** Forward $\alpha$, vector $V$, costs, temperature $T$
**Output:** HVP $\nabla_C^2 V_T \cdot V$
*// Forward tangent*
**for** $k = 2$ **to** $n + m$ **do**
   **for** $(i, j)$ on anti-diagonal $k$ **do**
      $w \leftarrow \text{softmin}((\alpha_{i-1,j-1} + C_{ij}, \alpha_{i-1,j} + c_d, \alpha_{i,j-1} + c_i)/T)$
      $\dot{\alpha}_{ij} \leftarrow w_1(\dot{\alpha}_{i-1,j-1} + V_{ij}) + w_2\dot{\alpha}_{i-1,j} + w_3\dot{\alpha}_{i,j-1}$
   **end for**
**end for**
$\dot{D}_T \leftarrow \dot{\alpha}_{n,m}$
*// Backward tangent*
**for** $k = n + m$ **down to** 2 **do**
   **for** $(i, j)$ on anti-diagonal $k$ **do**
      $\dot{w}_k \leftarrow -w_k(\dot{v}_k - \dot{\alpha}_{ij})/T$
      $\text{HVP}_{ij} \leftarrow \dot{\beta}_{ij} w_1 + \beta_{ij} \dot{w}_1$ {transition-scored; softmin sign carried by $\dot{w}_1$}
      Propagate: $\dot{\beta}_{\text{pred}} \mathrel{+}= \dot{\beta}_{ij} w_k + \beta_{ij} \dot{w}_k$
   **end for**
**end for**
**return** HVP

---

---

**Algorithm 29** Levenshtein: Insertion Cost Cross-Jacobian

---

**Input:** Forward $\alpha$, backward $\beta$, costs, temperature $T$
**Output:** Cross-Jacobian $\partial P_T / \partial c_i$
*// Forward U pass*
Initialize: $U_{0,j} \leftarrow j$; $U_{i,0} \leftarrow 0$
**for** $k = 2$ **to** $n + m$ **do**
    **for** $(i, j)$ on anti-diagonal $k$ **do**
        $w \leftarrow \mathrm{softmin}((\alpha_{i-1,j-1} + C_{ij}, \alpha_{i-1,j} + c_d, \alpha_{i,j-1} + c_i)/T)$
        $U_{ij} \leftarrow w_1 U_{i-1,j-1} + w_2 U_{i-1,j} + w_3(U_{i,j-1} + 1)$
    **end for**
**end for**
$\partial V_T / \partial c_i \leftarrow U_{n,m}$
*// Backward W pass*
Initialize: $W_{n,m} \leftarrow 0$
**for** $k = n + m$ **down to** $2$ **do**
    **for** $(i, j)$ on anti-diagonal $k$ **do**
        $\dot{w}_k \leftarrow -w_k(U_{\mathrm{pred}} + \mathbf{1}[k = 3] - U_{ij})/T$
        $J_{ij} \leftarrow W_{ij}$
        Propagate: $W_{\mathrm{pred}} \mathrel{+}= W_{ij} w_k + \beta_{ij} \dot{w}_k$
    **end for**
**end for**
**return** $J = \partial P_T / \partial c_i$

---

**Algorithm 30** OSA: Soft Forward Pass

---

**Input:** Sub costs $C$, trans mask $T_m$, costs $c_i, c_d, c_t$, temperature $T$
**Output:** Forward table $\alpha$, distance $V_T$
Initialize boundaries as Levenshtein
**for** $k = 2$ **to** $n + m$ **do**
    **for** $(i, j)$ on anti-diagonal $k$ **in parallel do**
        $v_1 \leftarrow \alpha_{i-1,j-1} + C_{ij}$
        $v_2 \leftarrow \alpha_{i-1,j} + c_d$
        $v_3 \leftarrow \alpha_{i,j-1} + c_i$
        $v_4 \leftarrow \alpha_{i-2,j-2} + c_t$ **if** $T_m[i, j] = 1$ **else** $+\infty$
        $\alpha_{ij} \leftarrow \mathrm{smin}_T(v_1, v_2, v_3, v_4)$
    **end for**
**end for**
$V_T \leftarrow \alpha_{n,m}$
**return** $\alpha, V_T$

---

---

**Algorithm 31** OSA: Backward Pass

---

**Input:** Forward $\alpha$, trans mask $T_m$, costs, temperature $T$
**Output:** Edit marginals, cost gradients
Initialize: $\beta_{n,m} \leftarrow 1$; all others $\leftarrow 0$
$G_i, G_d, G_t \leftarrow 0$
**for** $k = n + m$ **down to** 2 **do**
  **for** $(i, j)$ on anti-diagonal $k$ **in parallel do**
    Compute $v_1, v_2, v_3, v_4$ as in forward pass
    $w \leftarrow \text{softmin}((v_1, v_2, v_3, v_4)/T)$
    $\beta_{i-1,j-1} \mathrel{+}= \beta_{ij} w_1$
    $\beta_{i-1,j} \mathrel{+}= \beta_{ij} w_2$; $G_d \mathrel{+}= \beta_{ij} w_2$
    $\beta_{i,j-1} \mathrel{+}= \beta_{ij} w_3$; $G_i \mathrel{+}= \beta_{ij} w_3$
    **if** $T_m[i, j] = 1$ **then**
      $\beta_{i-2,j-2} \mathrel{+}= \beta_{ij} w_4$; $G_t \mathrel{+}= \beta_{ij} w_4$
    **end if**
  **end for**
**end for**
**return** $\beta, G_i, G_d, G_t$

---

**Algorithm 32** OSA: Substitution Cost HVP

---

**Input:** Forward $\alpha$, trans mask $T_m$, vector $V$, temperature $T$
**Output:** HVP $\nabla_C^2 V_T \cdot V$
*// Forward tangent*
**for** $k = 2$ **to** $n + m$ **do**
  **for** $(i, j)$ on anti-diagonal $k$ **do**
    $w \leftarrow$ softmin weights as in forward
    $\dot{\alpha}_{ij} \leftarrow w_1(\dot{\alpha}_{i-1,j-1} + V_{ij}) + w_2 \dot{\alpha}_{i-1,j} + w_3 \dot{\alpha}_{i,j-1}$
    **if** $T_m[i, j] = 1$ **then**
      $\dot{\alpha}_{ij} \mathrel{+}= w_4 \dot{\alpha}_{i-2,j-2}$
    **end if**
  **end for**
**end for**
*// Backward tangent*
**for** $k = n + m$ **down to** 2 **do**
  **for** $(i, j)$ on anti-diagonal $k$ **do**
    Compute weight tangents $\dot{w}_k = -w_k(\dot{v}_k - \dot{\alpha}_{ij})/T$
    $\text{HVP}_{ij} \leftarrow \dot{\beta}_{ij} w_1 + \beta_{ij} \dot{w}_1$ {transition-scored; softmin sign carried by $\dot{w}_1$}
    Propagate through all four edges
  **end for**
**end for**
**return** HVP

---

---

**Algorithm 33** OSA: Transposition Cost Cross-Jacobian

---

**Input:** Forward $\alpha$, backward $\beta$, trans mask $T_m$, temperature $T$
**Output:** Cross-Jacobian $\partial P_T / \partial c_t$
*// Forward U pass*
Initialize: $U_{ij} \leftarrow 0$
**for** $k = 2$ **to** $n + m$ **do**
    **for** $(i, j)$ on anti-diagonal $k$ **do**
        $w \leftarrow$ softmin weights
        $U_{ij} \leftarrow w_1 U_{i-1,j-1} + w_2 U_{i-1,j} + w_3 U_{i,j-1}$
        **if** $T_m[i, j] = 1$ **then**
            $U_{ij} \mathrel{+}= w_4(U_{i-2,j-2} + 1)$
        **end if**
    **end for**
**end for**
$\partial V_T / \partial c_t \leftarrow U_{n,m}$
*// Backward W pass*
**for** $k = n + m$ **down to** $2$ **do**
    **for** $(i, j)$ on anti-diagonal $k$ **do**
        Compute weight tangents from $U$
        $J_{ij} \leftarrow W_{ij}$
        Propagate through all edges
    **end for**
**end for**
**return** $J = \partial P_T / \partial c_t$

---

**Algorithm 34** Damerau–Levenshtein: Soft Forward Pass

---

**Input:** Strings $x, y$, costs $c_i, c_d, c_t$, substitution costs $C$, temperature $T$
**Output:** Forward table $\alpha$, distance $V_T$
Initialize: $\alpha_{0,0} \leftarrow 0$; $\alpha_{i,0} \leftarrow i \cdot c_d$; $\alpha_{0,j} \leftarrow j \cdot c_i$
Initialize: $DA[c] \leftarrow 0$ for all characters $c$
**for** $i = 1$ **to** $n$ **do**
    $DB \leftarrow 0$
    **for** $j = 1$ **to** $m$ **do**
        $k \leftarrow DA[y_j]$; $l \leftarrow DB$
        $v_1 \leftarrow \alpha_{i-1,j-1} + C_{ij}$
        $v_2 \leftarrow \alpha_{i-1,j} + c_d$
        $v_3 \leftarrow \alpha_{i,j-1} + c_i$
        **if** $k > 0$ **and** $l > 0$ **then**
            $v_4 \leftarrow \alpha_{k-1,l-1} + (i - k - 1)c_d + c_t + (j - l - 1)c_i$
        **else**
            $v_4 \leftarrow +\infty$
        **end if**
        $\alpha_{ij} \leftarrow \mathrm{smin}_T(v_1, v_2, v_3, v_4)$
        **if** $x_i = y_j$ **then**
            $DB \leftarrow j$
        **end if**
    **end for**
    $DA[x_i] \leftarrow i$
**end for**
$V_T \leftarrow \alpha_{n,m}$
**return** $\alpha, V_T$

---

**Algorithm 35** Damerau–Levenshtein: Backward Pass

**Input:** Forward $\alpha$, precomputed $(k_{ij}, l_{ij})$ pairs, temperature $T$
**Output:** Edit marginals $\beta$, cost gradients
Initialize: $\beta_{n,m} \leftarrow 1$; all others $\leftarrow 0$
$G_i, G_d, G_t \leftarrow 0$
**for** $i = n$ **down to** 1 **do**
  **for** $j = m$ **down to** 1 **do**
    Retrieve $k, l$ for position $(i, j)$
    Compute softmin weights $w$ from forward values
    $\beta_{i-1,j-1} \mathrel{+}= \beta_{ij} w_1$
    $\beta_{i-1,j} \mathrel{+}= \beta_{ij} w_2$; $G_d \mathrel{+}= \beta_{ij} w_2$
    $\beta_{i,j-1} \mathrel{+}= \beta_{ij} w_3$; $G_i \mathrel{+}= \beta_{ij} w_3$
    **if** $k > 0$ **and** $l > 0$ **then**
      $\beta_{k-1,l-1} \mathrel{+}= \beta_{ij} w_4$
      $G_t \mathrel{+}= \beta_{ij} w_4$
      $G_d \mathrel{+}= \beta_{ij} w_4 (i - k - 1)$
      $G_i \mathrel{+}= \beta_{ij} w_4 (j - l - 1)$
    **end if**
  **end for**
**end for**
**return** $\beta, G_i, G_d, G_t$

---

**Algorithm 36** Damerau–Levenshtein: Substitution Cost HVP

**Input:** Forward $\alpha$, $(k_{ij}, l_{ij})$ pairs, vector $V$, temperature $T$
**Output:** HVP $\nabla_C^2 V_T \cdot V$
*// Forward tangent*
Initialize: $\dot{\alpha}_{ij} \leftarrow 0$
**for** $i = 1$ **to** $n$ **do**
  **for** $j = 1$ **to** $m$ **do**
    Retrieve $k, l$ for position $(i, j)$
    Compute softmin weights $w$
    $\dot{\alpha}_{ij} \leftarrow w_1(\dot{\alpha}_{i-1,j-1} + V_{ij}) + w_2 \dot{\alpha}_{i-1,j} + w_3 \dot{\alpha}_{i,j-1}$
    **if** $k > 0$ **and** $l > 0$ **then**
      $\dot{\alpha}_{ij} \mathrel{+}= w_4 \dot{\alpha}_{k-1,l-1}$
    **end if**
  **end for**
**end for**
$\dot{D}_T \leftarrow \dot{\alpha}_{n,m}$
*// Backward tangent*
**for** $i = n$ **down to** 1 **do**
  **for** $j = m$ **down to** 1 **do**
    Compute weight tangents $\dot{w}_k = -w_k(\dot{v}_k - \dot{\alpha}_{ij})/T$
    $\text{HVP}_{ij} \leftarrow \dot{\beta}_{ij} w_1 + \beta_{ij} \dot{w}_1$ {transition-scored; softmin sign carried by $\dot{w}_1$}
    Propagate through all edges including transposition
  **end for**
**end for**
**return** HVP

---

## D.4. Parsing Pseudocode

---

**Algorithm 37** CKY: Soft Forward (Inside) Pass

---

**Input:** Leaf scores $L \in \mathbb{R}^n$, merge scores $M \in \mathbb{R}^{n \times n \times n}$, span penalty $\lambda$, temperature $T$
**Output:** Chart $Z \in \mathbb{R}^{n \times n}$, partition $V_T$
Initialize: $Z[i,j] \leftarrow -\infty$ for all $i,j$
$Z[i,i] \leftarrow L_i$ for $i = 0, \ldots, n-1$
**for** $\ell = 2$ **to** $n$ **do**
   **for** $i = 0$ **to** $n - \ell$ **in parallel do**
      $j \leftarrow i + \ell - 1$
      $Z[i,j] \leftarrow \text{LSE}_T(\{Z[i,k] + Z[k+1,j] + M[i,k,j] + \lambda \cdot (j-i)\}_{k=i}^{j-1})$
   **end for**
**end for**
$V_T \leftarrow Z[0, n-1]$
**return** $Z, V_T$

---

**Algorithm 38** CKY: Backward (Outside) Pass

---

**Input:** Chart $Z$, merge scores $M$, span penalty $\lambda$, temperature $T$
**Output:** Span marginals $\beta$, split marginals $P_{\text{split}}$, span penalty gradient $G_\lambda$
Initialize: $\beta[i,j] \leftarrow 0$ for all $i,j$; $G_\lambda \leftarrow 0$
$\beta[0, n-1] \leftarrow 1$
**for** $\ell = n$ **down to** $2$ **do**
   **for** $i = 0$ **to** $n - \ell$ **in parallel do**
      $j \leftarrow i + \ell - 1$
      $w \leftarrow \text{softmax}(\{Z[i,k] + Z[k+1,j] + M[i,k,j] + \lambda \cdot (j-i)\}_{k=i}^{j-1}/T)$
      $G_\lambda \mathrel{+}= \beta[i,j] \cdot (j-i)$ {Span length contribution}
      **for** $k = i$ **to** $j-1$ **do**
         $P_{\text{split}}[i,k,j] \leftarrow \beta[i,j] \cdot w_{k-i}$
         $\beta[i,k] \mathrel{+}= \beta[i,j] \cdot w_{k-i}$
         $\beta[k+1,j] \mathrel{+}= \beta[i,j] \cdot w_{k-i}$
      **end for**
   **end for**
**end for**
**return** $\beta, P_{\text{split}}, G_\lambda$

---

**Algorithm 39** CKY: Merge Score HVP

**Input:** Chart $Z$, span marginals $\beta$, split marginals $P_{\text{split}}$, vector $V$, merge scores $M$, span penalty $\lambda$, temperature $T$
**Output:** HVP $\nabla_M^2 V_T \cdot V$
*// Forward tangent pass*
Initialize: $\dot{Z}[i,i] \leftarrow 0$ for all $i$
**for** $\ell = 2$ **to** $n$ **do**
  **for** $i = 0$ **to** $n - \ell$ **in parallel do**
    $j \leftarrow i + \ell - 1$
    $w \leftarrow \text{softmax}(\{Z[i,k] + Z[k+1,j] + M[i,k,j] + \lambda \cdot (j-i)\}_{k=i}^{j-1}/T)$
    $\dot{Z}[i,j] \leftarrow \sum_{k=i}^{j-1} w_{k-i} \cdot (\dot{Z}[i,k] + \dot{Z}[k+1,j] + V[i,k,j])$
  **end for**
**end for**
$\dot{H}_T \leftarrow \dot{Z}[0, n-1]$
*// Backward tangent pass*
Initialize: $\dot{\beta}[0, n-1] \leftarrow 0$; all others $\leftarrow 0$
**for** $\ell = n$ **down to** $2$ **do**
  **for** $i = 0$ **to** $n - \ell$ **in parallel do**
    $j \leftarrow i + \ell - 1$
    $w \leftarrow \text{softmax}(\{Z[i,k] + Z[k+1,j] + M[i,k,j] + \lambda \cdot (j-i)\}_{k=i}^{j-1}/T)$
    $\bar{w} \leftarrow \dot{Z}[i,k] + \dot{Z}[k+1,j] + V[i,k,j] - \dot{Z}[i,j]$ *// residual*
    **for** $k = i$ **to** $j - 1$ **do**
      $\text{HVP}[i,k,j] \leftarrow \beta[i,j] \cdot w_{k-i} \cdot \bar{w}_{k-i}/T$
      $\dot{\beta}[i,k] \mathrel{+}= \dot{\beta}[i,j] \cdot w_{k-i} + \beta[i,j] \cdot w_{k-i} \cdot \bar{w}_{k-i}/T$
      $\dot{\beta}[k+1,j] \mathrel{+}= \dot{\beta}[i,j] \cdot w_{k-i} + \beta[i,j] \cdot w_{k-i} \cdot \bar{w}_{k-i}/T$
    **end for**
  **end for**
**end for**
**return** HVP

---

**Algorithm 40** Eisner: Soft Forward Pass

**Input:** Arc scores $A \in \mathbb{R}^{n \times n}$, arc length penalty $\lambda$, direction bias $\delta$, temperature $T$
**Output:** Charts $C_R, C_L, I_R, I_L \in \mathbb{R}^{n \times n}$, partition $V_T$
Initialize: $C_R[i,i], C_L[i,i] \leftarrow 0$; all others $\leftarrow -\infty$
**for** $\ell = 1$ **to** $n - 1$ **do**
  **for** $i = 0$ **to** $n - \ell - 1$ **in parallel do**
    $j \leftarrow i + \ell$
    *// Incomplete spans (with arc length penalty and direction bias)*
    $\text{lse} \leftarrow \text{LSE}_T(\{C_R[i,k] + C_L[k+1,j]\}_{k=i}^{j})$
    $I_R[i,j] \leftarrow A[i,j] + \lambda \cdot |j-i| + \delta + \text{lse}$ {Rightward arc}
    $I_L[i,j] \leftarrow A[j,i] + \lambda \cdot |j-i| + \text{lse}$ {Leftward arc}
    *// Complete spans*
    $C_R[i,j] \leftarrow \text{LSE}_T(\{C_R[i,k] + I_R[k,j]\}_{k=i}^{j-1})$
    $C_L[i,j] \leftarrow \text{LSE}_T(\{I_L[i,k] + C_L[k,j]\}_{k=i+1}^{j})$
  **end for**
**end for**
$V_T \leftarrow C_R[0, n-1]$
**return** $C_R, C_L, I_R, I_L, V_T$

---

**Algorithm 41** Eisner: Backward (Outside) Pass

---

**Input:** Charts $C_R, C_L, I_R, I_L$, arc scores $A$, arc length penalty $\lambda$, direction bias $\delta$, temperature $T$
**Output:** Arc marginals $\beta_{I_R}, \beta_{I_L}$, gradients $G_\lambda, G_\delta$
Initialize: $\beta_{C_R}[0, n-1] \leftarrow 1$; all others $\leftarrow 0$; $G_\lambda, G_\delta \leftarrow 0$
**for** $\ell = n-1$ **down to** $1$ **do**
  **for** $i = 0$ **to** $n - \ell - 1$ **in parallel do**
    $j \leftarrow i + \ell$
    *// Complete span weights*
    $w^{C_R} \leftarrow \text{softmax}(\{C_R[i,k] + I_R[k,j]\}_{k=i}^{j-1}/T)$
    $w^{C_L} \leftarrow \text{softmax}(\{I_L[i,k] + C_L[k,j]\}_{k=i+1}^{j}/T)$
    *// Propagate from complete to incomplete*
    **for** $k = i$ **to** $j - 1$ **do**
      $\beta_{I_R}[k,j] \mathrel{+}= \beta_{C_R}[i,j] \cdot w_{k-i}^{C_R}$
      $\beta_{C_R}[i,k] \mathrel{+}= \beta_{C_R}[i,j] \cdot w_{k-i}^{C_R}$
    **end for**
    **for** $k = i+1$ **to** $j$ **do**
      $\beta_{I_L}[i,k] \mathrel{+}= \beta_{C_L}[i,j] \cdot w_{k-i-1}^{C_L}$
      $\beta_{C_L}[k,j] \mathrel{+}= \beta_{C_L}[i,j] \cdot w_{k-i-1}^{C_L}$
    **end for**
    *// Incomplete span weights*
    $w^I \leftarrow \text{softmax}(\{C_R[i,k] + C_L[k+1,j]\}_{k=i}^{j-1}/T)$
    *// Propagate from incomplete to complete*
    **for** $k = i$ **to** $j - 1$ **do**
      $\beta_{C_R}[i,k] \mathrel{+}= (\beta_{I_R}[i,j] + \beta_{I_L}[i,j]) \cdot w_{k-i}^I$
      $\beta_{C_L}[k+1,j] \mathrel{+}= (\beta_{I_R}[i,j] + \beta_{I_L}[i,j]) \cdot w_{k-i}^I$
    **end for**
    *// Arc length penalty gradient (both directions)*
    $G_\lambda \mathrel{+}= (\beta_{I_R}[i,j] + \beta_{I_L}[i,j]) \cdot |j - i|$
    *// Direction bias gradient (rightward arcs only)*
    $G_\delta \mathrel{+}= \beta_{I_R}[i,j]$
  **end for**
**end for**
*// Arc marginals*
$P(\text{arc } i \to j) \leftarrow \beta_{I_R}[i,j]$
$P(\text{arc } j \to i) \leftarrow \beta_{I_L}[i,j]$
**return** $\beta_{I_R}, \beta_{I_L}, G_\lambda, G_\delta$

---

---

**Algorithm 42** Eisner: Arc Score HVP

---

**Input:** Charts $C_R, C_L, I_R, I_L$, arc marginals $\beta_{I_R}, \beta_{I_L}$, vector $V$, arc scores $A$, arc length penalty $\lambda$, direction bias $\delta$, temperature $T$

**Output:** HVP $\nabla_A^2 V_T \cdot V$

*// Forward tangent pass*

Initialize: $\dot{C}_R[i,i], \dot{C}_L[i,i] \leftarrow 0$; all others $\leftarrow 0$

**for** $\ell = 1$ **to** $n - 1$ **do**
  **for** $i = 0$ **to** $n - \ell - 1$ **in parallel do**
    $j \leftarrow i + \ell$
    *// Incomplete span tangents*
    $w^I \leftarrow \text{softmax}(\{C_R[i,k] + C_L[k+1,j]\}_{k=i}^{j-1}/T)$
    $\dot{\text{lse}} \leftarrow \sum_{k=i}^{j-1} w_{k-i}^I \cdot (\dot{C}_R[i,k] + \dot{C}_L[k+1,j])$
    $\dot{I}_R[i,j] \leftarrow V[i,j] + \dot{\text{lse}}$ {Arc score tangent for rightward arc}
    $\dot{I}_L[i,j] \leftarrow V[j,i] + \dot{\text{lse}}$ {Arc score tangent for leftward arc}
    *// Complete span tangents*
    $w^{C_R} \leftarrow \text{softmax}(\{C_R[i,k] + I_R[k,j]\}_{k=i}^{j-1}/T)$
    $\dot{C}_R[i,j] \leftarrow \sum_{k=i}^{j-1} w_{k-i}^{C_R} \cdot (\dot{C}_R[i,k] + \dot{I}_R[k,j])$
    $w^{C_L} \leftarrow \text{softmax}(\{I_L[i,k] + C_L[k,j]\}_{k=i+1}^{j}/T)$
    $\dot{C}_L[i,j] \leftarrow \sum_{k=i+1}^{j} w_{k-i-1}^{C_L} \cdot (\dot{I}_L[i,k] + \dot{C}_L[k,j])$
  **end for**
**end for**

$\dot{H}_T \leftarrow \dot{C}_R[0, n-1]$

*// Backward tangent pass*

Initialize: $\dot{\beta}_{C_R}[0, n-1] \leftarrow 0$; all others $\leftarrow 0$

**for** $\ell = n - 1$ **down to** $1$ **do**
  **for** $i = 0$ **to** $n - \ell - 1$ **in parallel do**
    $j \leftarrow i + \ell$
    *// Complete span HVP contributions*
    $w^{C_R} \leftarrow \text{softmax}(\{C_R[i,k] + I_R[k,j]\}_{k=i}^{j-1}/T)$
    $\bar{w}_k^{C_R} \leftarrow \dot{C}_R[i,k] + \dot{I}_R[k,j] - \dot{C}_R[i,j]$ *// residuals*
    **for** $k = i$ **to** $j - 1$ **do**
      $\text{HVP}[k,j] \mathrel{+}= \beta_{C_R}[i,j] \cdot w_{k-i}^{C_R} \cdot \bar{w}_{k-i}^{C_R}/T$
    **end for**
    $w^{C_L} \leftarrow \text{softmax}(\{I_L[i,k] + C_L[k,j]\}_{k=i+1}^{j}/T)$
    $\bar{w}_k^{C_L} \leftarrow \dot{I}_L[i,k] + \dot{C}_L[k,j] - \dot{C}_L[i,j]$
    **for** $k = i + 1$ **to** $j$ **do**
      $\text{HVP}[k,i] \mathrel{+}= \beta_{C_L}[i,j] \cdot w_{k-i-1}^{C_L} \cdot \bar{w}_{k-i-1}^{C_L}/T$
    **end for**
    *// Incomplete span HVP contributions*
    $w^I \leftarrow \text{softmax}(\{C_R[i,k] + C_L[k+1,j]\}_{k=i}^{j-1}/T)$
    $\bar{w}_k^I \leftarrow \dot{C}_R[i,k] + \dot{C}_L[k+1,j] - \dot{\text{lse}}$
    **for** $k = i$ **to** $j - 1$ **do**
      $\text{HVP}[i,j] \mathrel{+}= \beta_{I_R}[i,j] \cdot w_{k-i}^I \cdot \bar{w}_{k-i}^I/T$
      $\text{HVP}[j,i] \mathrel{+}= \beta_{I_L}[i,j] \cdot w_{k-i}^I \cdot \bar{w}_{k-i}^I/T$
    **end for**
  **end for**
**end for**
**return** HVP

---

# E. Lipschitz Bounds and Convexity

This appendix provides the full theorem statements for Lipschitz bounds and convexity properties of the soft DP value function.

## E.1. Lipschitz Continuity

**Theorem E.1** (Value Lipschitz). *For all $S, S'$, $|V_T(S) - V_T(S')| \leq L_{\max}\|S - S'\|_\infty$, where $L_{\max}$ is the maximum path length in the DAG.*

**Theorem E.2** (Score Gradient Lipschitz). *$\nabla_S V_T$ is $L_{\max}/T$-Lipschitz in $\ell_2$ norm: $\|\nabla V_T(S) - \nabla V_T(S')\|_2 \leq \frac{L_{\max}}{T}\|S - S'\|_2$.*

**Theorem E.3** (Parameter Gradient Lipschitz). *For any algorithm parameter $\eta$ (gap penalty, insertion cost, etc.), $\partial V_T/\partial\eta$ is Lipschitz in the scores: $\left|\frac{\partial V_T}{\partial\eta}(S) - \frac{\partial V_T}{\partial\eta}(S')\right| \leq \frac{\sqrt{|E|}\,C_\eta}{T}\|S - S'\|_2$, where $C_\eta$ bounds the $\eta$-count over all paths and $|E|$ is the number of scored transitions.*

**Theorem E.4** (Marginal Lipschitz). *The marginals $P = \nabla_S V_T$ are Lipschitz in algorithm parameters: $\|P(\eta) - P(\eta')\|_2 \leq \frac{\sqrt{|E|}\,C_\eta}{T}|\eta - \eta'|$.*

## E.2. Convexity and Hessian Structure

**Theorem E.5** (Score Convexity). *For maximization problems, $V_T(S)$ is convex in the scores $S$:*

$$\nabla_{SS}^2 V_T = \frac{1}{T}\mathrm{Cov}_{p_T}(Y, Y) \succeq 0.$$

*For minimization problems, $V_T(c)$ is concave in the costs $c$:*

$$\nabla_{cc}^2 V_T = -\frac{1}{T}\mathrm{Cov}_{q_T}(Y, Y) \preceq 0.$$

**Theorem E.6** (Parameter Convexity). *The soft value is convex (maximization) or concave (minimization) in algorithm parameters $\eta$ (gap penalties, insertion costs, etc.):*

$$\frac{\partial^2 V_T}{\partial\eta^2} = \frac{1}{T}\mathrm{Var}_{p_T}[\varphi_\eta(Y)] \geq 0 \quad \text{(max)}, \qquad \frac{\partial^2 V_T}{\partial\eta^2} = -\frac{1}{T}\mathrm{Var}_{q_T}[\varphi_\eta(Y)] \leq 0 \quad \text{(min)},$$

*where $\varphi_\eta(Y)$ is the count of $\eta$-weighted transitions in structure $Y$.*

**Theorem E.7** (Mixed Hessian Structure). *The mixed score–parameter Hessian (cross-Jacobian) is generally indefinite:*

$$\frac{\partial^2 V_T}{\partial S_e \partial\eta} = \frac{1}{T}\mathrm{Cov}_{p_T}(\mathbf{1}\{e \in Y\}, \varphi_\eta(Y)),$$

*which can be positive, negative, or zero depending on whether edge $e$ and parameter $\eta$ are positively correlated, negatively correlated, or independent under $p_T$.*

## E.3. Approximation Bounds

**Theorem E.8** (Approximation Bound). *The soft DP value approximates the hard DP value with bounded error:*

$$V^*(S) \leq V_T(S) \leq V^*(S) + T \cdot H_{\max},$$

*where $V^* = \max_Y \mathrm{score}(Y)$ is the hard maximum and $H_{\max} = L_{\max}\log D_{\max}$ with $L_{\max}$ the maximum path length and $D_{\max}$ the maximum in-degree.*

*For minimization: $D^\star - T \cdot H_{\max} \leq V_T \leq D^\star$.*

This bound, following [Mensch & Blondel (2018)](#), shows that $V_T \to V^*$ as $T \to 0$ at rate $O(T)$.

*Remark* E.9 (Optimization Implications). Score convexity describes the soft DP value as a function of its direct scores; it gives smoothness and curvature bounds that help explain the stability of gradient-based optimization, but it does not by itself imply global convexity of a downstream loss composed with the marginals, or of a neural network parameterization of the scores. The indefinite cross-Jacobian means that joint optimization over scores and parameters may have saddle points.

# F. Zero-Temperature Limit

As temperature approaches zero, the soft DP value converges to the hard optimum, marginals concentrate on the optimal structure, and, when the optimum is unique, all covariances vanish. When optima are tied, $p_T$ converges to the distribution supported on the optimal set, so marginals approach averages over the tied optima and covariances may remain nonzero within that set.

**Theorem F.1** (Zero-Temperature Convergence). *Assume the optimal structure $Y^\star$ is unique and separated from the runner-up by a positive score gap. Then, as $T \to 0^+$:*

1. *$V_T \to V^* = \max_Y \operatorname{score}(Y)$ (or $\min$ for costs).*

2. *Marginals $\partial V_T / \partial S_a \to \mathbf{1}\{a \in Y^\star\}$.*

3. *All covariances $\operatorname{Cov}_{p_T}(\cdot, \cdot) \to 0$, so cross-Jacobians vanish.*

*Proof.* Let $Y^\star = \arg\max_Y \operatorname{score}(Y)$ be the optimal path (assume unique for simplicity).

**Part 1: Value convergence.** As $T \to 0^+$, $\operatorname{LSE}_T(v) \to \max(v)$ pointwise. By continuity of composition, $V_T(S) \to V^*(S) = \max_Y \operatorname{score}(Y)$.

**Part 2: Marginal convergence.** The Gibbs distribution concentrates on $Y^\star$:

$$p_T(Y) = \frac{\exp(\operatorname{score}(Y)/T)}{\sum_{Y'} \exp(\operatorname{score}(Y')/T)} \to \mathbf{1}[Y = Y^\star]$$

as $T \to 0^+$. Therefore marginals converge: $P_{T,e} = \mathbb{E}_{p_T}[\mathbf{1}\{e \in Y\}] \to \mathbf{1}\{e \in Y^\star\}$.

**Part 3: Covariance vanishing.** As $p_T \to \delta_{Y^\star}$ (point mass), all covariances vanish: $\operatorname{Cov}_{p_T}(f, g) = \mathbb{E}(fg) - \mathbb{E}(f)\mathbb{E}(g) \to f(Y^\star)g(Y^\star) - f(Y^\star)g(Y^\star) = 0$. The cross-Jacobians carry a $1/T$ prefactor, so pointwise vanishing is not enough; we need the rate. Since $Y^\star$ is separated from every competing structure by a score gap $\Delta > 0$, the total Gibbs mass on non-optimal structures is $O(e^{-\Delta/T})$, and every covariance of bounded statistics is therefore $O(e^{-\Delta/T})$. Hence $\frac{1}{T}\operatorname{Cov} = O(T^{-1}e^{-\Delta/T}) \to 0$, and all cross-Jacobians $\partial P_T / \partial S = \frac{1}{T}\operatorname{Cov} \to 0$. $\square$

At finite $T$, marginals provide "soft" structures distributing probability across suboptimal solutions. Cross-Jacobians quantify sensitivity to parameters in this soft regime, enabling gradient-based learning of gap penalties, insertion costs, and other algorithm parameters.

# G. Supplemental Figures

This appendix provides additional visualizations of parsing marginals, training dynamics, learned substitution matrices, and runtime speedups.

## G.1. Parsing Marginals

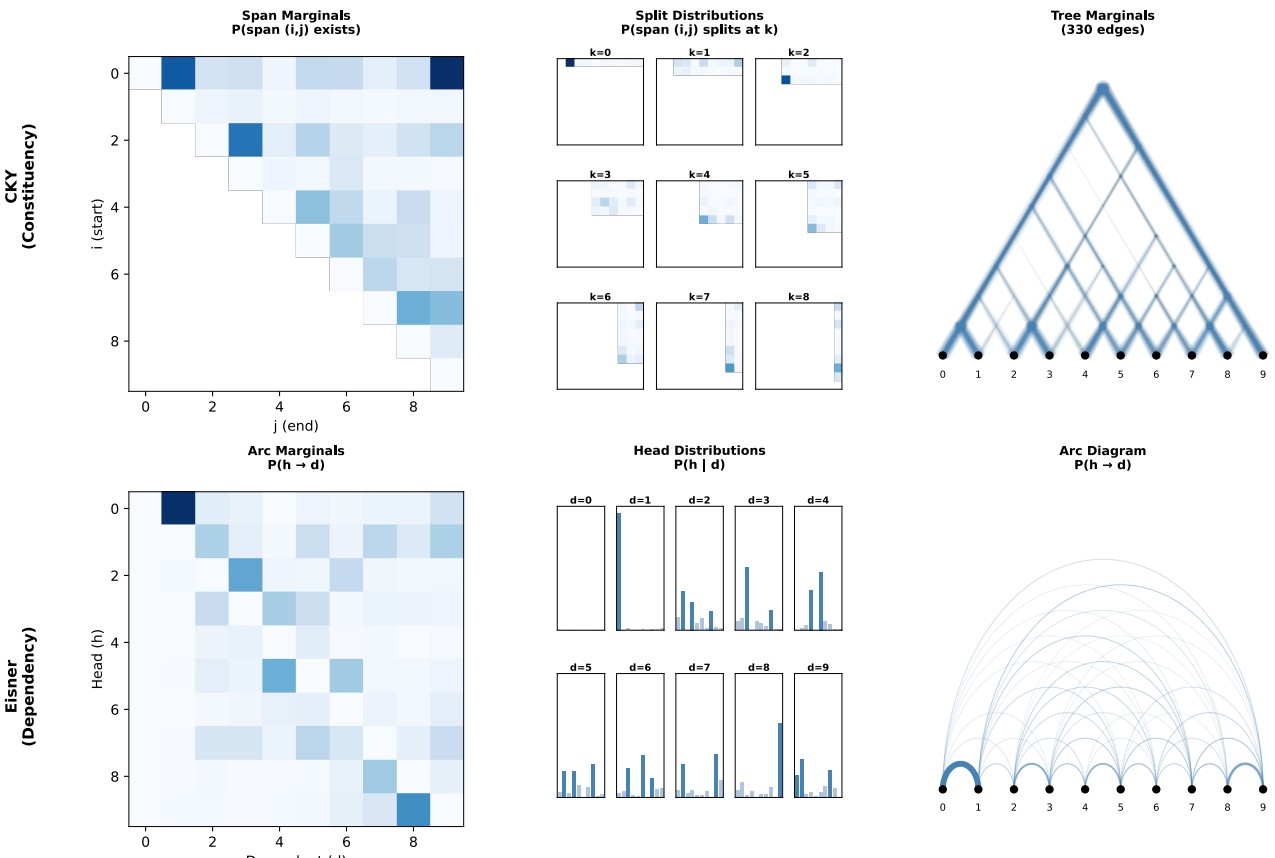

*Figure 4.* **Parsing marginals for CKY (top) and Eisner (bottom).** *Top row:* CKY constituency parsing: span marginals $P_{T,ij}$ (left), split distributions $P(k \mid i, j)$ for $k = 0$–8 (center), and overlaid parse trees weighted by probability (right, 4862 trees enumerated). *Bottom row:* Eisner dependency parsing: arc marginals $P(h \rightarrow d)$ (left), head distributions $P(h \mid d)$ for each dependent $d = 0$–9 (center), and arc diagram with edge thickness proportional to probability (right). Dark cells indicate high probability. All at $T = 1.0$.

## G.2. Training Dynamics

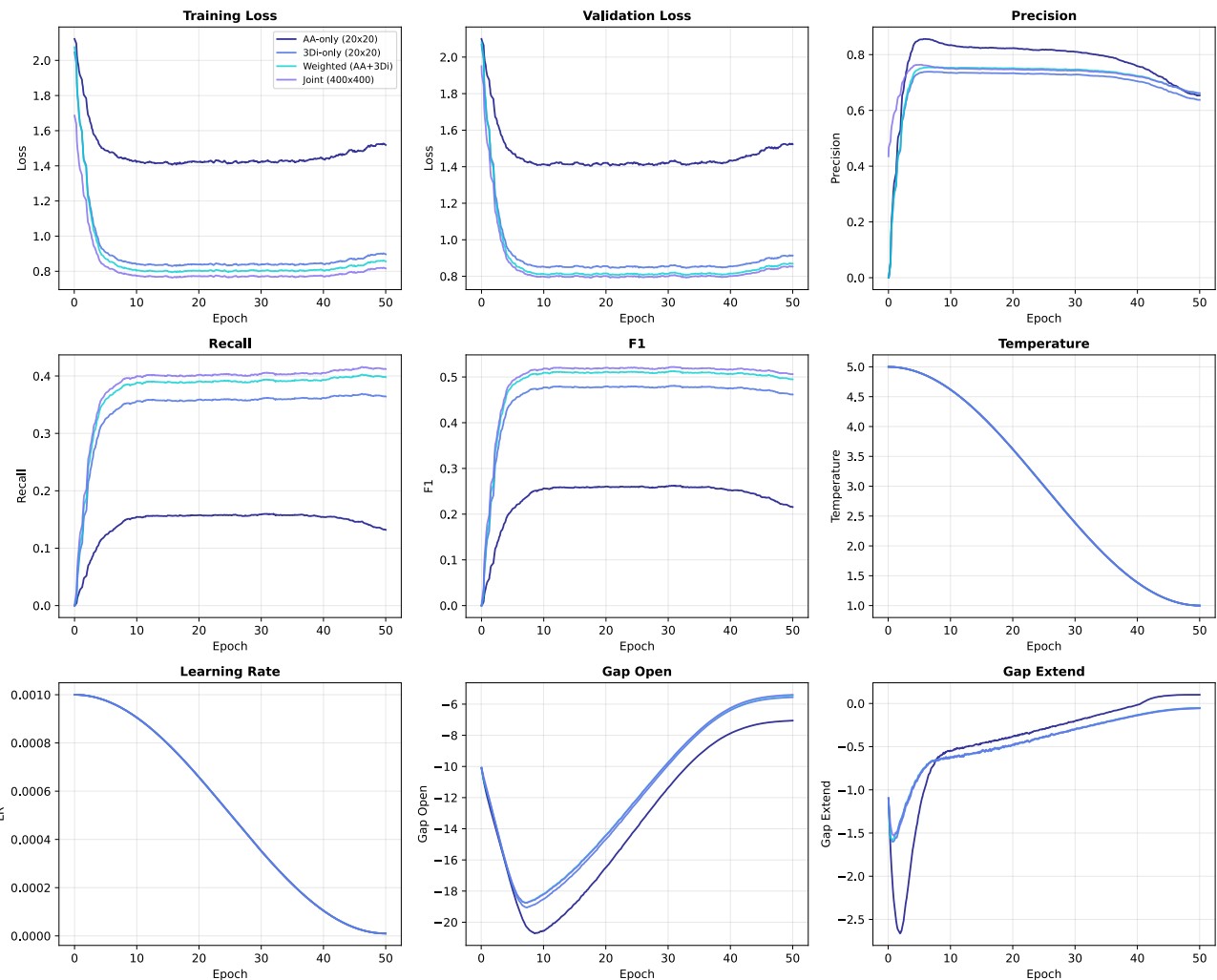

*Figure 5.* **Training dynamics for substitution matrix models.** $3\times3$ grid showing metrics over 50 epochs for four models: AA-only ($20\times20$), 3Di-only ($20\times20$), Weighted (AA+3Di), and Joint ($400\times400$). *Row 1:* Training loss, validation loss, and precision. *Row 2:* Recall, F1 score, and temperature schedule (annealed from $T = 5.0$ to $T = 1.0$). *Row 3:* Learning rate (cosine decay), gap open penalty, and gap extend penalty. 3Di-based models achieve substantially higher F1 (0.46–0.51) than AA-only (0.23). All models converge to similar gap open values ($\approx -5.5$), while AA-only learns a positive gap extend penalty.

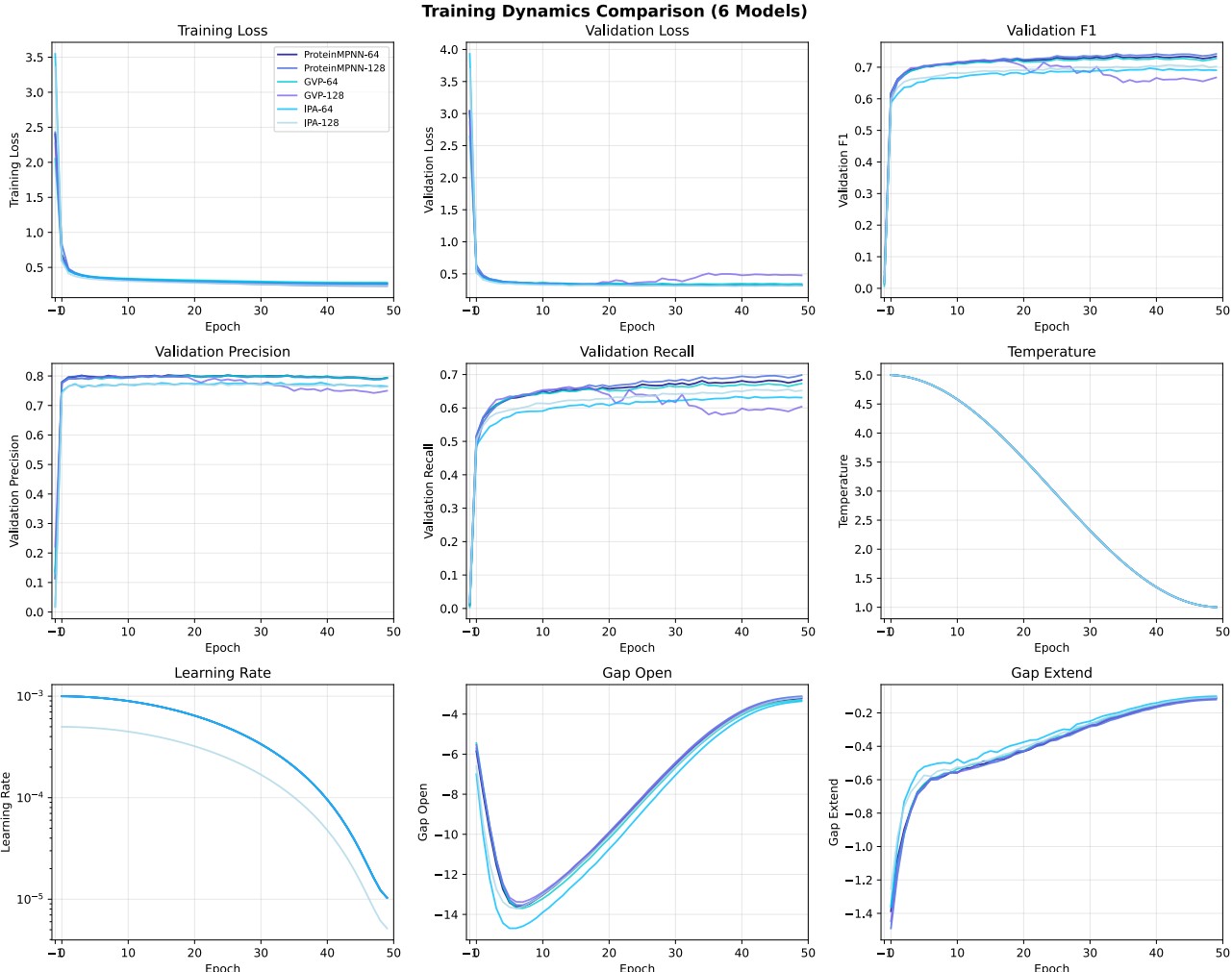

*Figure 6.* **Training dynamics for neural network encoders.** 3×3 grid showing metrics over 50 epochs for six models: ProteinMPNN, GVP, and IPA at 64 and 128 embedding dimensions. *Row 1:* Training loss, validation loss, and validation F1. *Row 2:* Validation precision, validation recall, and temperature schedule. *Row 3:* Learning rate (cosine decay), gap open penalty, and gap extend penalty. ProteinMPNN-128 achieves the highest final F1 (0.748), with all neural network encoders substantially outperforming substitution matrices. Learned gap penalties converge to consistent values across all architectures ($g_o \approx -3.2$, $g_e \approx -0.11$).

## G.3. Learned Substitution Matrices

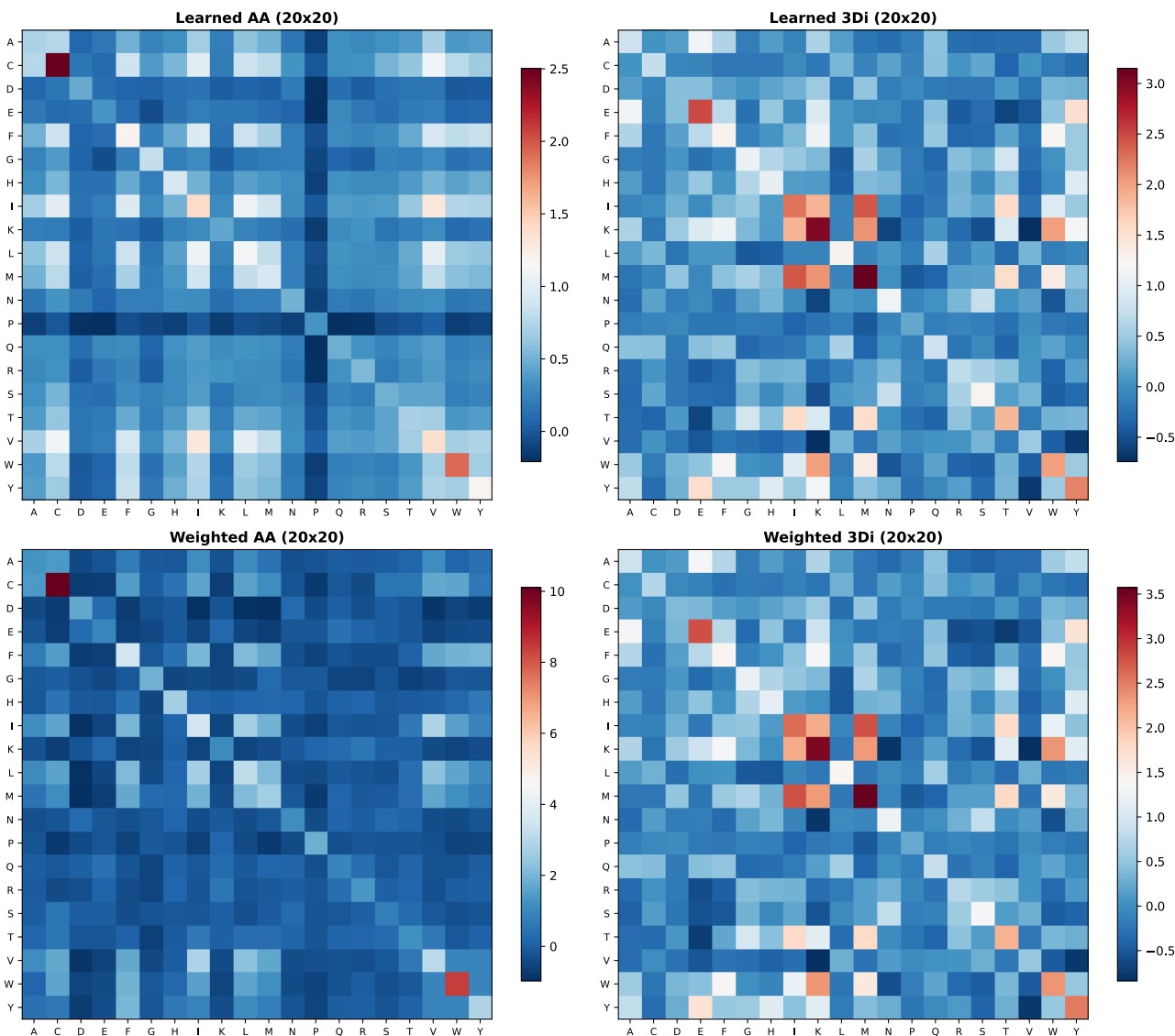

*Figure 7.* **Learned $20 \times 20$ substitution matrices.** $2 \times 2$ grid comparing AA and 3Di matrices from two training configurations. *Top row:* Learned AA (left) and Learned 3Di (right) from single-alphabet models. *Bottom row:* Weighted AA (left) and Weighted 3Di (right) from the combined model. Full-rank Euclidean parameterization $S = EE^{\top}$ with $E \in \mathbb{R}^{20 \times 20}$. AA matrices show diagonal dominance with off-diagonal structure capturing physicochemical similarities. 3Di matrices exhibit stronger block structure reflecting local backbone geometry, explaining the F1 gap (0.47 vs 0.23). Color scale: red (high similarity) to blue (low similarity).

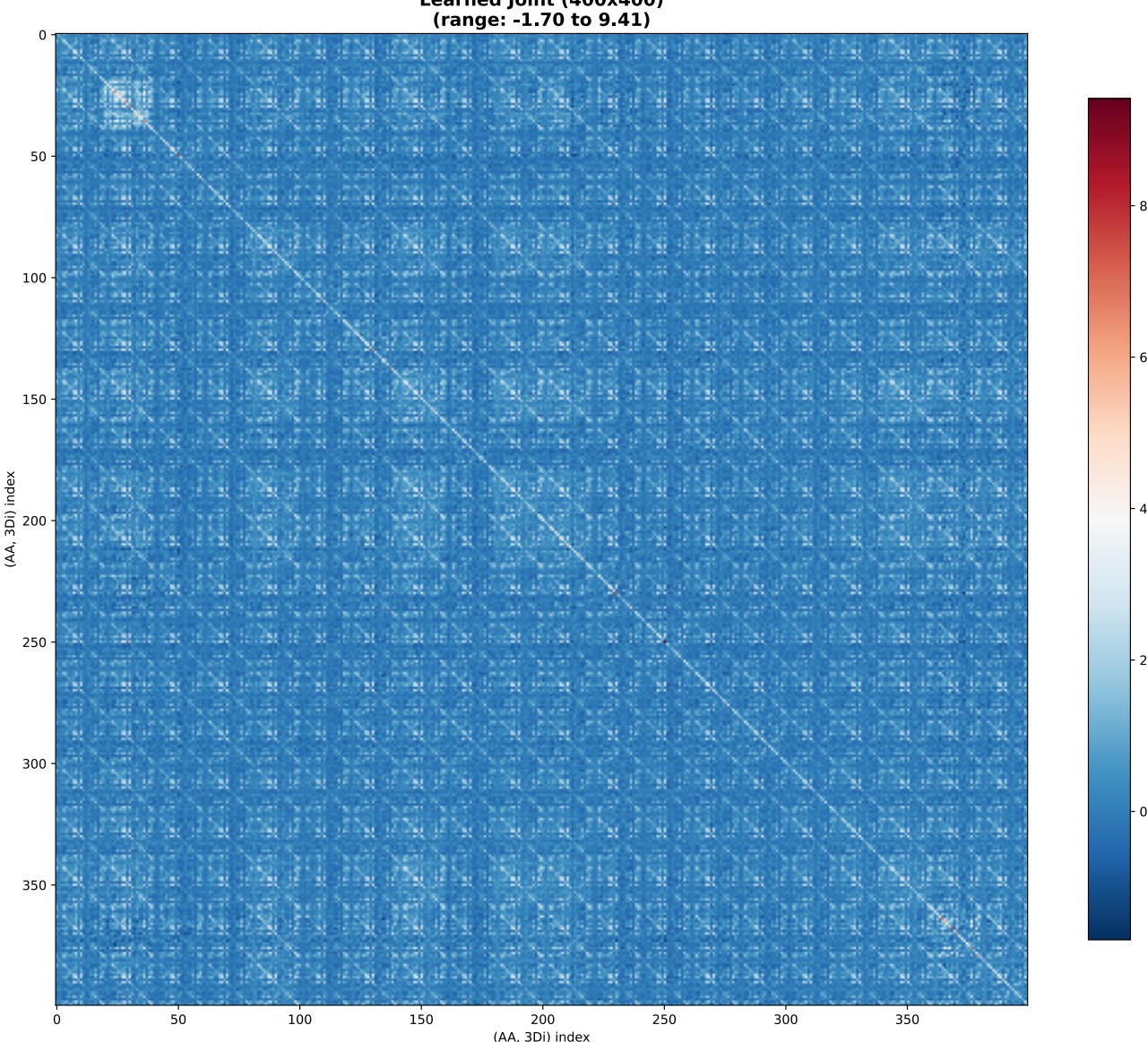

*Figure 8.* **Learned 400×400 joint (AA, 3Di) substitution matrix.** Each entry corresponds to a pair of (amino acid, 3Di) tokens, capturing correlations between sequence and local structure. The visible block structure reflects: (1) diagonal blocks where both AA and 3Di match, (2) off-diagonal blocks capturing AA–3Di cross-correlations (e.g., hydrophobic residues in helical conformations). This joint representation achieves the best substitution matrix performance (F1: 0.510), demonstrating that sequence–structure correlations provide complementary information beyond what either alphabet captures alone. Full-rank Euclidean parameterization $S = EE^{\top}$ with $E \in \mathbb{R}^{400 \times 400}$.

### G.4. PCA Analysis

Beyond quantitative metrics, the Euclidean parameterization $S = EE^{\top}$ yields interpretable representations (Figure 9). Despite training only on binary alignment labels with BCE loss, the learned amino-acid embeddings recover meaningful biochemical groupings: hydrophobic residues (A, I, L, M, V) cluster tightly, aromatics (F, W, Y) form a distinct group nearby, charged residues separate by sign (positive H, K, R vs. negative D, E), and polar residues cluster with chemically similar pairs adjacent (amides N, Q together; hydroxyls S, T together). Strikingly, the embeddings capture structure *beyond* BLOSUM62: traveling from the aromatic cluster toward the charged residues traces a gradient from most buried (W) to most solvent-exposed (D, E), reflecting the hydrophobicity scale that governs protein folding, which is not evident in the BLOSUM62 PCA. This emergent structure, discovered purely from structural alignment supervision without any

biochemistry labels, validates that the model has learned biologically meaningful similarity functions. The 3Di embeddings show different structure, reflecting that the structural alphabet captures local backbone geometry rather than side-chain chemistry.

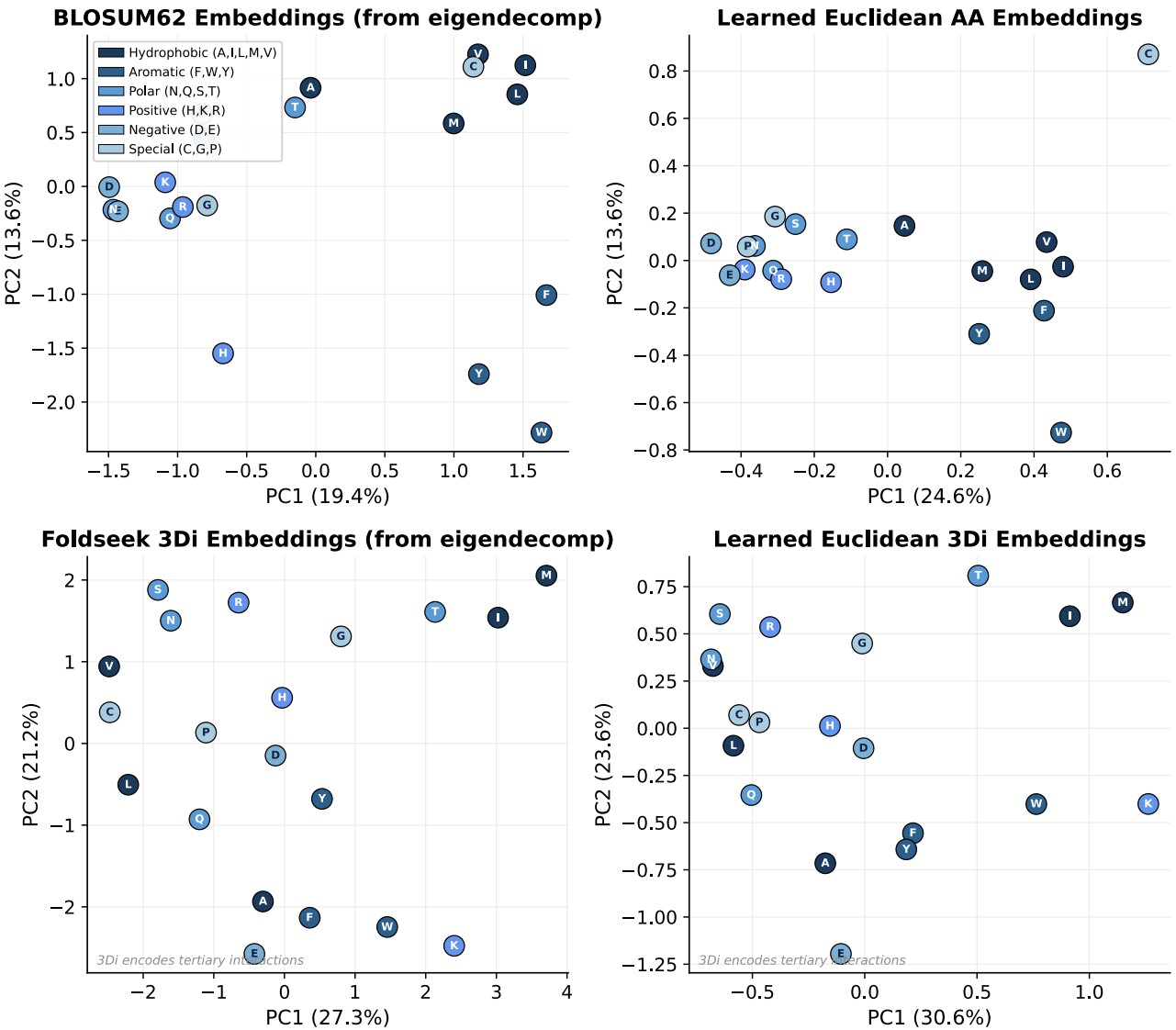

*Figure 9.* **PCA visualization of learned embeddings.** 2×2 grid comparing reference and learned embeddings for both alphabets. *Top row:* Amino acid embeddings: BLOSUM62 eigendecomposition (left) vs our learned Euclidean embeddings (right). *Bottom row:* 3Di embeddings: Foldseek eigendecomposition (left) vs our learned Euclidean embeddings (right). Points colored by physicochemical class: hydrophobic (A,I,L,M,V), aromatic (F,W,Y), polar (N,Q,S,T), positive (H,K,R), negative (D,E), and special (C,G,P). Learned embeddings recover meaningful clustering despite training purely on structural alignment; hydrophobic residues cluster together, charged residues separate by sign. Variance explained shown on axes.

## G.5. Substitution Matrix Heatmaps

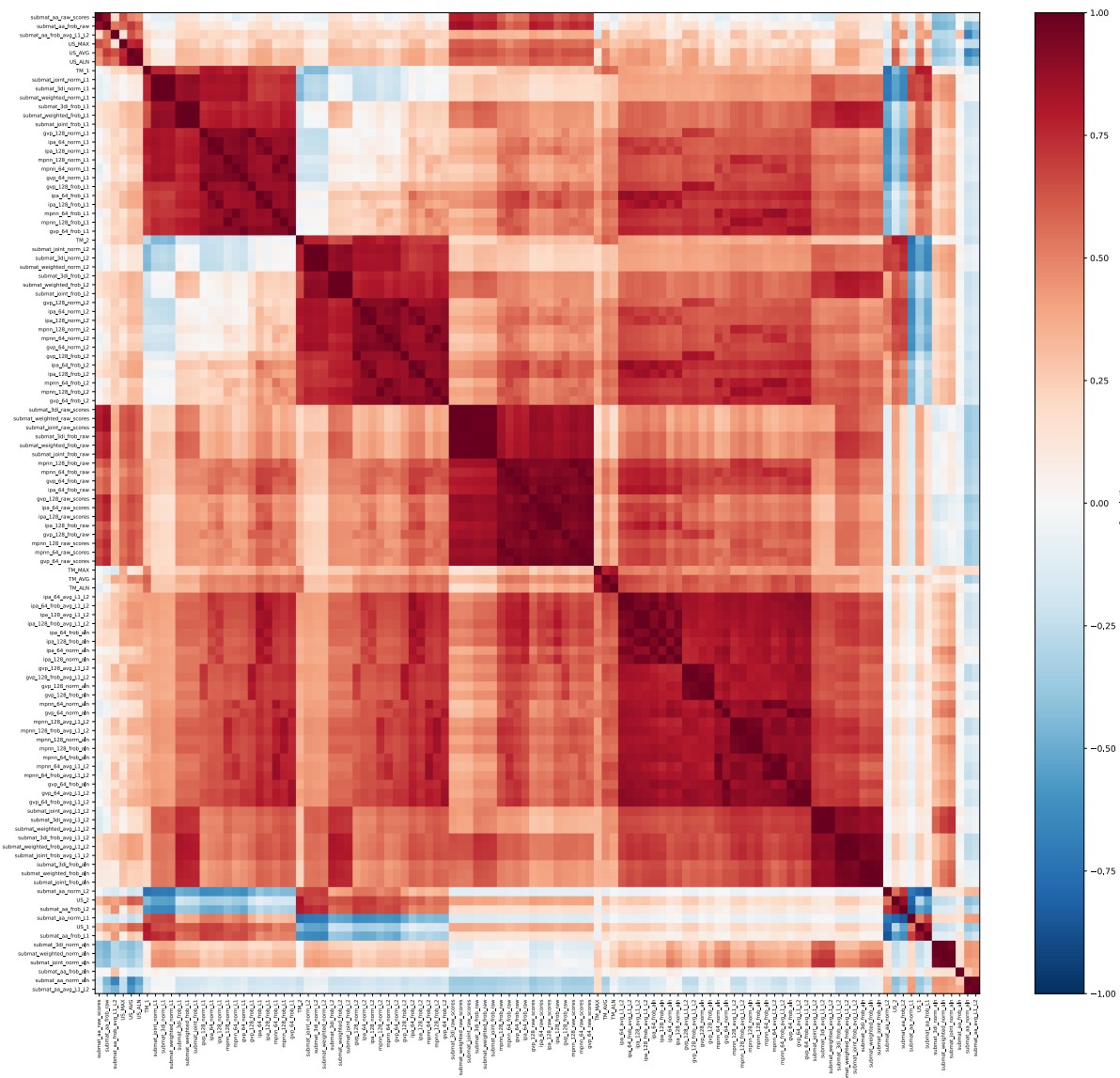

*Figure 10.* **Correlation matrix of alignment scores (hierarchically clustered).** Pairwise Pearson correlations between alignment scores from all models and metrics, reordered by hierarchical clustering (Ward linkage). Rows/columns represent different scoring methods: substitution matrices (AA, 3Di, weighted, joint), neural network encoders (ProteinMPNN, GVP, IPA at 64/128 dims), and reference metrics (TM-score, etc.). Clustering reveals that neural network encoders form a tight cluster (high mutual correlation), while substitution matrices show weaker correlations with learned methods. The block structure indicates which methods capture similar structural information.

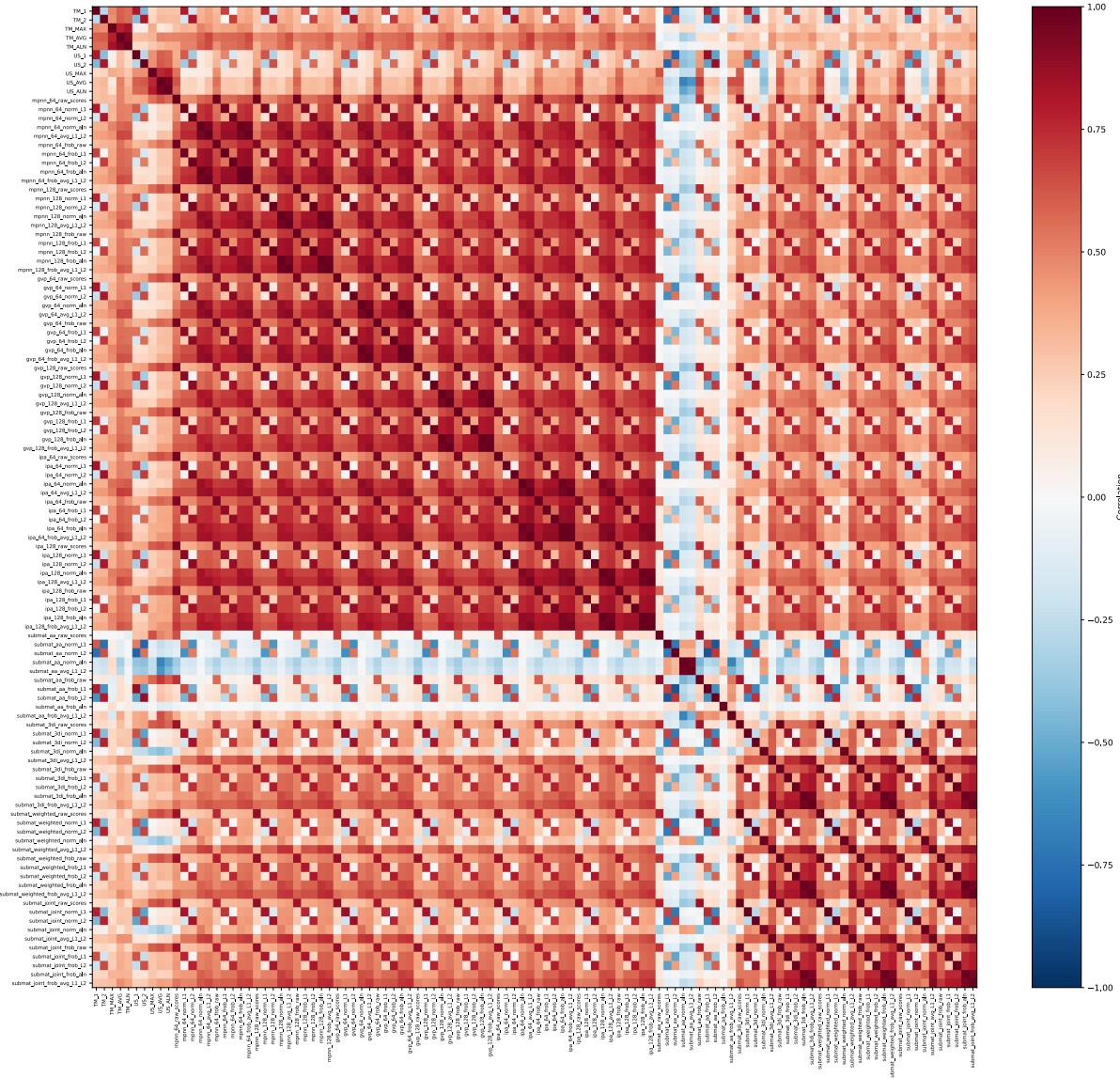

*Figure 11.* **Correlation matrix of alignment scores (unclustered).** The same correlation matrix as Figure 10, but with rows and columns in original order (not reordered by clustering). Methods are grouped by type: substitution matrices, neural network encoders (by architecture and dimension), and reference metrics. The checkerboard pattern reflects alternating raw scores and normalized variants for each method. Compare to Figure 10 to see how hierarchical clustering reveals the underlying similarity structure between scoring methods.

## G.6. Alignment Quality

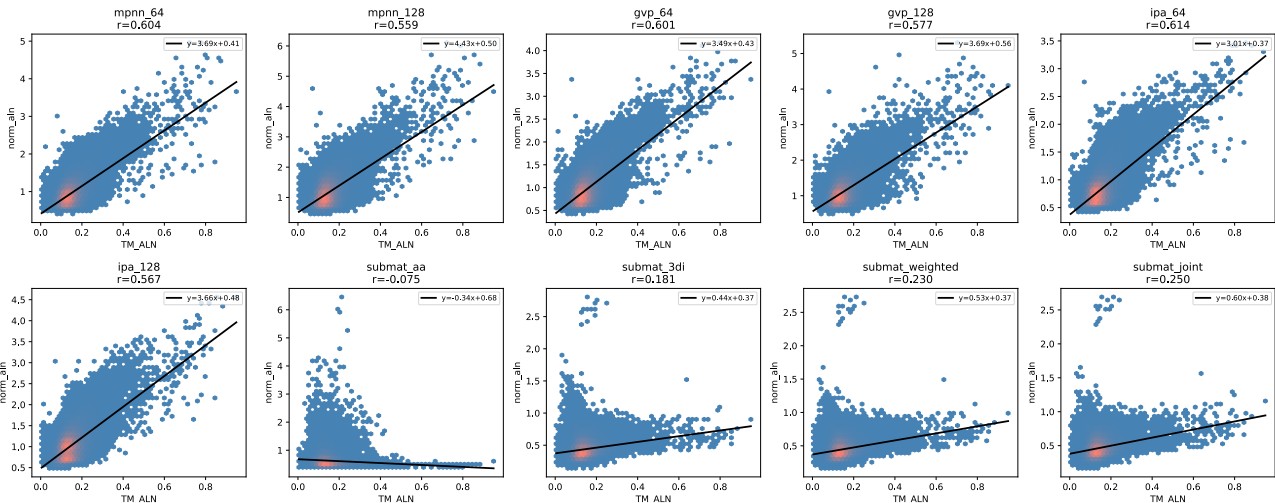

*Figure 12.* **Smith–Waterman score versus TM-align TM-score.** 2×5 grid comparing each model's Smith–Waterman alignment score against the TM-align TM-score for validation pairs. Each point is one protein pair; line shows linear regression. *Top row:* Neural network encoders (ProteinMPNN, GVP, and IPA at 64/128 dims) achieve $r \approx 0.56$–$0.61$. *Bottom row:* Substitution matrices: AA-only ($r = 0.08$), 3Di-only ($r = 0.18$), weighted ($r = 0.23$), and joint ($r = 0.25$) show weaker correlations. Neural network encoders achieve $\sim 2.5 \times$ higher correlation than the best substitution matrix, demonstrating that their alignment scores better track structural similarity.

## G.7. Example Alignments

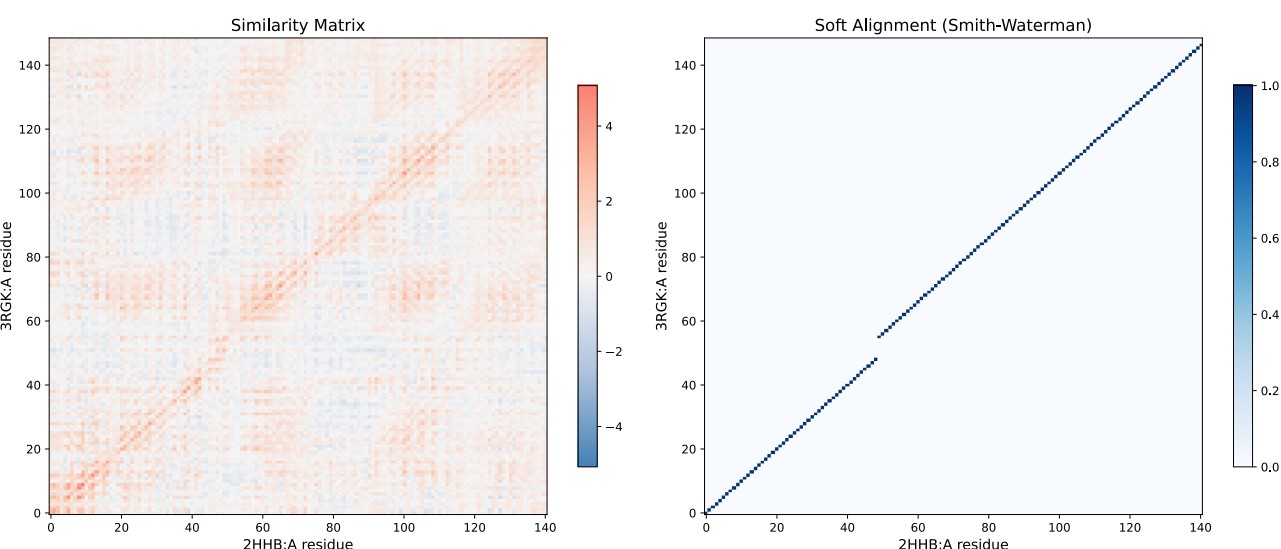

*Figure 13.* **Example alignment: myoglobin (3RGK:A) vs hemoglobin (2HHB:A).** These globin-family proteins share the characteristic 8-helix fold but differ in sequence (25% identity) and length (153 vs 141 residues). *Left:* Pairwise similarity matrix $S_{ij} = \langle h_i^{(1)}, h_j^{(2)} \rangle$ from ProteinMPNN encoder; bright diagonal bands correspond to the conserved helices (A–H), while dark regions indicate structurally divergent loops. *Right:* Soft alignment marginals $P_{T,ij}$ from differentiable Smith–Waterman at $T = 1.0$. The alignment confidently matches the conserved helical core (sharp diagonal in central region) while showing appropriate uncertainty at the N/C-termini and inter-helix loops where structural correspondence is ambiguous. This uncertainty quantification is a key advantage of soft alignment over hard Viterbi decoding.

## G.8. Runtime Speedups

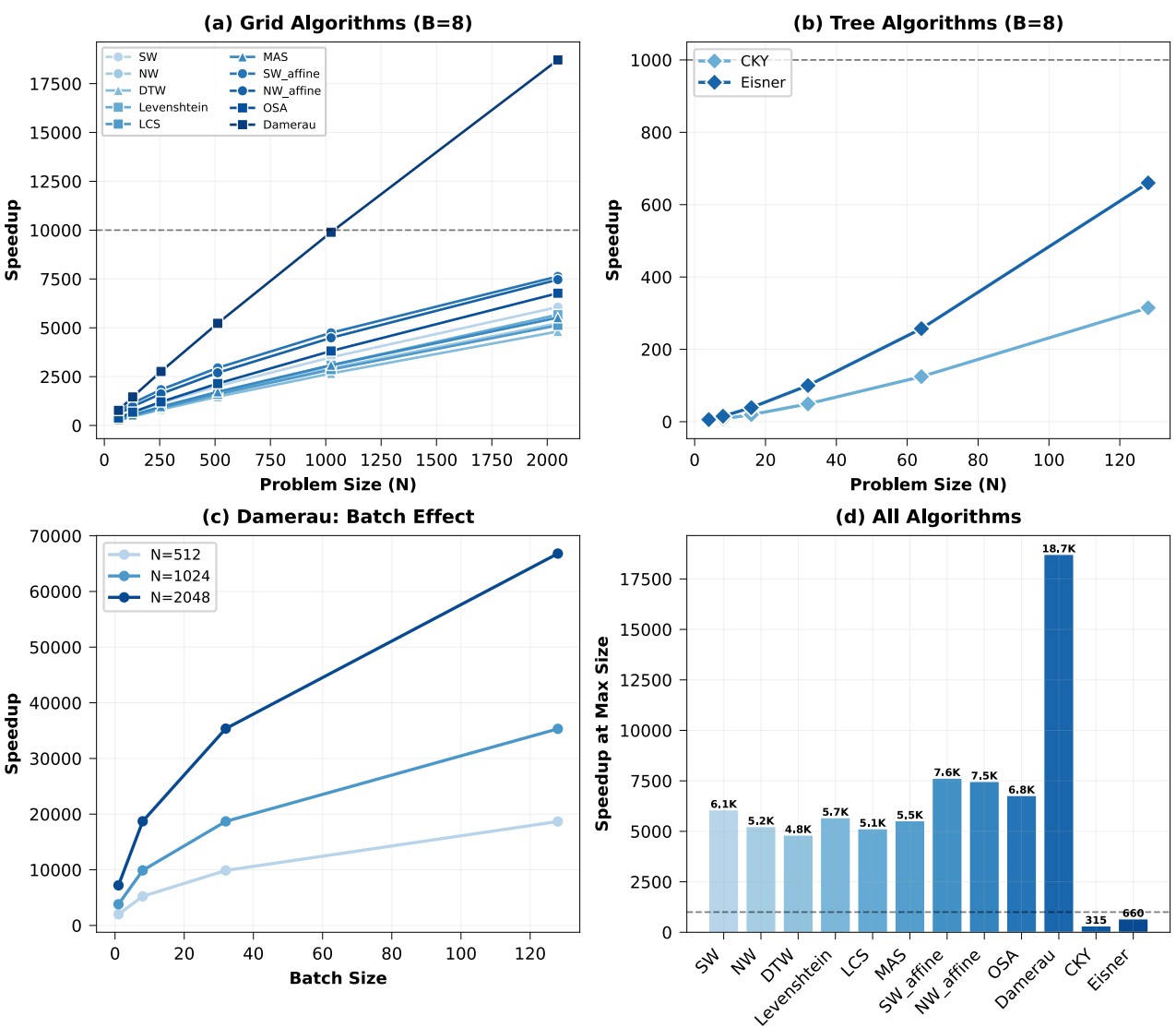

*Figure 14.* **Runtime speedup of d²p CUDA kernels.** *(a)* Grid-based algorithms vs sequence length $N$: SW, NW, DTW, and edit distances achieve 100–20,000× speedup via anti-diagonal wavefront parallelism. *(b)* Tree-based algorithms vs $N$: CKY and Eisner achieve 21–700× speedup via span-length parallelism, from ∼21× at small $N$ to several hundred-fold at the largest sizes (panel d). *(c)* Batch size effect for Damerau at different $N$. *(d)* Summary across all algorithms at maximum tested size. Benchmarks on NVIDIA GH200 with FP32 precision.

# H. Parsing Experiments

Alignment is the structure-decisive regime; constituency parsing is the opposite end of the spectrum, where data are abundant and the bitter lesson bites. We use the same differentiable machinery, now the CKY chart of Section C whose span marginals are the structured self-attention of Figure 4, to ask a sharper question: how much supervision does the chart need? We compare two supervised objectives over one encoder-to-CKY model: weighted BCE, which labels every candidate span as constituent or not, and a structured CKY CRF, which supervises only the gold tree's aggregate score against the partition over all valid binary trees. In the CRF the chart, not the labels, supplies tree validity; the gold tree enters through a single score term, with no per-span or split labels. We run both across nine domains spanning natural language and eight programming languages.

*Table 5.* **Parsing transfer: structured gold-tree-score supervision matches dense span labels.** Constituency-parsing validation $F_1$ across nine domains, comparing local span-label supervision (weighted BCE) against a structured CKY CRF supervised only by the gold tree's score (tree validity supplied by the chart, not learned from labels). The CRF exceeds BCE on three of nine domains and is within 0.003 $F_1$ on two more; the mean gap is $-0.008$ $F_1$.

| Domain | Type | Span labels (wBCE) | Tree score (CRF) |
|---|---|---|---|
| English (PTB) | natural lang. | 0.575 | 0.572 |
| C | programming | 0.653 | **0.661** |
| C++ | programming | 0.641 | **0.651** |
| Python | programming | 0.564 | 0.563 |
| Common Lisp | programming | 0.524 | 0.503 |
| Rust | programming | 0.520 | **0.532** |
| Racket | programming | 0.512 | 0.500 |
| Scheme | programming | 0.494 | 0.455 |
| Clojure | programming | 0.411 | 0.388 |

*Table 6.* **Parsing ablation (validation $F_1$, final epoch).** Encoder and temperature-regime sweep under supervised (BCE) training on English (PTB) and Scheme. Stronger encoders help; the learnable temperature finds very different equilibria by domain ($T \approx 8.2$ for PTB, $T \approx 0.008$ for Scheme). The NLL objective, lacking partition-function normalization, collapses ($F_1 = 0.10$).

| | English (PTB) | | Scheme | |
|---|---|---|---|---|
| Encoder | Cosine $T$ | Learn $T$ | Cosine $T$ | Learn $T$ |
| Embedding | 0.472 | 0.407 | 0.439 | 0.364 |
| MLP | 0.535 | 0.529 | 0.458 | 0.400 |
| Transformer | 0.563 | **0.575** | **0.494** | 0.391 |
| *NLL objective* (PTB, transformer): $F_1 = 0.104$ (collapse) | | | | |

Table 5 shows the structured gold-tree-score CRF essentially matches dense span-label supervision: the gap is only $-0.003$ $F_1$ on English (PTB) and $-0.008$ on average across the nine domains, and the CRF *exceeds* BCE on three (C, C++, Rust). The partition normalization makes the weaker supervision suffice: valid trees compete for probability mass, so raising the gold tree's score reshapes the whole distribution, whereas a plain NLL objective without it collapses (Table 6, $F_1 = 0.10$ on PTB). Encoder strength still matters in this data-rich regime: a transformer encoder beats MLP and embedding baselines (Table 6), and the learnable temperature converges to sharply different equilibria by domain, from $T \approx 8.2$ on English to $T \approx 0.008$ on Scheme, the model annealing nearly to a hard parser where the grammar is deterministic. Parsing accuracy here trails scaled autoregressive language models; structure earns its place through supervision efficiency and an interpretable parse, not raw $F_1$.

# I. Experimental Details

This appendix provides complete details on training methodology and benchmark configurations for reproducibility.

## I.1. Full Ablation Tables

Tables 7 to 10 report the complete ablation grids: validation $F_1$ for every algorithm crossed with every training regime (default learned gaps, learned temperature, straight-through, and frozen $g=-10$), for the ProteinMPNN and GVP encoders and the learned and stock substitution matrices. Two patterns recur. Learning the gap penalties (default), or jointly learning the temperature, is consistently best; freezing gaps at $g=-10$ collapses the single-penalty linear models in particular (e.g. MPNN-128 SW-linear $0.743 \rightarrow 0.321$), whereas straight-through estimation, which commits to a hard alignment, recovers much of the loss and attains higher lDDT at lower $F_1$. Gap-free relaxations (DTW, softmax, Sinkhorn) trail every gap-aware DP variant at both encoder widths.

*Table 7.* **Full ProteinMPNN ablation.** Validation $F_1$ across algorithms (rows) and training regimes (columns) for both encoder widths. STE: straight-through; Frozen: $g=-10$. Dashes mark settings without that regime (e.g. gap-free attention has no frozen-gap variant).

| Encoder | Algorithm | Default | Learn $T$ | STE | Frozen |
|---------|-----------|---------|-----------|-----|--------|
| MPNN-128 | SW-affine | 0.748 | 0.745 | 0.712 | 0.705 |
| | SW-linear | 0.743 | 0.747 | 0.723 | 0.321 |
| | NW-affine | 0.744 | 0.740 | 0.713 | 0.704 |
| | NW-linear | 0.750 | 0.748 | 0.733 | 0.718 |
| | DTW | 0.046 | 0.043 | – | – |
| | Softmax | 0.691 | – | – | – |
| | Sinkhorn | 0.682 | – | – | – |
| MPNN-64 | SW-affine | 0.739 | 0.736 | 0.697 | 0.692 |
| | SW-linear | 0.736 | 0.706 | 0.718 | 0.128 |
| | NW-affine | 0.732 | 0.733 | 0.697 | 0.691 |
| | NW-linear | 0.738 | 0.737 | 0.722 | 0.704 |
| | DTW | 0.043 | 0.043 | – | – |
| | Softmax | 0.675 | – | – | – |
| | Sinkhorn | 0.663 | – | – | – |

*Table 8.* **Full GVP ablation.** Validation $F_1$ across algorithms and training regimes for both encoder widths.

| Encoder | Algorithm | Default | Learn $T$ | STE | Frozen |
|---------|-----------|---------|-----------|-----|--------|
| GVP-128 | SW-affine | 0.725 | 0.639 | 0.698 | – |
| | SW-linear | 0.727 | 0.730 | 0.718 | – |
| | NW-affine | 0.735 | 0.706 | 0.700 | – |
| | NW-linear | 0.744 | 0.727 | 0.650 | – |
| | DTW | 0.043 | 0.041 | – | – |
| | Softmax | 0.529 | – | – | – |
| | Sinkhorn | 0.622 | – | – | – |
| GVP-64 | SW-affine | 0.733 | 0.732 | 0.686 | – |
| | SW-linear | 0.727 | 0.678 | 0.710 | – |
| | NW-affine | – | – | 0.690 | – |
| | NW-linear | – | – | 0.712 | – |
| | DTW | 0.039 | – | – | – |
| | Softmax | 0.664 | – | – | – |
| | Sinkhorn | 0.653 | – | – | – |

### I.2. Substitution-Matrix Baselines

We evaluate learned classical substitution matrices using a full-rank Euclidean parameterization $S = EE^\top$ with $E \in \mathbb{R}^{n \times n}$ where $n$ is the alphabet size: each residue's one-hot encoding $x \in \{0,1\}^n$ maps to an embedding $h = Ex \in \mathbb{R}^n$ with pairwise similarity $S_{ij} = \langle h_i^{(1)}, h_j^{(2)} \rangle$, identical to the neural network encoder case (Llinares-López et al., 2023). We compare four configurations: amino acids only (AA, 20×20), the 3Di structural alphabet only (van Kempen et al., 2024) (a re-learned, end-to-end version of Foldseek's approach, given every advantage over its hand-crafted matrix), a weighted combination of separate AA and 3Di matrices, and a joint 400×400 matrix over (AA, 3Di) pairs. Training uses AdamW (learning rate $10^{-3}$), cosine temperature annealing from $T = 5.0$ to $T = 1.0$, and the self-balancing weighted BCE loss against TM-align correspondences, with gap penalties learned jointly via cross-Jacobians.

### I.3. Neural Network Encoder Results

Table 12 reports the full neural network encoder results: validation F1, precision, recall, lDDT, and learned gap penalties for ProteinMPNN, GVP, and IPA at 64 and 128 dimensions, all with affine Smith–Waterman and cross-Jacobian-learned gaps. The lDDT ceiling is relative to the TM-align reference (0.489); ProteinMPNN-128 reaches 91% of it.

*Table 9.* **Full learned substitution-matrix ablation.** Validation $F_1$ for the four Euclidean parameterizations across algorithms and regimes.

| Encoder | Algorithm | Default | Learn $T$ | STE | Frozen |
|---------|-----------|---------|-----------|-----|--------|
| AA | SW-affine | 0.232 | 0.255 | 0.167 | 0.051 |
| | SW-linear | 0.162 | 0.136 | 0.141 | 0.049 |
| | NW-affine | 0.227 | 0.253 | 0.163 | 0.189 |
| | NW-linear | 0.062 | 0.053 | 0.187 | 0.154 |
| | DTW | 0.053 | 0.053 | – | – |
| | Softmax | 0.000 | 0.000 | – | – |
| 3Di | SW-affine | 0.468 | 0.478 | 0.381 | 0.459 |
| | SW-linear | 0.378 | 0.343 | 0.403 | 0.084 |
| | NW-affine | 0.442 | 0.460 | 0.363 | 0.434 |
| | NW-linear | 0.330 | 0.317 | 0.367 | 0.391 |
| | DTW | 0.054 | 0.054 | – | – |
| | Softmax | 0.000 | 0.000 | – | – |
| Weighted | SW-affine | 0.500 | 0.509 | 0.410 | 0.476 |
| | SW-linear | 0.422 | 0.392 | 0.428 | 0.094 |
| | NW-affine | 0.476 | 0.493 | 0.395 | 0.456 |
| | NW-linear | 0.377 | 0.365 | 0.394 | 0.409 |
| | DTW | 0.055 | 0.054 | – | – |
| | Softmax | 0.001 | 0.001 | – | – |
| Joint | SW-affine | 0.510 | 0.519 | 0.422 | 0.488 |
| | SW-linear | 0.431 | 0.402 | 0.432 | 0.102 |
| | NW-affine | 0.488 | 0.504 | 0.409 | 0.465 |
| | NW-linear | 0.388 | 0.377 | 0.403 | 0.422 |
| | DTW | 0.055 | 0.055 | – | – |
| | Softmax | 0.003 | 0.003 | – | – |

*Table 10.* **Full stock (fixed, non-learned) substitution-matrix ablation.** Validation $F_1$ with published gap penalties; only Default and Learn-$T$ apply (no encoder to freeze or straight-through).

| Encoder | Algorithm | Default | Learn $T$ | STE | Frozen |
|---------|-----------|---------|-----------|-----|--------|
| BLOSUM62 | SW-affine | 0.173 | 0.131 | – | – |
| | SW-linear | 0.120 | 0.100 | – | – |
| | NW-affine | 0.231 | 0.246 | – | – |
| | NW-linear | 0.105 | 0.045 | – | – |
| 3Di (MAT3DI) | SW-affine | 0.420 | 0.454 | – | – |
| | SW-linear | 0.371 | 0.286 | – | – |
| | NW-affine | 0.413 | 0.451 | – | – |
| | NW-linear | 0.336 | 0.308 | – | – |
| Foldseek | SW-affine | 0.433 | 0.472 | – | – |
| | SW-linear | 0.389 | 0.328 | – | – |
| | NW-affine | 0.431 | 0.490 | – | – |
| | NW-linear | 0.374 | 0.353 | – | – |

## I.4. Temperature Scaling of Learned Gap Penalties

We examined how the learned affine gap penalties vary with the alignment temperature in the SW-affine setting. To avoid the high-temperature transient observed early in the sweep, we restricted the analysis to the equilibrated window $T \in [1, 4]$ and fit each learned gap parameter as a linear function of temperature, $g(T) = \kappa T + b$, for all ten SW-affine modalities (six neural network encoders, four learned substitution matrices). The fits are nearly perfect (Table 13): across the six neural network encoders the gap-open penalty has mean slope $\kappa_o = -2.79$ with $R^2 \geq 0.9994$ and the gap-extend penalty mean slope $\kappa_e = -0.120$ with $R^2 \geq 0.9976$; the four learned substitution-matrix baselines show the same behavior with somewhat steeper gap-open scaling ($\kappa_o = -3.55$, minimum $R^2 = 0.994$). Temperature and the learned gap penalties are thus strongly coupled: over the operating range, changing temperature is largely absorbed by a compensatory linear

*Table 11.* **Learned substitution matrices for protein structure alignment.** Validation metrics on SCOPe40 (65,880 pairs) for the four parameterizations, against TM-align ground truth. Gap penalties ($g_o$: open, $g_e$: extend) are learned jointly. 3Di dramatically outperforms sequence-only (F1: 0.47 vs 0.23); the joint matrix is best.

| Model | Val F1 | Precision | Recall | lDDT | Gap Open | Gap Extend |
|---|---|---|---|---|---|---|
| AA-only (20×20) | 0.232 | **0.704** | 0.142 | 0.238 | −7.06 | +0.10 |
| 3Di-only (20×20) | 0.468 | 0.656 | 0.366 | **0.297** | −5.55 | −0.05 |
| Weighted (AA+3Di) | 0.500 | 0.680 | 0.398 | 0.288 | −5.44 | −0.05 |
| Joint (400×400) | **0.510** | 0.678 | **0.411** | 0.293 | −5.39 | −0.06 |

*Table 12.* **Learned neural network encoders for protein structure alignment.** Validation metrics on SCOPe40. lDDT is computed with the OpenStructure implementation (Mariani et al., 2013; Biasini et al., 2010) on the hard (argmax) alignment, using the symmetric coverage-normalized protocol of Section I.5 so that unaligned residues count against the score.

| Model | Val F1 | Precision | Recall | lDDT | Gap Open | Gap Extend |
|---|---|---|---|---|---|---|
| ProteinMPNN-128 | **0.748** | 0.797 | **0.705** | **0.445** | −3.11 | −0.12 |
| ProteinMPNN-64 | 0.739 | **0.798** | 0.689 | 0.435 | −3.21 | −0.11 |
| GVP-64 | 0.733 | 0.797 | 0.680 | 0.417 | −3.30 | −0.11 |
| GVP-128 | 0.725 | 0.786 | 0.675 | 0.430 | −3.16 | −0.11 |
| IPA-128 | 0.706 | 0.768 | 0.656 | 0.421 | −3.17 | −0.11 |
| IPA-64 | 0.690 | 0.761 | 0.632 | 0.425 | −3.36 | −0.10 |
| TM-align (reference) | 1.000 | – | – | 0.489 | – | – |

rescaling of the affine gap terms, with a Boltzmann-like slope $\kappa_o \approx -2.8$ that is encoder-agnostic within the neural network family (Figure 15).

## I.5. lDDT Evaluation (OpenStructure)

We report lDDT (Mariani et al., 2013) computed with the OpenStructure implementation (Biasini et al., 2010) as an external check on structural quality, on the hard (argmax) alignment. Because lDDT is asymmetric, we report the average of the two directional scores: protein 1 against protein 2 and protein 2 against protein 1. The number of unaligned positions depends on each protein's length, so we normalize each directional score by the fraction of residues covered in that reference chain before averaging, ensuring that unaligned residues count against the score rather than inflating an aligned-subset value. The lDDT ceiling percentages are taken relative to TM-align scored under the same protocol ($0.489$). DTW is excluded from lDDT evaluation because its paths may contain repeated residues and therefore do not define the one-to-one residue correspondence that lDDT requires; DTW is still evaluated by $F_1$ against the reference alignment.

## I.6. Training Configuration

**Dataset.** We use the SCOPe40 dataset (Chandonia et al., 2022), filtered to 40% sequence identity to reduce redundancy. The dataset contains 15,176 protein structures. We generate all unique protein pairs and compute structural alignments using TM-align (Zhang & Skolnick, 2005).

To prevent data leakage, we split proteins (not pairs) 66.7%/33.3% into train and validation sets. This induces three pair categories: train–train, val–val, and cross-split pairs. We discard all cross-split pairs, ensuring no protein appears in both training and validation. This yields 297,601 training pairs and 65,880 validation pairs (approximately 80%/20% of within-split pairs).

All models train on pairs with TM-score $> 0.6$ to focus on structurally similar proteins with meaningful alignments. Pairs with TM-score $< 0.6$ serve as an implicit out-of-distribution test set.

**Optimization.** Substitution matrix models use Adam optimizer with learning rate $10^{-3}$, cosine annealing to $10^{-5}$, batch size 128, and 50 epochs. Neural network encoders use AdamW with weight decay 0.01, gradient clipping (max norm 1.0), learning rate $10^{-3}$ (or $5 \times 10^{-4}$ for IPA-128), cosine annealing to 1% of initial, batch size 8 with length-based batching, and 50 epochs. Early stopping (patience 10, F1 metric) is enabled for IPA-128 only. Both use bfloat16 mixed precision.

*Table 13.* **Linear temperature scaling of learned SW-affine gap penalties over** $T \in [1, 4]$**.** Slope $\kappa$ and intercept $b$ of $g(T) = \kappa T + b$, fit independently for gap-open and gap-extend per modality. All $R^2 \geq 0.99$.

| | Gap open | | Gap extend | |
|---|---|---|---|---|
| Modality | $\kappa_o$ | $b_o$ | $\kappa_e$ | $b_e$ |
| MPNN-64 | $-2.744$ | $-0.373$ | $-0.124$ | $+0.005$ |
| MPNN-128 | $-2.741$ | $-0.270$ | $-0.122$ | $-0.006$ |
| GVP-64 | $-2.784$ | $-0.422$ | $-0.121$ | $+0.009$ |
| GVP-128 | $-2.734$ | $-0.340$ | $-0.122$ | $+0.001$ |
| IPA-64 | $-2.962$ | $-0.361$ | $-0.112$ | $+0.007$ |
| IPA-128 | $-2.768$ | $-0.366$ | $-0.120$ | $+0.009$ |
| AA (20×20) | $-3.650$ | $-2.381$ | $-0.174$ | $+0.193$ |
| 3Di (20×20) | $-3.487$ | $-1.306$ | $-0.156$ | $+0.044$ |
| Weighted (AA+3Di) | $-3.425$ | $-1.148$ | $-0.151$ | $+0.017$ |
| Joint (400×400) | $-3.644$ | $-0.453$ | $-0.119$ | $-0.077$ |

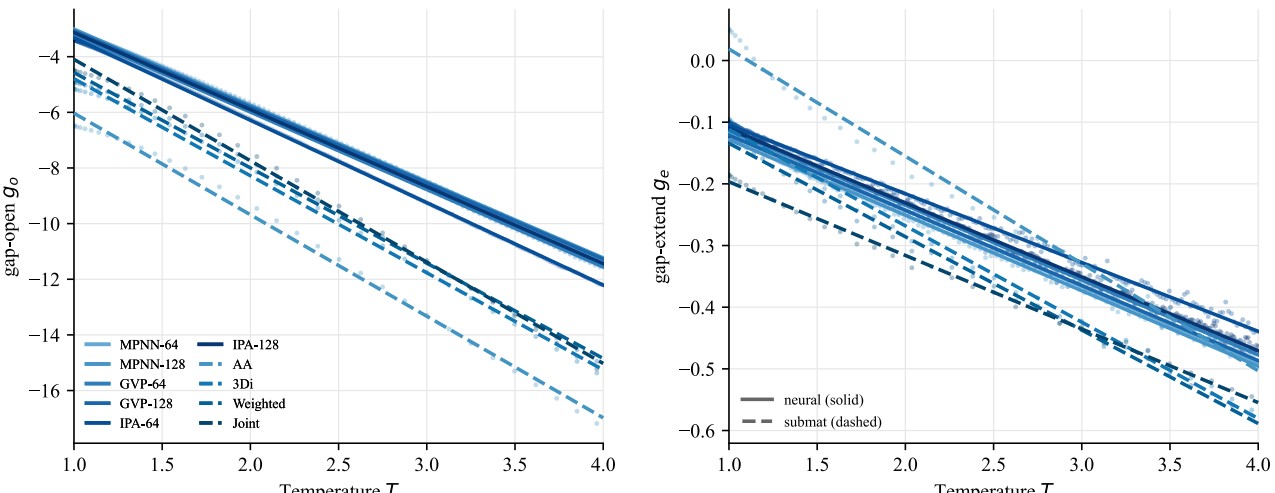

*Figure 15.* **Temperature scaling of learned SW-affine gap penalties.** Learned gap-open and gap-extend penalties over the equilibrated $T \in [1, 4]$ window, with per-modality linear regressions. Gap penalties scale nearly linearly with temperature, especially for the neural network encoders, indicating that temperature changes are largely compensated by rescaling the affine gap terms.

Neural network training applies Gaussian coordinate noise ($\sigma = 0.05$Å) for data augmentation.

**Temperature Schedule.** We use temperature annealing to stabilize training. The temperature $T$ follows a cosine schedule from $T_{\text{init}} = 5.0$ to $T_{\text{final}} = 1.0$ over all 50 training epochs. Higher initial temperatures produce smoother gradients, preventing early convergence to poor local minima.

**Loss Function.** Training uses self-balancing weighted BCE loss between predicted alignment marginals $P_{T,ij}$ and TM-align ground truth indicators $Y_{ij}$:

$$\mathcal{L} = -\frac{1}{nm} \sum_{i,j} [w_+ Y_{ij} \log P_{T,ij} + w_-(1 - Y_{ij}) \log(1 - P_{T,ij})],$$

where $w_+ = n_-/(n_+ + n_-)$ and $w_- = n_+/(n_+ + n_-)$ balance the positive and negative classes. We also experimented with soft F1 loss but found weighted BCE more stable.

**Gap Penalty Initialization.** Gap penalties are learned jointly with similarity parameters. For substitution matrices, we initialize $g_o = -10.0$ and $g_e = -1.0$. For neural network encoders, we initialize from $g_o \sim \mathcal{N}(-3.0, 0.1)$ and

$g_e \sim \mathcal{N}(-1.0, 0.1)$ following Trinquier et al. (2025). Gap penalties are unconstrained learnable parameters. The small positive extension score the AA-only baseline learns (Table 11) reflects model misspecification in the weak alphabet-only setting rather than a biological preference; neural network encoders and 3Di-based matrices converge to negative extension scores.

**Hardware and Training Time.** Training and inference were run on a mixture of single-node machines, with each run confined to a single GPU. GPUs used span Hopper (GH200, H100), Blackwell (RTX PRO 6000, RTX 5090), and Ada (RTX 6000 Ada, RTX 4090); machines were rented from Lambda Labs, Massed Compute, RunPod, and Vast AI. Substitution matrix models train in approximately 12 hours (50 epochs). Neural network encoder training time varies by architecture and embedding dimension: ProteinMPNN and GVP train in 24 hours (64d) or 36 hours (128d); IPA is significantly slower at 72 hours (64d) or 108 hours (128d) due to the attention computation overhead.

### I.7. Neural Network Encoder Architectures

**ProteinMPNN.** We use the encoder portion of ProteinMPNN (Dauparas et al., 2022) with 3 message passing layers. Edge features encode 25 atom-pair distances (all pairs of N, CA, C, O, C$\beta$) using 16 RBF centers ($D_{\min} = 2$Å, $D_{\max} = 22$Å), projected to hidden dimension. Relative position embeddings ($\pm 32$ positions + inter-chain token) provide sequence context. Each layer has 3-layer MLPs for node messages and edge updates with GELU activation and $4\times$ feedforward expansion. We use $k$-NN graphs with $k = 64$ based on C$\alpha$ distances. Parameters: 256K (64d), 962K (128d).

**GVP (Geometric Vector Perceptron).** We use a 3-layer GVP-GNN (Jing et al., 2021) following ESM-IF. Node scalars: 6 dihedral angles (cos/sin of $\eta, \psi, \omega$) + mask; node vectors: forward/backward chain directions + sidechain direction. Edge scalars: 16 RBF + 16 sinusoidal positional; edge vectors: unit displacement. Each layer has 3 GVPs for messages and 2 GVPs for feedforward with $4\times/2\times$ expansion (scalar/vector). Vector gating ensures SE(3) equivariance. $k$-NN graphs with $k = 64$. Parameters: 263K (64d), 1.03M (128d).

**IPA (Invariant Point Attention).** We adapt IPA from AlphaFold2 (Jumper et al., 2021) as a standalone encoder without pair representations. We use 6 layers (64d) or 8 layers (128d) with 8 attention heads, $c_{\text{hidden}} = 4$ (64d) or 8 (128d) per head, 4 query/key points and 8 value points per head. Rigid frames are computed from backbone via Gram-Schmidt orthogonalization. Initial node features use sinusoidal positional encoding. Parameters: 299K (64d), 925K (128d).

### I.8. Substitution Matrix Parameterization

All substitution matrices use Euclidean (PSD) parameterization $S = EE^\top$, constraining the matrix to be positive semi-definite. The embedding matrix $E$ is initialized from $\mathcal{N}(0, 0.1^2)$.

**Model configurations.** AA-only: $E \in \mathbb{R}^{20 \times 20}$ (402 parameters including gaps). 3Di-only: $E \in \mathbb{R}^{20 \times 20}$ (402 parameters). Joint: $E \in \mathbb{R}^{400 \times 400}$ over (AA, 3Di) pairs (160,002 parameters). Weighted: separate $E_{\text{AA}}, E_{\text{3Di}} \in \mathbb{R}^{20 \times 20}$ with learned combination weights $[\alpha, \beta] = \text{softmax}(w)$, yielding $S = \alpha \cdot E_{\text{AA}} E_{\text{AA}}^\top + \beta \cdot E_{\text{3Di}} E_{\text{3Di}}^\top$. The learned weights converge to $\alpha = 0.19$, $\beta = 0.81$, indicating structure (3Di) is $\sim 4\times$ more informative than sequence (AA).

### I.9. Parsing Experimental Setup

We evaluate whether the differentiable CKY layer can serve as a general structured-attention module for constituency parsing. All parsing experiments use the same encoder-to-CKY interface: an encoder maps each token sequence to contextual token embeddings, a span-scoring head converts those embeddings into span potentials, and the CKY layer defines a distribution over binary constituency trees.

**Data.** The parsing experiments span nine domains. English uses Penn Treebank constituency parses (via the NLTK treebank corpus) and Scheme uses S-expressions extracted from ChezScheme source; for the remaining seven languages (C, C++, Python, Rust, Racket, Clojure, and Common Lisp) we build the dataset by parsing valid source with the language's own compiler/interpreter parser and taking the resulting parse trees as gold. Examples are filtered to lengths 5–40 tokens. Each example is converted into a token sequence and a set of unlabeled constituent spans; the root span is removed from the training and evaluation targets because it is present in every valid tree. Data are split 80/10/10 into train/validation/test using seed 42, and vocabularies are built from the training split only. Both objectives are supervised by the domain's gold parses:

wBCE uses the gold spans as per-span labels, and the CRF uses the right-binarized gold tree's aggregate score. The root span is excluded from both training and evaluation.

**Encoder-agnostic parsing model.** Let a sentence be $x = (x_1, \ldots, x_n)$. We test three encoder families: a token-plus-position embedding encoder, an MLP encoder over token-plus-position embeddings, and a Transformer encoder. Each encoder produces contextual embeddings $h_1, \ldots, h_n = f_\theta(x)$. The CKY layer only sees these embeddings through span scores, so the parsing method is encoder-agnostic: the same structured layer can sit on top of non-contextual embeddings, MLP features, Transformer features, or any other encoder that emits one vector per token. For every candidate span $(i, j)$ we compute a bilinear span potential

$$s_{ij} = h_i^\top W h_j + b,$$

with invalid spans masked out. Leaf scores are learned from token position embeddings. For CKY, each binary merge over $(i, k)$ and $(k+1, j)$ receives the parent span score $s_{ij}$, and a learned span-length penalty is added:

$$m_{i,k,j} = s_{ij} + \lambda(j - i).$$

The differentiable CKY layer computes a partition function over all binary trees and returns span/split marginals. Temperature is either annealed from $T = 5$ to $T = 1$ with a cosine schedule or learned directly.

**Supervised wBCE span-marginal training.** The supervised parsing objective uses gold unlabeled spans but does not supervise split decisions directly. CKY produces posterior span probabilities by summing split marginals:

$$p_{ij} = \sum_k q_\theta(i, k, j \mid x).$$

Let $y_{ij} \in \{0, 1\}$ indicate whether span $(i, j)$ is present in the gold parse. The loss is a weighted binary cross-entropy over all valid non-root spans,

$$\mathcal{L}_{\text{wBCE}} = -\frac{1}{|\mathcal{S}|} \sum_{(i,j) \in \mathcal{S}} \left[ w_+ \, y_{ij} \log p_{ij} + (1 - y_{ij}) \log(1 - p_{ij}) \right],$$

where $\mathcal{S}$ is the set of valid non-root spans, and the positive weight is computed per batch to compensate for span sparsity:

$$w_+ = \text{clip}\left( \frac{N_{\text{neg}}}{N_{\text{pos}} + 1}, \, 1, \, 100 \right).$$

This objective trains the encoder, span scorer, temperature (when learnable), and span-length penalty through the CKY marginals. At evaluation time we decode a single hard tree with Viterbi CKY using the learned span scores and span penalty; predicted non-root spans are compared to gold non-root spans using unlabeled precision, recall, and $F_1$.

**Structured CRF (gold-tree-score supervision).** The CRF condition supervises the model through the *score of the gold tree*, not through per-span labels. Let $Y^\star$ be the gold parse right-binarized into the CKY search space, with non-leaf span set $\mathcal{S}(Y^\star)$ (the trivial root span, present in every tree, is excluded as in evaluation). The tree score and CKY partition are

$$S_\theta(Y, x) = \sum_{(i,j) \in \mathcal{S}(Y)} \left[ s_\theta(i, j) + \lambda(j - i) \right] + \sum_i \ell_\theta(i), \quad Z_\theta(x) = \sum_{Y' \in \mathcal{Y}(x)} \exp\big(S_\theta(Y', x)/T\big),$$

and the loss is the temperature-scaled negative log-likelihood of the gold tree,

$$\mathcal{L}_{\text{CRF}}(x, Y^\star) = T \log Z_\theta(x) - S_\theta(Y^\star, x) = -T \log p_T(Y^\star \mid x).$$

Because every binary tree spans all $n$ leaves exactly once, $\sum_i \ell_\theta(i)$ is constant across $\mathcal{Y}(x)$ and cancels from $p_T(\cdot \mid x)$, from the span marginals, and from Viterbi decoding; we retain it only for implementation uniformity. The gold tree enters *only* through $S_\theta(Y^\star, x)$; the partition $\log Z_\theta$ runs over the full CKY tree space and carries no gold labels. Tree validity and the noncrossing constraint are imposed by the chart, not learned: the loss never sees split labels or independent negative spans, only the aggregate score of the gold tree relative to all valid trees. This is a strictly weaker form of supervision than the per-span labels of wBCE, yet Table 5 shows it matches them to within 0.003 $F_1$. The two conditions also exercise

different parts of the framework. Weighted BCE is a loss *on the CKY marginals*, so its gradients flow through the HVP and cross-Jacobian machinery of Table 2; the structured CRF is a standard structured likelihood, whose gradients with respect to the encoder, the biaffine span scorer $s_\theta$, and the span-length penalty $\lambda$ are the usual gold-count-minus-expected-count terms supplied by inside–outside, with the learnable temperature $T$ (and, in the "all" variants, a split bias) handled by the same differentiable CKY chart.

**NLL ablation (collapse).** Replacing the tree-structured CRF partition with an unnormalized, independent per-span objective (no normalization over competing trees) removes the competition that makes gold-tree-score supervision work: the model trivially inflates all span scores and parsing $F_1$ collapses (PTB $F_1 \approx 0.10$). This isolates the partition normalization, not the gold-tree score alone, as what lets the structured CRF match dense span labels.

**Protocol and evaluation.** Each reported number is a *single* training run at fixed hyperparameters: no seed search, no checkpoint selection, no tuning. We report the final-epoch model rather than the best-validation checkpoint, so gold annotations never enter model selection. Gold spans supervise the wBCE loss and the gold tree's score supervises the CRF loss; neither uses split labels or independent negative spans. At test, both conditions decode a single tree by Viterbi CKY and compare predicted non-root spans to gold spans by unlabeled precision, recall, and $F_1$. Parsing is a demonstration that the identical d2p machinery (now CKY self-attention) transfers to a chart DP and learns its own parameters $(T, \lambda)$; it is not a state-of-the-art parser.

## I.10. Benchmark Configuration

**Hardware.** Runtime benchmarks were conducted on an NVIDIA GH200 Grace Hopper Superchip (96GB HBM3, 480GB LPDDR5, 64 ARM Neoverse V2 cores) from Lambda Labs, with CUDA 12.8 and PyTorch 2.11.

**Baseline Implementation.** The "standard PyTorch" baseline behind the headline speedups is the naive reference implementation shipped with $d^2p$ (`tests/reference.py`): each DP recurrence in native PyTorch tensor operations and autograd, looping over the sequential dimension in Python. PyTorch has no equivalent of JAX's `lax.scan`, so *every* PyTorch DP library parallelizes the sequential wavefront poorly; this is a structural limitation of the framework, not a strawman. We separately benchmark torch-struct (Rush, 2020), the state-of-the-art PyTorch library for differentiable structured prediction, as a stronger baseline on the operators it shares with $d^2p$, and use it for numerical validation, $d^2p$ reproducing its partition values and marginals to floating-point error.

**d2p CUDA Implementation.** Our CUDA kernels use wavefront parallelism for grid-based algorithms: each anti-diagonal is computed in parallel across all $(i, j)$ pairs satisfying $i + j = k$. For an $n \times m$ grid, this requires $n + m - 1$ sequential kernel launches with up to $\min(n, m)$ parallel threads per launch. Tree-based algorithms (CKY, Eisner) use span-length parallelism, processing all spans of length $\ell$ in parallel before proceeding to $\ell + 1$.

**Precision.** All benchmarks use FP32 precision. We verified numerical agreement between baseline and CUDA implementations to within $10^{-5}$ relative error for forward values and $10^{-4}$ for gradients.

**Memory.** CUDA kernels use $O(nm)$ memory for grid algorithms (storing $\alpha$, $\beta$, and gradients) and $O(n^2)$ for tree algorithms (storing chart entries). The baseline uses identical memory layouts for fair comparison.

**Timing Protocol.** Each configuration was run 100 times with 10 warmup iterations discarded. We report median wall-clock time for combined forward and backward passes. Speedup is computed as $t_{\text{baseline}}/t_{\text{cuda}}$.

## I.11. Numerical Stability

Differentiable DP requires careful numerical implementation to avoid overflow and underflow, particularly for long sequences where scores can span many orders of magnitude.

**Log-Space Computation.** All forward and backward values are computed and stored in log-space. Rather than computing $\alpha_i = \sum_j \exp(S_j)$ directly (which overflows for large $S_j$), we compute $\log \alpha_i = \text{LSE}(S_j)$ using the numerically stable

log-sum-exp identity:

$$\text{LSE}(v_1, \ldots, v_K) = v_{\max} + \log \sum_{k=1}^{K} \exp(v_k - v_{\max}),$$

where $v_{\max} = \max_k v_k$. Subtracting $v_{\max}$ ensures all exponents are non-positive, preventing overflow. The smallest exponent is at most $-\max_k |v_k - v_{\max}|$, which may underflow to zero, but this is numerically benign since those terms contribute negligibly to the sum.

**Temperature Scaling.** The temperature parameter $T$ provides additional numerical control. At low temperatures ($T \ll 1$), the distribution concentrates on optimal paths, but gradients can become numerically unstable. At high temperatures ($T \gg 1$), the distribution is smoother but may lose discriminative power. We found $T \in [0.5, 5.0]$ to be numerically stable for most applications, with $T = 1.0$ as a robust default.

**Kahan Summation.** For the backward pass and gradient accumulation, we employ Kahan summation (compensated summation) to reduce floating-point rounding errors when summing many small terms. Given a running sum $s$ and compensation term $c$ (initially zero), each new value $x$ is added as:

$$\begin{aligned}
y &= x - c, \\
t &= s + y, \\
c &= (t - s) - y, \\
s &= t.
\end{aligned}$$

This captures the low-order bits lost in the addition $s + y$, achieving effectively doubled precision for summation. This is particularly important for marginal computation where we sum over $O(nm)$ path contributions.

**Gradient Checkpointing.** For very long sequences where storing the full $O(nm)$ DP table exceeds GPU memory, we support gradient checkpointing. The forward pass stores only every $k$-th anti-diagonal (or span length for tree algorithms); the backward pass recomputes intermediate values as needed. This trades $O(k)$ additional compute for $O(nm/k)$ memory reduction.

### I.12. Parallelization Strategy

We benchmark d$^2$p CUDA kernels against standard PyTorch reference implementations (the naive tensor-and-autograd versions shipped in the repository, which we validate against torch-struct (Rush, 2020) to machine precision on the partition function and marginals) across all twelve algorithms (Figure 14) for sequence lengths $N \in \{128, 256, 512, 1024, 2048\}$. Grid-based algorithms achieve 100–20,000$\times$ speedup via anti-diagonal wavefront parallelism, scaling with problem size; tree-based algorithms (CKY, Eisner) achieve 21–700$\times$ via span-length parallelism, rising from $\sim$21$\times$ at small $N$ to several hundred-fold at the largest sizes; and affine-gap variants, with sequential state dependencies, still exceed 1000$\times$ at $N = 1024$. The figures above are against the naive reference; against torch-struct, the strongest existing PyTorch baseline for the four operators it shares with d$^2$p (NW, SW, CKY, Eisner), d$^2$p is 100–1000$\times$ faster at equal batch, scaling with problem size. torch-struct also exhausts GPU memory far sooner: its dense alignment chart OOMs at batched $N \geq 1024$, whereas d$^2$p runs those cases and continues to larger sizes. In absolute terms, d$^2$p computes Smith–Waterman forward+backward for $N = 256$ sequences in $\sim$26 ms (vs $\sim$19 s for standard PyTorch), all benchmarks including both passes with full gradient computation. Across frameworks, d$^2$p is about 2$\times$ faster than the differentiable Smith–Waterman implementation of SoftAlign (Trinquier et al., 2025) (JAX, which serializes the anti-diagonals through a `lax.scan` rotation trick) on a like-for-like steady-state kernel (hardware-dependent), but pays no per-shape XLA compilation: alignment workloads expose many distinct shapes (cross-attention over length pairs is quadratic in the length range), so JAX must recompile per shape or pad heavily, and amortized over such a workload the gap reaches 10–100$\times$. d$^2$p is 10–250$\times$ faster than DeepBLAST (Hamamsy et al., 2024), scaling with problem size (its Numba CUDA kernels run the recurrence serially per batch element, so cost grows quadratically); DEDAL (Llinares-López et al., 2023) relies on TensorFlow operations rather than a fused kernel. It comes within an order of magnitude of non-differentiable GPU aligners such as CUDASW++ (Liu et al., 2013) and Accelign, which use a hard max in place of the transcendental log-sum-exp that the differentiable kernels require.

d2p employs a hierarchical parallelization strategy that maps naturally to both CPU and GPU architectures.

**Single-Instance Parallelism.** Within a single DP instance (one sequence pair), parallelism is limited by data dependencies: each cell $(i, j)$ depends on its predecessors. For grid-based algorithms, we exploit *wavefront parallelism*: all cells on anti-diagonal $k = i + j$ can be computed simultaneously since they depend only on anti-diagonals $k - 1$ and $k - 2$. For an $n \times m$ grid, this yields up to $\min(n, m)$ parallel operations per wavefront, with $n + m - 1$ sequential wavefronts.

For tree-based algorithms (CKY, Eisner), we use *span-length parallelism*: all spans of length $\ell$ can be computed in parallel since they depend only on shorter spans. For a sequence of length $n$, this yields up to $O(n)$ parallel operations per span length, with $n$ sequential levels.

**Batch Parallelism.** The primary source of parallelism is across independent DP instances in a batch. Each sequence pair is completely independent, enabling embarrassingly parallel execution.

*CPU execution:* Each DP instance is assigned to a single CPU core. With a batch of $B$ pairs and $C$ cores, we achieve $\min(B, C)$-way parallelism. Within each core, the DP proceeds sequentially along wavefronts (or span lengths), exploiting SIMD vector instructions where possible for the LSE/softmin operations.

*GPU execution:* Each DP instance is assigned to a single Streaming Multiprocessor (SM). With a batch of $B$ pairs and $S$ SMs (e.g., 108 on A100), we achieve $\min(B, S)$-way parallelism at the instance level. Within each SM, the wavefront cells are distributed across CUDA cores (e.g., 64 FP32 cores per SM), achieving fine-grained parallelism within each wavefront. For a wavefront of width $w$, we launch $w$ threads that execute in parallel.

**Memory Access Patterns.** Grid algorithms access memory in predictable patterns along anti-diagonals, enabling coalesced memory access on GPUs. We store the DP table in row-major order and access anti-diagonals via index arithmetic: $\text{idx}(i, j) = i \cdot (m + 1) + j$. For the backward pass, we traverse anti-diagonals in reverse order, maintaining the same access pattern.

Tree algorithms require more complex access patterns due to the span indexing. We use a triangular storage scheme where span $(i, j)$ maps to index $i \cdot n + j - i(i + 1)/2$, giving compact triangular storage.

**Workload Balancing.** Variable-length sequences in a batch create workload imbalance; short sequences finish before long ones. We address this through:

- *Sequence bucketing:* Grouping sequences of similar length into batches to minimize padding and balance workload.

- *Dynamic parallelism (GPU):* Long sequences can spawn additional thread blocks for their wavefronts, while short sequences use fewer resources.

- *Padding to common length:* For simplicity, we often pad all sequences in a batch to the maximum length, using masking to ignore padded positions in the loss computation.

# J. Proofs

### J.1. Proof of Theorem 2.2

*Proof.* Let $p = \text{softmax}(v/T)$ with $p_k = \exp(v_k/T)/\sum_{k'} \exp(v_{k'}/T)$.

(1) **Limit.** Let $v_{\max} = \max_k v_k$. Then

$$\text{LSE}_T(v) = v_{\max} + T \log \sum_k \exp((v_k - v_{\max})/T).$$

As $T \to 0^+$, each term $\exp((v_k - v_{\max})/T) \to 0$ for $v_k < v_{\max}$ and equals 1 for $v_k = v_{\max}$. If $v_{\max}$ is achieved by $m$ indices, $\text{LSE}_T(v) \to v_{\max} + T \log m \to v_{\max}$.

(2) **Convexity.** $\text{LSE}_T$ is the conjugate of the negative entropy, hence convex.

(3) **Gradient.** $\frac{\partial}{\partial v_k} \text{LSE}_T(v) = T \cdot \frac{\exp(v_k/T)/T}{\sum_{k'} \exp(v_{k'}/T)} = p_k$.

**(4) Hessian.** Differentiating again: $\frac{\partial^2 \text{LSE}_T}{\partial v_j \partial v_k} = \frac{1}{T}(p_k \mathbf{1}[j = k] - p_j p_k) = \frac{1}{T}[\text{Diag}(p) - pp^\top]_{jk}$. This matrix equals $\frac{1}{T}\text{Cov}_p(e_j, e_k)$ where $e_k$ are standard basis vectors under the categorical distribution $p$. It is positive semidefinite with eigenvalues $\lambda_i \in [0, 1/T]$, giving operator norm $\leq 1/T$. □

## J.2. Proof of Theorem J.2

*Proof.* By the relation $\text{smin}_T(c) = -\text{LSE}_T(-c)$:

(1) $\lim_{T \to 0} \text{smin}_T(c) = -\lim_{T \to 0} \text{LSE}_T(-c) = -\max(-c) = \min(c)$.

(2) $\nabla \text{smin}_T(c) = -\nabla_c[-\text{LSE}_T(-c)] = \nabla_c \text{LSE}_T(-c) \cdot (-1) = \text{softmax}(-c/T) = \text{softmin}(c/T)$.

(3) $\nabla^2 \text{smin}_T(c) = -\nabla^2 \text{LSE}_T(-c) = -\frac{1}{T}(\text{Diag}(q) - qq^\top)$ where $q = \text{softmin}(c/T)$. □

## J.3. Proof of Theorem 3.1

*Proof.* We prove by induction that for every node $i$,

$$\alpha_i = T \log \sum_{Y:s \to i} \exp(\text{score}(Y)/T),$$

where the sum is over all $s \to i$ paths.

**Base:** $\alpha_s = 0 = T \log \exp(0/T)$. ✓

**Inductive step:** Assume the claim holds for all parents $j \in \text{pa}(i)$.

$$
\begin{aligned}
\alpha_i &= \text{LSE}_T(\{\alpha_j + S_{ji}\}_{j \in \text{pa}(i)}) \\
&= T \log \sum_{j \in \text{pa}(i)} \exp((\alpha_j + S_{ji})/T) \\
&= T \log \sum_j \exp(S_{ji}/T) \sum_{Y:s \to j} \exp(\text{score}(Y)/T) \\
&= T \log \sum_j \sum_{Y:s \to j} \exp((\text{score}(Y) + S_{ji})/T) \\
&= T \log \sum_{Y:s \to i} \exp(\text{score}(Y)/T).
\end{aligned}
$$
□

## J.4. Proof of Theorem 3.4

*Proof.* By Theorem 3.1, $V_T(S) = T \log Z(S)$ where $Z = \sum_Y \exp(\langle Y, S \rangle/T)$. The gradient is $\nabla V_T = \frac{T}{Z}\nabla Z = \frac{1}{Z}\sum_Y Y \exp(\langle Y, S \rangle/T) = \mathbb{E}_{p_T}[Y]$. Differentiating again:

$$
\begin{aligned}
\nabla^2 V_T &= \frac{1}{T}\left(\mathbb{E}_{p_T}[YY^\top] - \mathbb{E}_{p_T}[Y]\mathbb{E}_{p_T}[Y]^\top\right) \\
&= \frac{1}{T}\text{Cov}_{p_T}(Y, Y).
\end{aligned}
$$
□

## J.5. Proof of Theorem 5.2

*Proof.* Let $\eta$ be a scalar parameter with sufficient statistic $\varphi_\eta(Y)$ (e.g., for gap-open, $f_{g_o}(Y) = N_{\text{opens}}(Y)$).

The log-partition function is $V_T(\eta) = T \log \sum_Y \exp(\text{score}(Y; \eta)/T)$ where $\text{score}(Y; \eta) = \langle S, Y_S \rangle + \eta \cdot \varphi_\eta(Y) + \dots$

By standard exponential family theory:

$$\frac{\partial V_T}{\partial \eta} = \mathbb{E}_{p_T}[\varphi_\eta(Y)],$$

$$\frac{\partial^2 V_T}{\partial S_a \partial \eta} = \frac{\partial}{\partial \eta}\mathbb{E}_{p_T}[\mathbf{1}\{a \in Y\}] = \frac{1}{T}\text{Cov}_{p_T}(\mathbf{1}\{a \in Y\}, \varphi_\eta(Y)).$$

The last equality uses the general identity for log-partition Hessian blocks. □

### J.6. Proof of Theorem 5.3

*Proof.* The marginal at position $a$ is $P_a = \mathbb{E}_{p_T}[\mathbf{1}\{a \in Y\}] = \sum_{Y \ni a} p_T(Y)$.

The Gibbs probability is $p_T(Y) = \exp((\text{score}(Y) - V_T)/T)$.

Differentiating with respect to $T$:

$$
\begin{aligned}
\frac{\partial p_T(Y)}{\partial T} &= p_T(Y) \cdot \frac{\partial}{\partial T}\left(\frac{\text{score}(Y) - V_T}{T}\right) \\
&= p_T(Y) \cdot \left(-\frac{\text{score}(Y) - V_T}{T^2} - \frac{1}{T}\frac{\partial V_T}{\partial T}\right) \\
&= p_T(Y) \cdot \left(-\frac{\text{score}(Y) - V_T}{T^2} - \frac{V_T - \mathbb{E}(\text{score})}{T^2}\right) \\
&= \frac{p_T(Y)}{T^2}(\mathbb{E}(\text{score}) - \text{score}(Y)).
\end{aligned}
$$

Therefore:

$$
\begin{aligned}
\frac{\partial P_a}{\partial T} &= \sum_{Y \ni a} \frac{p_T(Y)}{T^2}(\mathbb{E}(\text{score}) - \text{score}(Y)) \\
&= \frac{1}{T^2}\left(P_a \cdot \mathbb{E}(\text{score}) - \mathbb{E}(\text{score}(Y) \cdot \mathbf{1}\{a \in Y\})\right) \\
&= -\frac{1}{T^2}\left(\mathbb{E}(\text{score} \cdot \mathbf{1}\{a\}) - \mathbb{E}(\text{score})\mathbb{E}(\mathbf{1}\{a\})\right) \\
&= -\frac{1}{T^2}\text{Cov}_{p_T}(\mathbf{1}\{a \in Y\}, \text{score}(Y)). \qquad \square
\end{aligned}
$$

### J.7. Proof of Theorem E.1

*Proof.* We show $\text{LSE}_T$ is non-expansive in $\ell_\infty$:

$$
|\text{LSE}_T(u) - \text{LSE}_T(v)| \leq \|u - v\|_\infty.
$$

Let $\delta = \|u - v\|_\infty$. Then $v_k - \delta \leq u_k \leq v_k + \delta$ for all $k$. By monotonicity of $\text{LSE}_T$:

$$
\text{LSE}_T(v) - \delta \leq \text{LSE}_T(u) \leq \text{LSE}_T(v) + \delta.
$$

For the full DP, apply this inductively along the topological order. At each node $i$, the value $\alpha_i$ depends on predecessor values plus edge weights. If we perturb $S$ by $\delta$ in $\ell_\infty$, each $\alpha_i$ changes by at most $L_i \cdot \delta$ where $L_i$ is the longest path from $s$ to $i$. At the sink $t$, this gives $|V_T(S) - V_T(S')| \leq L_{\max}\|S - S'\|_\infty$. $\qquad \square$

### J.8. Proof of Theorem E.2

*Proof.* The gradient is $\nabla V_T = \mathbb{E}_{p_T}[Y]$, a vector of marginal probabilities. By Theorem 3.4, the Hessian satisfies $\nabla^2 V_T = \frac{1}{T}\text{Cov}_{p_T}(Y, Y)$.

For any path $Y$, $\|Y\|_2 \leq \sqrt{L_{\max}}$ (at most $L_{\max}$ edges). The covariance matrix of a bounded random variable has operator norm at most $\|Y\|_2^2 \leq L_{\max}$.

Therefore $\|\nabla^2 V_T\|_{\text{op}} \leq L_{\max}/T$, which implies $\nabla V_T$ is $L_{\max}/T$-Lipschitz by the mean value theorem. $\qquad \square$

### J.9. Proof of Theorem E.3

*Proof.* Let $\eta$ be an algorithm parameter with sufficient statistic $\varphi_\eta(Y)$ (e.g., gap count). Then $\frac{\partial V_T}{\partial \eta} = \mathbb{E}_{p_T}[\varphi_\eta(Y)]$.

Differentiating with respect to scores $S$:

$$
\frac{\partial^2 V_T}{\partial S_e \partial \eta} = \frac{1}{T}\text{Cov}_{p_T}(\mathbf{1}\{e \in Y\}, \varphi_\eta(Y)).
$$

Since $|\varphi_\eta(Y)| \leq C_\eta$ for all paths and $\mathbf{1}\{e \in Y\} \in \{0, 1\}$, the covariance is bounded by $C_\eta$. The cross-Jacobian matrix has Frobenius norm at most $\sqrt{|E|} \cdot C_\eta/T$, giving the Lipschitz bound. $\quad\square$

### J.10. Proof of Theorem E.4

*Proof.* The marginal at edge $e$ is $P_{T,e} = \frac{\partial V_T}{\partial S_e}$. By Theorem 5.2:

$$\frac{\partial P_{T,e}}{\partial \eta} = \frac{1}{T}\mathrm{Cov}_{p_T}(\mathbf{1}\{e \in Y\}, \varphi_\eta(Y)).$$

Each of the $|E|$ coordinates of $\partial P_T/\partial \eta$ is a covariance bounded by $C_\eta/T$, so the vector has $\ell_2$ norm at most $\sqrt{|E|}\,C_\eta/T$, giving Lipschitz continuity of the marginals in the parameter $\eta$. $\quad\square$

### J.11. Proof of Theorem E.8

*Proof.* **Lower bound.** For any distribution $p$ over paths, Jensen's inequality gives:

$$V_T(S) = T \log \sum_Y \exp(\mathrm{score}(Y)/T) \geq T \log \exp(\max_Y \mathrm{score}(Y)/T) = V^*(S).$$

Alternatively, $\mathrm{LSE}_T(v) \geq \max(v)$ since adding more positive terms increases the sum.

**Upper bound.** The standard log-sum-exp bound states that for $v \in \mathbb{R}^D$:

$$\max(v) \leq \mathrm{LSE}_T(v) \leq \max(v) + T \log D.$$

The upper bound follows from $\sum_k \exp(v_k/T) \leq D \cdot \exp(\max(v)/T)$.

Applying this inductively through the DAG: at each node $i$ with in-degree $d_i$, the local $\mathrm{LSE}_T$ adds at most $T \log d_i$ to the bound. Over a path of length $L$, the accumulated error is at most $T \sum_{i \in \mathrm{path}} \log d_i \leq T \cdot L \cdot \log D_{\max}$.

Taking the maximum over all paths:

$$V_T(S) \leq V^*(S) + T \cdot L_{\max} \cdot \log D_{\max}.$$

**Minimization.** For softmin, $\mathrm{smin}_T(c) = -\mathrm{LSE}_T(-c)$, so:

$$\min(c) - T \log D \leq \mathrm{smin}_T(c) \leq \min(c).$$

The bounds reverse: $D^\star - T \cdot H_{\max} \leq V_T \leq D^\star$. $\quad\square$

### J.12. Proof of Theorem E.5

*Proof.* **Maximization (convexity).** By Theorem 3.4, $\nabla^2_{SS}V_T = \frac{1}{T}\mathrm{Cov}_{p_T}(Y, Y)$. A covariance matrix is always positive semidefinite: for any vector $v$,
$$v^\top \mathrm{Cov}(Y, Y)v = \mathrm{Var}(\langle v, Y\rangle) \geq 0.$$

Hence $\nabla^2 V_T \succeq 0$, so $V_T$ is convex.

**Minimization (concavity).** For cost minimization with softmin, $V_T(c) = -\mathrm{LSE}_T(-c)$. The Hessian is $\nabla^2_{cc}V_T = -\frac{1}{T}\mathrm{Cov}_{q_T}(Y, Y) \preceq 0$, so $V_T$ is concave in costs.

Alternatively, $-V_T(c)$ is convex, which is the natural formulation for minimization (we minimize a convex function of the negative soft distance). $\quad\square$

### J.13. Proof of Theorem E.6

*Proof.* Let $\eta$ be a scalar parameter with sufficient statistic $\varphi_\eta(Y)$. The score decomposes as $\mathrm{score}(Y; \eta) = \mathrm{score}_0(Y) + \eta \cdot \varphi_\eta(Y)$.

First derivative:

$$\frac{\partial V_T}{\partial \eta} = \mathbb{E}_{p_T}[\varphi_\eta(Y)].$$

Second derivative (using the exponential family identity):

$$\begin{aligned}
\frac{\partial^2 V_T}{\partial \eta^2} &= \frac{\partial}{\partial \eta} \mathbb{E}_{p_T}[\varphi_\eta(Y)] \\
&= \frac{1}{T} \left( \mathbb{E}_{p_T}[\varphi_\eta(Y)^2] - \mathbb{E}_{p_T}[\varphi_\eta(Y)]^2 \right) \\
&= \frac{1}{T} \mathrm{Var}_{p_T}[\varphi_\eta(Y)] \geq 0.
\end{aligned}$$

The variance is always non-negative, so $V_T$ is convex in $\eta$. For minimization, $V_T = -T \log \sum_Y \exp(-\mathrm{cost}(Y; \eta)/T)$ gives $\partial^2 V_T/\partial \eta^2 = -\frac{1}{T} \mathrm{Var}_{q_T}[\varphi_\eta(Y)] \leq 0$, so $V_T$ is concave in cost parameters. $\square$

## J.14. Proof of Theorem E.7

*Proof.* The mixed partial is:

$$\frac{\partial^2 V_T}{\partial S_e \partial \eta} = \frac{\partial}{\partial \eta} P_{T,e} = \frac{\partial}{\partial \eta} \mathbb{E}_{p_T}[\mathbf{1}\{e \in Y\}].$$

Using the exponential family covariance identity:

$$\frac{\partial}{\partial \eta} \mathbb{E}_{p_T}[g(Y)] = \frac{1}{T} \mathrm{Cov}_{p_T}(g(Y), \varphi_\eta(Y))$$

with $g(Y) = \mathbf{1}\{e \in Y\}$, we obtain:

$$\frac{\partial^2 V_T}{\partial S_e \partial \eta} = \frac{1}{T} \mathrm{Cov}_{p_T}(\mathbf{1}\{e \in Y\}, \varphi_\eta(Y)).$$

This covariance can have either sign:

- **Positive**: If paths using edge $e$ tend to have higher $\varphi_\eta$ counts (e.g., a gap-heavy region where gaps and a specific match co-occur).

- **Negative**: If paths using edge $e$ tend to have lower $\varphi_\eta$ counts (e.g., matches that avoid gaps).

- **Zero**: If edge usage and parameter count are independent under $p_T$.

The mixed Hessian matrix is therefore indefinite in general, which means the joint objective in $(S, \eta)$ may have saddle points even when each marginal objective is convex/concave. $\square$

## J.15. Softmin/Softmax Duality

**Definition J.1** (Temperature-Scaled Softmin). For $T > 0$ and costs $c_1, \ldots, c_K \in \mathbb{R}$, $\mathrm{smin}_T(c_1, \ldots, c_K) = -T \log \sum_{k=1}^K \exp(-c_k/T)$.

**Lemma J.2** (Softmin Properties). $\mathrm{smin}_T$ *satisfies: (i)* $\mathrm{smin}_T \to \min$ *as* $T \to 0^+$; *(ii) gradient* $\nabla \mathrm{smin}_T(c) = \mathrm{softmin}(c/T)$ *with* $[\mathrm{softmin}(c/T)]_k = \exp(-c_k/T)/\sum_{k'} \exp(-c_{k'}/T)$; *(iii) Hessian* $\nabla^2 \mathrm{smin}_T(c) = -\frac{1}{T}(\mathrm{Diag}(q) - qq^\top)$, $q = \mathrm{softmin}(c/T)$, *negative semidefinite.*

**Theorem J.3** (Duality). *For costs $c$ and scores $s = -c$:*

1. $\mathrm{smin}_T(c) = -\mathrm{LSE}_T(-c) = -\mathrm{LSE}_T(s)$

2. $\mathrm{softmin}(c/T) = \mathrm{softmax}(-c/T) = \mathrm{softmax}(s/T)$

3. *All gradient and Hessian identities transfer with appropriate sign changes*

*Proof.* Direct verification from definitions. □

This duality allows us to implement minimization algorithms (DTW, Levenshtein, OSA, Damerau) using the same infrastructure as maximization algorithms, with negated inputs and outputs where appropriate.

