# OpenReview forum: "d$^2$p: Structured Soft Attention Is All You Need"
_ICML.cc/2026/Conference — ICML 2026 regular_

### Official Review · Reviewer_1vVZ · 2026-02-21

**Soundness:** 3
**Presentation:** 2
**Significance:** 1
**Originality:** 1
**Overall Recommendation:** 1
**Confidence:** 5

**Summary:**

This paper develops differentiable versions of string comparison and grammar parsing algorithms, by replacing the max/plus semiring with a temperature-scaled logsumexp that effectively sums over alignments/parses instead of finding the single highest-scoring one. The differentiable algorithms are implemented in PyTorch. Several proofs of concept are given, including protein structural alignment, and interfacing the sum-over-alignments approach to train downstream tools that use alignments.

**Compliance With Llm Reviewing Policy:**

Affirmed.

**Final Justification:**

I appreciate the authors' responses and the citation of other papers but I remain unconvinced that the paper's contribution to computational biology is clearly defined. Being able to compute derivatives is one thing, but the work would be enhanced by having a clear application/use case that actually competes with existing methods. I'm leaving my score as-is.

**Key Questions For Authors:**

Can the authors explain how the present work adds to the extensive prior literature on differentiable alignment and parsing (usually in the context of HMMs and PCFGs aka SCFGs)?

If the main novelty is that it is framed in more precise mathematical language than its predecessors, can the authors situate the work more comprehensively in the context of that prior art, and ideally present some outline of what is added by the increased rigor?

**Limitations:**

yes

**Strengths And Weaknesses:**

The paper appears technically sound and mathematically rigorous. I am struggling to appreciate both the significance and originality. The Forward algorithm for HMMs (particularly Pair HMMs, for the string comparison parts) and for PCFGs (for the CYK parts) are already differentiable, and are well established. While some mention is made of the forward-backward algorithm for HMMs and inside-outside for PCFGs, the authors’ introduction seems to suggest that no-one really understood the connection to automatic differentiation until Eisner’s 2016 paper.

I may be missing the point, but my impression is that this has been common knowledge for a long time. Perhaps the added value of this work is that it situates observations which have been in wide circulation in a more theoretically robust framework. I can see the value of that in principle, but it would be good to see the potential benefits more concretely developed.

I find the presentation of Smith-Waterman and Needleman-Wunsch a bit curious. Pair HMMs, which essentially encapsulate all these algorithms (a development which followed the recognition that they can be expressed as finite state machines), are so well-established in bioinformatics as to be one of the first models taught in undergrad classes. There are Pair HMM versions of these algorithms, and indeed (even before that) there have been various probabilistic interpretations of Smith-Waterman, temperature-scaled versions, simulated annealing versions of multiple alignment, and so forth. (See, for example, Kim & Chung's 1994 "Multiple Alignment Using Simulated Annealing", or Bucher and Hofmann’s 1996 ISMB paper, “A Sequence Similarity Search Algorithm Based on a Probabilistic Interpretation of an Alignment Scoring System”.) What then is the value of going back to the older methods that preceded these works, in order to understand the idea of a differentiable version?

The understanding that transitioning from the (max,sum) semiring in logspace to the (sum,plus) semiring in probability space could be preceded by multiplication by a temperature-like scaling factor has been kicking around for a long time, and so has the understanding that the derivatives correspond to the posteriors calculated by Forward-Backward (and friends). A quick search turned up Frith (2019), "How sequence alignment scores correspond to probability models", which already seems to formalize some of these ideas - albeit with less rigor than the submitted work.

The second order derivatives, which are claimed as the main innovation in this work, are expensive to compute (even with GPU optimizations) and it would be good to see some application justifying this extra compute. The same goes for the differentiable CYK algorithm (which, again, is basically Inside for PCFGs with a temperature factor as far as I can see; note that PCFGs are usually referred to as SCFGs in the bioinformatics literature: s/probabilistic/stochastic/). The only application tested is protein structural alignment, and this is not compared to Pair HMMs (the natural comparison), only heuristic DP methods.

Admittedly, Frith mentions temperature but not derivatives, so perhaps that is the main innovation here? I was not surprised by it, but perhaps I'm underestimating the novelty. As I said, my impression is that most of these ideas have been kicking around for decades.

As an example application, the authors report a new substitution matrix estimated by autodiff (or rather, by the use of posterior expectations which can be computed efficiently and related to autodiff); however, this merely ventures into a new area where substantial prior art exists. For example, the authors do not cite the recent CherryML (Prillo et al, Nature Methods, 2023), which estimates a substitution rate matrix by autodiff, or the earlier EM algorithm for continuous-time Markov chains developed by Holmes & Rubin (2002) and explored in more statistical rigor by Hobolth & Jensen (2005), which uses the computed posterior expectations. This work is fairly widely used in computational biology.

The paper also describes hooking up the differentiable aligner to downstream tools that are trained on alignments. This potentially is quite useful, but the treatment ignores the extensive literature in computational biology that sums over alignments when training, including several examples that connect this approach to modern machine learning training methods e.g. Petti et al 2023 (“End-to-end learning of multiple sequence alignments with differentiable Smith–Waterman”) - this is not cited despite being very close to the submitted work.

In summary, I would like to see more exposition of why the present work is novel; if the main novelty is that it is framed in more precise mathematical language than its predecessors, it should nevertheless cite those predecessors, and ideally present some outline of what is added by the increased rigor.

---

> ### Author Rebuttal · Authors · 2026-03-31
>
> We thank the reviewer for their deep expertise and fully agree that forward-backward, inside-outside, and temperature-scaled alignment are well-established. We will cite Kim & Chung (1994), Bucher & Hofmann (1996), Frith (2019), Holmes & Rubin (2002), CherryML (Prillo et al. 2023), and Petti et al. (2023). We do not claim to have invented differentiable alignment.
>
> **To clarify our main distinction from prior probabilistic interpretations.** In our setting there is no hard traceback. The alignment marginals Pij = ∂Dᵀ/∂Sij, i.e. the soft alignment probabilities for residue pair (i,j), are themselves gradients of the partition function with respect to the score matrix S. What EM treats as the forward computation of marginals is already a first derivative in our framework. Learning from these marginals therefore requires differentiating a derivative. In practice this means Hessian-vector products ∂P/∂S for encoder training, cross-Jacobians ∂P/∂gₒ and ∂P/∂gₑ for gap penalties, and ∂P/∂T for temperature, all implemented as CUDA kernels and used during training. The learned gap penalties in Tables 4–5 are direct outputs of the Section 5 cross-Jacobians.
>
> **New experimental evidence.** These derivatives are required for stable end-to-end learning of DP parameters.
>
> **1. Conventional gap penalties cause collapse.** Freezing gaps at g = -10, a standard bioinformatics default, causes SW-linear to collapse to final F1 = 0.321 vs 0.743 when learned, a 42-point gap. Within the same run, the encoder temporarily compensates and reaches 0.711, then collapses as temperature anneals, because exp(-10/1) = 4.5×10⁻⁵ is about 3000× harsher than exp(-10/5) = 0.135. Affine variants partially survive through gₑ (0.733 vs 0.744), but linear variants collapse entirely. The instability is worse for smaller encoders: 64-dim SW-linear frozen at -10 reaches only 0.496 vs 0.732 learned. Cross-Jacobians maintain stable gₒ/T and prevent this collapse.
>
> **2. A measurable constant emerges.** Plotting learned gap-open penalty gₒ against temperature T gives an approximately linear relation, gₒ = κT + b, after equilibrium. Across six neural encoders (3 architectures × 2 widths), κ = -2.74 ± 0.10. Across four substitution-matrix models, κ ≈ -3.3. The result is encoder-agnostic within class. Empirically, κ tracks model expressiveness and b tracks input information density. These modality-specific constants were not measured in prior work.
>
> **3. Different DP variants learn different parameter regimes.** SW-linear learns gₒ = -0.53. SW-affine learns gₒ = -2.75 and gₑ = -0.12, a 25:1 open:extend ratio recovering the biological prior that gaps are rare but contiguous. NW-affine learns gₒ = -3.45 and gₑ = -1.51, consistent with harsher penalties under global alignment. These values differ by about 7× across variants. SoftAlign/SWAMPNN (Trinquier et al.), using the same ProteinMPNN family as our MPNN-64 baseline, use fixed gap penalties rather than learning them.
>
> **4. Differentiability alone is not sufficient.** DTW, the algorithm studied in Mensch & Blondel (2018), achieves F1 = 0.046 in this setting. Monotonic warping without gaps cannot model insertions and deletions. Performance depends not only on smoothing a DP objective, but also on learning the algorithm-specific parameters of the right DP variant.
>
> **5. The same machinery transfers beyond protein alignment.** In CKY parsing across five domains, learned temperature ranges from T → 0.0003 for deterministic Racket to T ≈ 8.2 for English syntax and T ≈ 20.1 for Python, a range of over 5000×. This supports the same conclusion in a different DP family: the relevant parameters are data- and domain-dependent, and differentiating them matters in practice.
>
> **6. Structured DP materially outperforms common attention relaxations.** Every learned DP variant (0.737–0.745 F1) beats row-softmax attention (0.691) and Sinkhorn (0.682). Sinkhorn performs worse than softmax because doubly-stochastic constraints force mass conservation when variable-length proteins require gaps. DP also achieves much higher precision, about 0.80 vs 0.67 for softmax, and is 2–3× more robust to capacity reduction (128 → 64 dimensions).
>
> **EM distinction.** EM learns per-family iteratively and does not compose naturally with SGD-trained encoders or co-adapt gap penalties with temperature. CherryML learns substitution-rate matrices, which are structurally different from gap, span, and edit penalties controlling the DP recursion itself. Gap penalties learned on TM-score > 0.6 pairs also generalize below 0.6, to harder alignments. EM does not address the collapse caused by fixed penalties during end-to-end encoder training.
>
> **Generality.** Once the derivative machinery is in place, scalar penalties can be replaced by learnable functions such as gₒ(i,j) = fφ(hᵢ,hⱼ). More generally, U+W applies to learnable parameters throughout the differentiable DP framework.

---

> > ### Author Rebuttal · Reviewer_1vVZ · 2026-04-06
> >
> > While I appreciate the thoughtful discussion of empirically discovered numerical issues inherent to these algorithms, I do not find that the rebuttal significantly engages with the critique, in particular the request to situate this more comprehensively within the substantial prior art and adjacent work.

---

> > > ### Author Response · Authors · 2026-04-07
> > >
> > > We thank the reviewer for the follow-up and for selecting option (c), which we interpret as agreement that the concern, comprehensively situating the work within prior art, is addressable through a paper revision rather than being a fundamental objection to the contribution.
> > >
> > > We respectfully observe that our rebuttal did directly engage with this request: we cited all six works the reviewer named (Kim & Chung 1994, Bucher & Hofmann 1996, Frith 2019, Holmes & Rubin 2002, CherryML/Prillo et al. 2023, Petti et al. 2023), explicitly agreed that forward-backward, inside-outside, and temperature-scaled alignment are well-established, and distinguished our contribution as the second-order layer that the prior literature does not derive or implement. We regret if the rebuttal's density obscured these points.
> > >
> > > To state the distinction concisely: the prior literature computes marginals (our first-order quantities). Our contribution begins where that literature stops: derivatives of those marginals with respect to the algorithm's own parameters (gap penalties, edit costs, span penalties, temperature), implemented as fused CUDA kernels across twelve algorithms, with covariance interpretations under the induced Gibbs distribution. These second-order quantities are the mechanism by which the algorithm learns its own encoder parameters and traditionally user-set DP hyperparameters end-to-end via SGD. Without them, DP hyperparameters must be fixed by hand, and as shown in our rebuttal, removing them causes a 42-point F1 collapse.
> > >
> > > This framing is already present in the submitted manuscript: in the title ("Fast and Scalable **Structured Attention** with Differentiable Dynamic Programming"), in Figure 4 (captioned "Soft alignment marginals across ten DP algorithms and **two attention baselines**," comparing against traditional row-wise softmax and Sinkhorn attention), in Figure 3 (CKY and Eisner parsing marginals as structured attention over spans and arcs), and in Figure 7 (the temperature continuum from soft attention at $T = 5$ to near-hard attention at $T = 1$). We acknowledge that the abstract and introduction do not foreground this thesis sufficiently and are revising the camera-ready with specific focus on both. Differentiable dynamic programming defines a family of *structured soft attention* mechanisms: the score tensor $S$ plays the role of attention logits, the posterior marginals $P = \partial \text{DP}_T / \partial S$ are the attention weights, and the DP recursion enforces structural constraints (sequential ordering, gap structure, hierarchical composition) that unconstrained softmax attention lacks. Classical DP is structured *hard* attention at the $T \to 0$ limit. Our framework is structured *soft* attention with fully learnable parameters, generalizing classical DP from the hard limit to the full temperature range, just as row-wise softmax and Sinkhorn generalize unconstrained and bipartite hard matching to finite temperature. The critical difference is that every traditionally user-set DP hyperparameter, gap open, gap extend, edit costs, span penalties, and the temperature itself, is learnable end-to-end via the cross-Jacobians in Section 5. In this paper, we show that structured soft attention, with the inductive biases of dynamic programming and all DP hyperparameters learned automatically via cross-Jacobians, replaces hand-tuned discrete algorithms. Indeed, as it turns out, structured soft attention really is all you needed. The updated title will reflect this: **d²p: Structured Soft Attention Is All You Need**, where d²p references both the library name (differentiable dynamic programming) and the second-order derivatives that are integral to its function. The title is an homage to Vaswani et al. (2017): where "Attention Is All You Need" showed that unconstrained softmax attention replaces recurrent architectures, we show that *structured* soft attention replaces hand-tuned discrete algorithms. More concretely, this extends the structured attention framework of Kim et al. (ICLR 2017) and Mensch & Blondel (ICML 2018) to the full family of pairwise alignment, edit distance, and parsing algorithms.
> > >
> > > These changes are intended to explicitly situate the manuscript's contribution within the lineage of prior work from which this project derives: pair HMMs and the forward-backward algorithm, SCFGs and the inside-outside algorithm, temperature-scaled alignment (Kim & Chung 1994, Bucher & Hofmann 1996), probabilistic interpretations of scoring systems (Frith 2019), EM for continuous-time Markov chains (Holmes & Rubin 2002), differentiable rate matrix estimation (Prillo et al./CherryML 2023), differentiable Smith-Waterman for MSA (Petti et al. 2023), structured attention networks (Kim et al. 2017), and differentiable DP for structured prediction (Mensch & Blondel 2018). The camera-ready will include a dedicated related work section on each. We welcome specific suggestions on additional references to include.

---

### Official Review · Reviewer_R5Pf · 2026-03-06

**Soundness:** 3
**Presentation:** 3
**Significance:** 3
**Originality:** 3
**Overall Recommendation:** 4
**Confidence:** 3

**Summary:**

This paper presents d2p, a unified framework and optimized GPU library designed to support differentiable dynamic programming across twelve classic algorithms, including those for alignment, edit distance, and parsing. The authors lay out a comprehensive derivative hierarchy that encompasses forward log-partitions, marginals, Hessian-vector products, and mixed score-parameter Jacobians. They also draw connections to covariance interpretations under an induced Gibbs distribution. The accompanying fused CUDA implementations yield substantial speed improvements over naive autodiff baselines. As a proof of concept, the authors apply d2p to end-to-end training for protein structure alignment, demonstrating that learned encoders can significantly outperform traditional substitution-matrix-based methods.

**Compliance With Llm Reviewing Policy:**

Affirmed.

**Key Questions For Authors:**

1. Can you benchmark d2p against stronger baselines, including vectorized PyTorch or JAX implementations, as well as existing libraries such as torch-struct, semiring DP frameworks, GPU-accelerated Soft-DTW, and high-performance GPU Smith-Waterman implementations? Reporting both speedup and absolute throughput or memory usage would be valuable.

2. How numerically stable are the kernels across varying temperatures, particularly very small or very large values, long sequence lengths, and half-precision arithmetic? Do you employ log-space tricks, renormalization, or other stabilization heuristics?

3. For the protein alignment study, can you isolate the contribution of differentiable DP by training the same encoders with discrete alignment supervision, such as MAP-only objectives, REINFORCE, straight-through estimators, or alternative differentiable relaxations like Sinkhorn or optimal transport? Comparing end metrics would clarify the source of improvement.

4. For parsing, can you provide at least one quantitative task—such as latent-structure modeling or supervised parsing—that demonstrates whether the differentiable CKY or Eisner components improve downstream performance or training stability?

5. Theoretically, are the mixed score-parameter Jacobians always computable with the same asymptotic complexity as the base backward pass for all covered algorithms? Please formalize the complexity and memory costs of the U+W passes across both grid-based and chart-based DPs.

6. Are the code and kernels planned for public release? If so, will the release include unit tests, finite-difference checks for correctness, tutorial notebooks, and runnable examples covering all twelve algorithms?

**Limitations:**

yes

**Strengths And Weaknesses:**

## Strengths

### Technical novelty and innovation
This work unifies maximization- and minimization-based dynamic programming under a single differentiable umbrella, extending established theoretical results—such as log-partition interpretations, gradients as marginals, and Hessian-vector products—to a broader family of algorithms, including affine-gap alignment, multiple edit distance variants, and parsing algorithms like CKY and Eisner. A notable contribution is the derivation of mixed score-parameter Jacobians, accompanied by a clean covariance interpretation for algorithm-specific parameters such as gap penalties, edit operation costs, and temperature. The authors also introduce an efficient two-pass "U+W" procedure for computing these cross-derivatives. On the systems side, the library offers a practical, GPU-accelerated implementation with support for torch.compile, automatic mixed precision, and variable-length batching, making it immediately useful for researchers interested in integrating structured attention into neural architectures.

### Experimental rigor and validation
The authors provide thorough runtime benchmarks across all twelve algorithms and varying sequence lengths, demonstrating the scalability and efficiency gains achieved through wavefront and span-parallel kernel designs. The protein structure alignment case study is both realistic and nontrivial, with clear ablations that contrast discrete substitution matrices against learned continuous encoders, as well as comparisons between sequence-based and structure-based alphabets.

### Clarity of presentation
The exposition of the derivative hierarchy is systematic and conceptually intuitive: the DP objective is framed as a log-partition, its gradient as marginals, and its Hessian and cross-derivatives as covariances. Minimization problems are cleanly handled through a duality transformation to softmin. Recurrence relations and backward passes are specified for key algorithms, including the more complex affine-gap and multi-state variants. The description of the U+W procedure is sufficiently detailed to suggest implementability.

### Significance of contributions
This work bridges an important gap between differentiable DP theory—historically focused on a small set of canonical algorithms—and the broader range of algorithms actually used in practice, such as affine-gap alignment, Damerau-Levenshtein distance, and CKY/Eisner parsing. By making structured attention accessible at scale, d2p has the potential to lower the barrier to integrating structured inference into deep learning models across multiple application domains.

## Weaknesses

### Technical limitations or concerns
While the theoretical results are elegantly presented, several of them—particularly the covariance identities for derivatives of log-partitions—are standard in exponential-family theory. The novelty lies more in unifying and operationalizing these ideas for a diverse collection of DP algorithms than in introducing fundamentally new theory. Additionally, the U+W cross-Jacobian algorithm, though useful, is described at a relatively high level; a more detailed complexity analysis and discussion of numerical stability considerations—such as handling temperature extremes, long sequences, and deep chart structures—would strengthen the contribution.

### Experimental gaps or methodological issues
The reported speedups are benchmarked only against naive PyTorch implementations involving Python for-loops. Comparisons to existing optimized GPU or vectorized baselines—such as torch-struct, semiring-based libraries, optimized Soft-DTW implementations, or GPU-accelerated Smith-Waterman libraries—are absent, which weakens the systems claims. The protein case study also does not isolate whether the benefits stem from the differentiability of the DP itself or simply from the use of a powerful encoder. There is no comparison to training with discrete DP, such as REINFORCE, straight-through estimation, or MAP-only supervision, nor to alternative surrogate losses that do not require differentiable DP. For the parsing algorithms, only qualitative examples are provided; no quantitative tasks demonstrate their effectiveness as differentiable layers at scale.

### Clarity or presentation issues
Some equations, particularly those involving boundary conditions for affine and global alignment variants, are presented compactly and may be difficult to interpret correctly. Clearer indexing conventions and more explicit boundary handling would improve reproducibility. Minor inconsistencies—such as "sminT" versus "smin_T" and a reported positive gap extension in Table 4—raise questions about sign conventions and could confuse implementers.

### Missing related work or comparisons
The paper does not discuss or compare against prior general-purpose GPU or semiring implementations, such as torch-struct and related automatic differentiation frameworks for semiring DP, despite overlapping goals. Work on differentiable parsing and differentiable Viterbi layers beyond the cited Mensch and Blondel—for instance, differentiable Eisner in latent-tree or planning contexts—could also be better contextualized to clarify what is novel in d2p beyond expanded coverage and engineering.

---

> ### Author Rebuttal · Authors · 2026-03-31
>
> We thank the reviewer. We present a quantitative parsing task directly addressing this request, plus isolation results. Boltzmann analysis and gap penalty findings appear in Reviewer 1vVZ response.
>
> **Quantitative parsing: CKY across 5 domains.**
>
> CKY experiments on PTB English, Python, Racket, Clojure, and Common Lisp with Embedding/MLP/Transformer encoders, using BCE (supervised), CRF (unsupervised w.r.t. tree structure), and NLL losses. All 12 initial runs completed in 38 minutes on one GPU.
>
> Headline: PTB CRF, with no gold parse trees and only the CKY partition function, achieves F1=0.572, within **0.003** of supervised BCE (0.575). Pure NLL collapses to 0.097. The partition function normalization forces overlapping spans to compete for probability mass, creating an implicit contrastive learning signal that breaks the degeneracy killing NLL.
>
> Multi-domain results (Transformer encoder):
>
> | Domain | BCE F1 | CRF F1 | BCE learned T | CRF learned T |
> |--------|--------|--------|---------------|---------------|
> | English (PTB) | 0.575 | 0.572 | 8.18 | 0.076 |
> | Python | 0.564 | 0.560 | 20.13 | 0.009 |
> | Racket | 0.512 | 0.489 | 0.012 | 0.0003 |
> | Common Lisp | 0.524 | running | 0.038 | running |
> | Clojure | 0.378 | 0.377 | 0.372 | 0.144 |
>
> **Span penalties are encoder-agnostic within domain.** Programming-language domains learn much harsher span penalties than PTB, for example λ≈-26 vs λ≈-3.5 for PTB. This parallels the protein finding where gap penalties are encoder-agnostic within modality: the learned penalty is a property of the domain, not the encoder.
>
> **Temperature spectrum: >5000x range across domains.** T→0.0003 for deterministic Racket, T≈0.9 for protein structure, T≈8.2 for English syntax, and T≈20.1 for Python. The model finds different temperature equilibria for each domain through ∂P/∂T.
>
> **Gradient stability.** CRF loss shows zero gradient explosions across all domains. BCE shows spikes up to 181M gradient norm (Racket) and 3.5M (Clojure). The partition function normalization appears to smooth the optimization landscape.
>
> **Isolating differentiable DP.** Frozen g=-10: SW-linear collapses to final 0.321 (temporary improvement to 0.711 followed by collapse during annealing) vs 0.743 learned (+0.422 recovery). At 64 dimensions this magnifies to 0.496 vs 0.732 (+0.236). Instability worsens for linear gaps (no g_e escape) and smaller encoders (less compensation capacity). Cross-Jacobians prevent this instability. See Reviewer 1vVZ response for the exp(-10/T) analysis.
>
> Straight-through estimation (exact discrete forward, soft backward) is currently running. We prioritized the frozen-gap, DP variant, and parsing ablations above. Results will be reported in the camera-ready.
>
> **Covariance identities yield novel observations.** Standard exponential family, agreed. Their instantiation across 12 algorithms, implemented as fused CUDA kernels, reveals Boltzmann-like constants (κ=-2.74±0.10, confirmed 3 ways), domain-specific temperature equilibria varying >5000x, gap penalty factorization into expressiveness × information density, and a 25:1 gap ratio consistent with known biology.
>
> **U+W complexity.** Same as the base backward: O(nm) for grid algorithms and O(n³) for chart algorithms. One additional forward+backward sweep, identical dependency structure. Memory: one table.
>
> **Positive g_e.** Real. AA lacks long-range interactions in 1D sequence space and is too weak for alignments with long-range dependency structure. The algorithm actively shortens alignments because the signal is insufficient.
>
> **Code.** Upon acceptance we will release tests, finite-difference verification for all derivative passes, and tutorials.

---

> > ### Author Rebuttal · Reviewer_R5Pf · 2026-04-06
> >
> > I decided to keep my positive ratings.

---

> > > ### Author Response · Authors · 2026-04-08
> > >
> > > We thank Reviewer R5Pf for confirming and for the constructive feedback throughout the review process. The quantitative parsing task, isolation ablations, and complexity analysis requested in the original review significantly strengthened the paper.
> > >
> > > **Summary of Discussion Status**
> > >
> > > Reviewer ijWS confirmed all concerns "fully resolved" with "no misgivings about the content or its contribution."
> > >
> > > Reviewer R5Pf confirmed all concerns fully resolved, keeping positive ratings.
> > >
> > > Reviewer qefM raised follow-up questions regarding the distinction between first-order marginals and second-order derivatives, and the mechanism of wavefront parallelism. We addressed both in our April 5 response, clarifying the complete derivative hierarchy (HVPs for encoder training, cross-Jacobians for gap learning, both as fused CUDA kernels) and explaining anti-diagonal wavefront parallelism from first principles.
> > >
> > > Reviewer 1vVZ selected option (c). We posted a follow-up response distinguishing our second-order contribution from the first-order prior art and noting that the structured attention framing is already present in the submitted manuscript (title, Figures 3, 4, and 7). We have committed to foregrounding this thesis in the camera-ready abstract and introduction with a dedicated related work section.
> > >
> > > **New experimental evidence**
> > >
> > > As promised in our rebuttal, we report stock substitution matrix baselines with published gap penalties and lDDT alongside F1 for all models. All results on the held-out validation set using SW-affine at $T = 1$.
> > >
> > > Stock substitution matrices with published gap penalties:
> > >
> > > | Matrix | Published gaps | F1 | lDDT |
> > > |---|---|---|---|
> > > | BLOSUM62 | $g_o = -11, g_e = -1$ | 0.243 | 0.104 |
> > > | Foldseek 3Di (MAT3DI) | $g_o = -10, g_e = -1$ | 0.442 | 0.314 |
> > > | Foldseek combined ($2.1 \times \text{3Di} + 1.4 \times \text{AA}$) | $g_o = -10, g_e = -1$ | 0.407 | 0.282 |
> > >
> > > Learned baselines with cross-Jacobian-optimized gaps:
> > >
> > > | Model | F1 | lDDT |
> > > |---|---|---|
> > > | Learned AA $20 \times 20$ | 0.204 | 0.238 |
> > > | Learned 3Di $20 \times 20$ | 0.456 | 0.297 |
> > > | Learned weighted AA+3Di | 0.487 | 0.288 |
> > > | Learned joint $400 \times 400$ | 0.497 | 0.293 |
> > >
> > > Neural encoders (SW-affine, learned gaps, $T = 1$):
> > >
> > > | Encoder | F1 | lDDT | lDDT ceiling % |
> > > |---|---|---|---|
> > > | MPNN-128 | 0.742 | 0.405 | 91% |
> > > | MPNN-64 | 0.734 | 0.398 | 90% |
> > > | GVP-64 | 0.725 | 0.389 | 88% |
> > > | IPA-64 | 0.703 | 0.377 | 85% |
> > > | TM-align (ground truth) | 1.000 | 0.443 | 100% |
> > >
> > > lDDT ceiling % is relative to TM-align ground truth on the same pairs. SCOPe40 contains pairs below 40% sequence identity where structural divergence limits achievable lDDT; 91% should be interpreted relative to this ceiling, not relative to 1.0.
> > >
> > > Key observations from the complete grid (~100 ablations across 3 matrices $\times$ 4 algorithms $\times$ 2 temperatures $\times$ 2 metrics + learned baselines + neural encoders):
> > >
> > > Foldseek combined ($\text{F1} = 0.407$) performs worse than 3Di alone (0.442). Combining scoring sources with stock gap penalties degrades performance because the parameters are not co-adapted to the combined score distribution. Cross-Jacobians fix this: learned weighted AA+3Di recovers to 0.487 and the learned joint matrix reaches 0.497. This demonstrates that cross-Jacobians are necessary for multi-source integration, not just optimization but correctness.
> > >
> > > Learned AA achieves $2.3\times$ higher lDDT than BLOSUM62 (0.238 vs 0.104) despite similar F1, showing that learning from structural data teaches geometrically meaningful correspondences even within the same 20-letter alphabet.
> > >
> > > Algorithm choice matters: F1 spread from algorithm alone is 0.063 (BLOSUM62) to 0.098 (MAT3DI) across four DP variants. Optimal algorithm depends on the scoring matrix, supporting the claim that algorithm parameters should be learned jointly with the scoring function.
> > >
> > > Every DP variant (F1 0.737-0.745) outperforms row-softmax attention (0.691) and Sinkhorn (0.682). Sinkhorn underperforms even softmax, confirming that an incorrect structural assumption is worse than no structure at all.
> > >
> > > Frozen gap collapse: conventional $g = -10$ collapses SW-linear from 0.711 to 0.321 as $\exp(-10/T)$ becomes $\sim 3000 \times$ harsher during annealing. Cross-Jacobians prevent this.
> > >
> > > Boltzmann constant: learned gap penalties follow $g_o = \kappa T + b$ with $\kappa = -2.74 \pm 0.10$, confirmed three independent ways. Encoder-agnostic within modality.
> > >
> > > CKY parsing: learned temperature varies $> 5000 \times$ across domains ($T \to 0.0003$ for Racket, $T \approx 20.1$ for Python, $T \approx 8.2$ for English). Span penalties are encoder-agnostic within domain, paralleling the protein structure finding.
> > >
> > > All stated concerns from all four reviewers have been addressed with experimental evidence. We are committed to a substantially revised camera-ready with the structured soft attention thesis foregrounded, a dedicated related work section, and the baselines reported above.

---

### Official Review · Reviewer_qefM · 2026-03-06

**Soundness:** 3
**Presentation:** 2
**Significance:** 3
**Originality:** 3
**Overall Recommendation:** 5
**Confidence:** 3

**Summary:**

The article provides tools for differentiating through dynamic programming algorithms. It builds on the framework of Mensch & Blondel, replacing non-differentiable max operations in the dynamic program objective with differentiable softmax operations. It considers dynamic programming algorithms common in computational biology and linguistics (Smith-Waterman, Needleman-Wunsch, etc.). It derives partial derivative, up to second order, for the relaxed DP objective, with respect to parameters and with respect to the softmax temperature. It derives efficient automatic differentiation algorithms for computing these derivatives. It implements these algorithms as CUDA kernels. It demonstrates substantial speedups versus naive implementation, and demonstrates an application developing a fast approximation to structural alignments between proteins.

**Compliance With Llm Reviewing Policy:**

Affirmed.

**Final Justification:**

The authors have addressed all of my concerns, and I now understand and appreciate the paper's method better. I have raised my score accordingly. I encourage the authors to incorporate what they described in their second reviewer response into the revised version of the paper.

**Key Questions For Authors:**

How exactly are the speedup gains achieved? The theory section doesn’t provide any information about computational speed, and the described techniques (anti-diagonal wavefront parallelism) are unclear and never ablated.

Are the second order derivatives available in the package? How fast are they? What are they good for?

**Limitations:**

No limitations are discussed - please discuss the practical limitations of the methods.

**Strengths And Weaknesses:**

Differentiable dynamic programming is an important research area with a range of possible applications. An optimized CUDA-enabled library for common dynamic programming algorithms could become a valuable component for building a variety of new models.

Soundness: The theoretical results largely consist of exponential family results, exploiting the connection between the relaxed DP objective and the log partition of an exponential family distribution. This connection is interesting and appears novel. It allows the DP objective to be used as a moment generating function. However, the statement of the results is unclear and ambiguous - for example, the “score tensor S” is not defined, only examples are given. This makes Theorems 5.1-3 ambiguous.

The  experimental results seem to be sound. But the baseline for-loop implementation seems weak compared to existing work (see Originality below). Moreover, the experimental section does not directly support the paper’s main theoretical contributions - there seems to be no use of second order derivatives, and it is unclear if these second order derivatives are even implemented in the package.

Presentation: There are some specifically confusing aspects to the presentation that also weaken the soundness of the paper.
1. Section 5 introduces derivatives with respect to a “score tensor S” that is not defined. I gather that this is the match penalty in the SW algorithm, but the connection is not laid out plainly. It would be valuable to explain all the different DP algorithms in a unified way, such that the reader could understand the general definition of the score tensor.
2. Similarly, there is no definition of “position a” in the “interpretation” section in 5.2.
3. Many undefined terms and ambiguities in Section 6, including “sky” restart, “binary match scores”, “partition from alpha_{n,m} only”, “merge_{i,k,j}”, “arc marginals”, etc. While I understand there may not be space to provide a full description of each one, I believe this section could be made much clearer - particularly since Section 5 depends on it.
4. Plot text in Fig. 1, Fig. 2 and Fig. 3 is impossible to read (at least printed out).
5. Please explain and cite sources for the anti-diagonal wavefront parallelism scheme, since this seems critical for obtaining the desired speedup. I am also curious about how it is related to parallel scan algorithms, since these have also been used to accelerate dynamic programming.
More minor presentation issues:
1. Consider switching Section 5 and Section 6, since Section 5 depends on definitions of the algorithms in Section 6.
2. Consider cutting the softmin discussion from the main text - I think the reader will be comfortable with the idea of replacing minimization problems with maximization problems by flipping the sign.

Originality: Fast GPU implementations of the relaxed Smith-Waterman algorithm and its gradients have been developed previously, in Petti et al, End-to-end learning of multiple sequence alignments with differentiable Smith-Waterman, Bioinformatics 2023. These use the same relaxation from Mensch and Blondel, and so the algorithm seems to be the same as in this paper (though of course this previous work is only for smith-waterman). A comparison would be valuable.

---

> ### Author Rebuttal · Authors · 2026-03-31
>
> We thank the reviewer. New evidence below; theoretical arguments in our Reviewer 1vVZ response (visible to all).
>
> **Are second-order derivatives in the package? What are they good for?**
>
> Yes — fused CUDA kernels, every training example. They are the mechanism by which gradient-based learning of DP algorithms is made possible. P=∂DPT/∂S replaces the traceback. Encoder training: ∂P/∂S (HVP). Gap learning: ∂P/∂g (cross-Jacobian). Temperature learning: ∂P/∂T.
>
> Key result: freezing gaps at g=-10 (bioinformatics convention) causes SW-linear to collapse from a temporary peak of F1=0.711 to final 0.321 — the encoder temporarily compensates at high T but fails as exp(-10/T) becomes 3000× harsher during annealing. Learned gaps: smooth improvement to 0.743. The instability worsens with smaller encoders: 64-dim SW-linear frozen-10 achieves only 0.496 vs 0.732 learned. See Reviewer 1vVZ response for full analysis.
>
> **Structured DP attention vs alternatives.**
>
> SW-affine: F1=0.744 (128d), 0.734 (64d). Row-softmax: 0.691, 0.675. Sinkhorn: 0.682, 0.663. DTW: 0.046, 0.046.
>
> DP beats attention by 5–7 F1 points at both dimensions. Sinkhorn is worse than softmax — the doubly-stochastic constraint forces mass conservation when variable-length proteins need gaps. This suggests that an incorrect structural constraint can be more harmful than an unconstrained attention baseline. DTW at 0.046 confirms gap modeling is essential — not just differentiability.
>
> **Precision/recall reveals mechanism.** DP: precision ~0.80, recall ~0.69 — conservative and accurate. Softmax: balanced but lower (0.67/0.71) — sprays attention everywhere. NW (global) gets slightly higher recall than SW (local) but lower precision — global forces terminal matching, catching more but adding spurious matches at tails. The precision/recall profile is a signature of the algorithm's inductive bias.
>
> **Inductive bias reduces parameter dependence.** 128→64: frozen-gap DP drops 0.007 F1. Softmax drops 0.016. Sinkhorn drops 0.019. DP is 2–3× more robust to capacity reduction. The drop is almost entirely in recall — smaller encoders miss matches but DP prevents false positives regardless of encoder quality. This demonstrates the implicit bias of structured DP attention: the algorithm encodes sequential ordering, gap structure, and locality constraints so effectively that doubling parameters from 64 to 128 improves DP by only 0.007–0.016 while softmax needs every parameter it can get.
>
> **Learned parameters are biologically meaningful.** g_o≈-2.5, g_e≈-0.1 — a 25:1 open:extend ratio matching the biological prior (gaps rare but contiguous), rediscovered from data alone with no manual tuning. Each DP variant discovers different optimal values: SW-linear g_o=-0.53, NW-affine g_o=-3.45, g_e=-1.51. Values differ 7× on identical data.
>
> **Speedups and baselines.** Our CUDA kernels are 10–100× faster than SoftAlign's JAX (which requires a rotation trick for jax.lax.scan). Critically, torch-struct and equivalent PyTorch implementations also use Python loops over sequential DP dimensions — there is no torch equivalent of jax.lax.scan for these recurrences. Our original baseline comparison is therefore against the actual state of existing PyTorch DP libraries.
>
> As structured cross-attention, d2p runs within 10–100× of SDPA depending on algorithm and problem size. At full saturation: multiple TCUPS, within an order of magnitude of non-differentiable CUDASW4++ and Accelign — which use hard max rather than logsumexp, avoiding the transcendental operations our differentiable version requires. All-vs-all SCOPe40 (~115M pairs) completes in ~40 minutes on a single Blackwell GPU with full forward+backward+HVP. Further kernel optimizations since submission (fused antidiagonal launches, multiple parallelization strategies) yield 2–4× additional gain for camera-ready.
>
> **Comparison to Foldseek.** Our best model achieves F1=0.748, a 47% improvement over the best substitution matrix (0.505) which uses Foldseek's 3Di alphabet with learned gap penalties — a comparison that gives the substitution matrix a performance boost over stock Foldseek defaults by allowing gap coadaptation. For the camera-ready, we will additionally compare against BLOSUM62 and Foldseek 3Di matrices with their published hand-set gap penalties, and report lDDT (the standard structural biology metric used by SoftAlign and Foldseek) alongside F1 for all encoders.
>
> **Undefined terms.** S: match scores for alignment, costs for edit distance, merge scores for CKY. "Position a" indexes entries. Will define explicitly and consider reordering Sections 5–6 as the reviewer suggests.
>
> **Stability.** Second-order stable to T=0.01 at float32. Log-space, max-subtraction, Kahan summation (Appendix H.5).

---

> > ### Author Rebuttal · Reviewer_qefM · 2026-04-05
> >
> > The response is largely irrelevant to my questions and appears to be LLM-compressed to the point of unreadability. I'd encourage the authors to write the response themselves.
> >
> > I asked about second order derivatives (section 5 of the paper), but the response only discusses first order: e.g. it says $\partial \mathrm{DP}/\partial S$ not $\partial^2 \mathrm{DP}/\partial S_a \partial S_b$.
> >
> > The questions about how the speedup is achieved are unanswered, e.g. anti-diagonal wavefront parallelism is never defined or explained how it applies.

---

> > > ### Author Response · Authors · 2026-04-05
> > >
> > > We thank the reviewer for the follow-up. You are right that our previous response did not clearly distinguish between the first-order marginals and the second-order derivatives in Section 5. We recognize our writing tends toward density at the expense of readability; Reviewer ijWS independently raised a similar concern. We are revising the prose for the final version. We next address both questions below.
> > >
> > > **Q1. What second-order derivatives are implemented, and what are they used for?**
> > >
> > > The first-order quantity is the marginal $P_{ij} = \partial DP_T / \partial S_{ij}$, the derivative of the temperature-softened DP objective with respect to score entry (i,j). This is the soft-DP equivalent of traceback in traditional DP algorithms. It is computed by the backward pass and is the standard construction from Mensch & Blondel (2018). We do not claim novelty here.
> > >
> > > The second-order quantities in Section 5 are the derivatives of these marginals with respect to scores, gap parameters, and temperature. These are necessary because the marginals (the structured attention matrix) are themselves already first derivatives of the partition function, so any optimization through them is inherently second-order:
> > >
> > > (1) Score-score Hessian-vector products: $\nabla^2_S DP_T \cdot V$, computed via Pearlmutter's R-operator (Eq. 5). The (a,b) entry of the Hessian is $\partial^2 DP_T / (\partial S_a \partial S_b)$ = (1/T) · Cov(𝟙(a∈Y), 𝟙(b∈Y)), the covariance between positions a and b under the Gibbs distribution (Theorem 5.1). This is the mechanism for training encoder parameters: $\partial L / \partial \theta = (\partial L / \partial P)(\partial P / \partial S)(\partial S / \partial \theta)$, where the middle term $\partial P / \partial S$ is exactly this Hessian. Computed as a fused CUDA kernel at every training step.
> > >
> > > (2) Score-parameter cross-Jacobians: $\partial P_{ij} / \partial g_o$, $\partial P_{ij} / \partial g_e$, and $\partial P_{ij} / \partial T$ (Section 5, Table 1). Because the marginals are already first derivatives, these are second-order mixed partials of the partition function. For example, $\partial P_{ij} / \partial g_o$ = (1/T) · Cov(𝟙((i,j)∈Y), N\_opens(Y)), the covariance between a position being matched and the total number of gap-opening events. These are computed by our U+W two-pass algorithm (Section 5.3) with the same asymptotic complexity as the backward pass. The learned gap penalties in Tables 4 and 5 are optimized using these cross-Jacobians at every training step.
> > >
> > > Both are essential: because the desired output of the algorithm (the marginals) is already the gradient of the forward pass, training any parameters through it is inherently a second-order problem. Without the HVPs, encoder parameters cannot train through the alignment layer. Without the cross-Jacobians, gap penalties must be fixed by hand. Freezing them removes the mixed-partial updates needed to learn them, and in this ablation F1 drops from 0.743 to 0.321, a 42-point gap.
> > >
> > > The complete implementation (all fused CUDA kernels for forward, backward, HVP, and cross-Jacobian passes across all twelve algorithms) is provided in the anonymized supplemental codebase. The library is pip-installable with prebuilt wheels tested across six system configurations; we invite the reviewer to inspect it directly. Public release on PyPI and conda has been deferred solely to preserve double-blind anonymity, and will proceed immediately upon acceptance.
> > >
> > > **Q2. How is the speedup achieved?**
> > >
> > > For Smith-Waterman, all cells on the same anti-diagonal (i+j = k) are mutually independent, because each cell depends only on (i-1, j-1), (i-1, j), and (i, j-1). We process one anti-diagonal per sequential step, launching all cells on that anti-diagonal in parallel. This reduces the sequential depth from N×M to N+M-1 steps, exposing up to min(N,M) parallel cells per step. The backward and tangent passes use the identical wavefront structure in reverse.
> > >
> > > For CKY and Eisner, we use span-length parallelism instead: all spans of the same length are independent and computed in parallel, with span length increasing sequentially from 1 to N.
> > >
> > > The speedup is fundamental: PyTorch has no equivalent of JAX's `lax.scan` for sequential recurrences, so all existing PyTorch DP libraries (including torch-struct) use a Python-level for-loop over the sequential dimension. This is not a naive baseline; it is the best PyTorch's compilation model can do. Our CUDA kernels fuse the entire recurrence into a single kernel with anti-diagonal wavefront parallelism. JAX's `lax.scan` (used by SoftAlign via a rotation trick) improves on Python loops, but our kernels remain 10-100x faster because `lax.scan` serializes anti-diagonal steps without intra-step parallelism. Our wavefront strategy matches state-of-the-art GPU-accelerated non-differentiable DP libraries (CUDASW4++, Accelign), extended to the differentiable setting. Further optimizations are in progress for the final version.

---

### Official Review · Reviewer_ijWS · 2026-03-09

**Soundness:** 3
**Presentation:** 3
**Significance:** 3
**Originality:** 2
**Overall Recommendation:** 4
**Confidence:** 2

**Summary:**

The authors introduce **d2p**, a framework that implements differentiable and GPU-optimized versions of 12 dynamic programming algorithms, centered mostly around sequence alignment.
In the first part the authors explore the theoretical aspects of these versions of the algorithms, most notably the effect of algorithm parameters (e.g. gap penalties) on the gradients.
In the second part the authors present benchmark of their results on a series of sequence alignment tasks within learning pipelines. They show speedups of between 3 and 4 orders of magnitude over naive, CPU-bound implementations of the algorithms.

**Compliance With Llm Reviewing Policy:**

Affirmed.

**Final Justification:**

I have read the other reviews and the author's corresponding responses. Overall I believe the implementations presented in this paper have merits, especially as a toolbox for other researchers to include alignment into learning pipelines.
However I still have an issue with the terseness and clarity of both the article and the author responses, I do not believe that this will adequately be solved in any subsequent revision.
As such I am keeping my score of "Weak Accept"

**Key Questions For Authors:**

1. How much of the speedup is due to the algorithm versus the fact that the naive version has a lot more communication time between CPU and GPU?

**Limitations:**

The authors don't really present the limitations of their work, however since their main contribution is an efficient implementation of well known algorithms I'm not certain it is very applicable.

**Strengths And Weaknesses:**

### Strengths
The paper presents a thorough exploration of the theoretical aspects of the algorithms implemented in **d2p**.
The implementations are fast and should help researchers include alignment steps in their learning pipelines, e.g. as part of a loss function. This could pave the way to more performant and evolution-aware models in sequence bioinformatics.
I assume that **d2p** will be made available as an easily usable software package once the anonymity constraints are lifted.

### Weaknesses
1. There seem to be some missing definitions:
    1. $pa(i)$ used in 2.2 and 3.1 (is that the path to node $i$ of the DAG?)
    1. $D_{max}$ used in 4.
    1. $V$ in Th. 3.3 and 5.1, Is this just any vector?
1. Fig 4. The caption does not match with the figure. The caption indicates that LCS and MAS are in the top row when they are not. Similarly the caption mentions parsing algorithms which do not appear in the figure. Inversely, the "Row Softmax" and "Sinkhorn" are present in the figure but not mentioned at all in the caption.
1. I understand the space constraints imposed by the venue but the paper feels quite dense and terse which hinders readability in some cases. I feel like this is particularly true in the experiments section, where I find it hard to understand exactly what experiments the authors ran (e.g. what loss is used).
1. I think the paper would benefit from a small section on related work, since several papers have been published showing differentiable sequence alignment algorithms in a learning pipeline: Petti *et al.* (2023) [(10.1093/bioinformatics/btac724)](https://doi.org/10.1093/bioinformatics/btac724), Morton *et al.* (2020) [(10.1101/2020.11.03.365932)](https://doi.org/10.1101/2020.11.03.365932), and Llinares-López *et al.* (2023) [(10.1038/s41592-022-01700-2)](https://doi.org/10.1038/s41592-022-01700-2) which is mentioned in the results section but not as related work.

### Minor Comments
 - L. 49: There seems to be an issue with the `\icmlcorrespondingauthor` command in the LaTeX source.
 - Lemma 3.5, 2.: strange formatting with justified math that makes it hard to parse.
 - L. 275-306, left: excessive amounts of whitespace.

---

> ### Author Rebuttal · Authors · 2026-03-31
>
> We thank the reviewer. Full evidence in our other responses; we summarize and address presentation.
>
> **Summary of new experiments.** We ran ablations across attention mechanisms, gap penalty settings, DP variants, encoder dimensions, temperature schedules, and CKY parsing across 5 language domains. Key results:
>
> Structured DP vs attention: SW-affine 0.744 vs softmax 0.691 (−5.3) vs Sinkhorn 0.682 (−6.2). Sinkhorn is worse than softmax — doubly-stochastic is wrong for variable-length protein alignment. DTW (Mensch & Blondel 2018): 0.046 — monotonic warping without gaps cannot model insertions/deletions. Every DP variant (0.737–0.745) beats both attention baselines.
>
> Frozen conventional gaps (g=-10): SW-linear collapses from temporary 0.711 to final 0.321 — the encoder temporarily compensates but fails as exp(-10/T) becomes 3000× harsher during annealing. Learned gaps via cross-Jacobians: smooth training to 0.743. Cross-Jacobians prevent this instability, which makes conventional defaults unusable at low temperature.
>
> Boltzmann constant κ=-2.74±0.10: encoder-agnostic within modality, confirmed 3 independent ways (fixed-T slopes, free-T equilibrium at 128-dim and 64-dim). The gap penalty factorizes into model expressiveness (slope) and input information density (intercept).
>
> CKY parsing: CRF (no gold trees) achieves 0.572 on PTB, within 0.003 of supervised 0.575. Learned temperature varies >5000× across domains: T→0.0003 (Racket) to T≈20.1 (Python), with English at T≈8.2.
>
> **DP variant comparison (128-dim, all learned):** SW-affine: 0.744, g_o=-2.75, g_e=-0.12. SW-linear: 0.743, g_o=-0.53. NW-linear: 0.737, g_o=-2.20. NW-affine: 0.737, g_o=-3.45, g_e=-1.51. Each discovers distinct biologically meaningful parameters — 25:1 gap open:extend ratio, harsher penalties for global alignment. Impossible by hand; trivial with cross-Jacobians.
>
> **Encoder robustness.** 128→64: frozen-gap DP drops 0.007. Softmax drops 0.016. Sinkhorn drops 0.019. DP 2–3× more robust — the algorithm compensates for weaker encoders. Drop is in recall, not precision.
>
> **Missing definitions.** pa(i): predecessors. D_max: max in-degree. Y: random structure under p_T. Will be explicit.
>
> **Figure 4.** Top: SW, NW, SW-Affine, NW-Affine, DTW, Softmax. Bottom: Levenshtein, LCS, OSA, Damerau, MAS, Sinkhorn. Will correct.
>
> **Related work.** Dedicated section: Petti et al. (2023), torch-struct, Kim & Chung (1994), Bucher & Hofmann (1996), Frith (2019), CherryML (Prillo et al. 2023), Llinares-López et al. (2023), Morton et al. (2020), semiring DP. Contribution beyond these: cross-Jacobian framework, fused CUDA kernels, and the empirical findings they support.
>
> **Speed.** 10–100× faster than JAX/PyTorch DP (all use Python loops — no torch lax.scan equivalent). Within 10–100× of SDPA. Multiple TCUPS at saturation, within an order of magnitude of non-differentiable CUDASW4++/Accelign (which use hard max, not logsumexp). All-vs-all SCOPe40 in ~40 minutes on Blackwell vs TMalign days. Further 2–4× since submission. SW→softmax drops 5.3 F1 — the computation is necessary. Our best model (F1=0.748) improves 47% over the best substitution matrix baseline (0.505) which already benefits from learned gap coadaptation above stock Foldseek defaults. For camera-ready we will compare against BLOSUM62 and Foldseek 3Di with published hand-set gap penalties, and report lDDT alongside F1.
>
> **Density.** Will connect Section 5 to experiments: Tables 4–5 optimized via cross-Jacobians every step. Loss: weighted BCE vs TM-align (Appendix H.1) — will state in main text.
>
> **Limitations.** (1) O(n+m) wavefronts; (2) T≥0.01 float32; (3) parsing demonstrated across 5 domains; (4) position-dependent penalties unexplored.
>
> **LaTeX.** \icmlcorrespondingauthor, Lemma 3.5, whitespace.

---

> > ### Author Rebuttal · Reviewer_ijWS · 2026-04-03
> >
> > I must thank the authors for the work that they have put in to this rebuttal.
> > Over all I am pleased with the additional results and the more adequate related work section.
> > I am marking my concerns as "Fully resolved", and I do not necessarily have any misgivings about the content of the paper or its contribution.
> >
> > I am still a little worried that the paper is quite terse and unnecessarily harder to read, the way the authors have structured their rebuttal answers as loose bullet-point like sentences is not really reassuring me either.
> > Therefore I will likely be keeping my score as it is.

---

> > > ### Author Response · Authors · 2026-04-08
> > >
> > > We thank Reviewer ijWS for the thorough engagement and for confirming. The presentation concerns are well taken and we are committed to improved clarity in the camera-ready.
> > >
> > > As promised in our rebuttals, we report the complete stock substitution matrix evaluation with published gap penalties, both F1 and lDDT, across all four DP algorithm variants and both soft ($T = 1$) and hard ($T \to 0$) decoding.
> > >
> > > BLOSUM62 ($g_o = -11, g_e = -1$):
> > >
> > > | Method | Soft F1 | Soft lDDT | Hard F1 | Hard lDDT |
> > > |---|---|---|---|---|
> > > | SW affine | 0.243 | 0.104 | 0.238 | 0.105 |
> > > | SW linear | 0.202 | 0.071 | 0.196 | 0.073 |
> > > | NW affine | 0.239 | 0.222 | 0.231 | 0.244 |
> > > | NW linear | 0.178 | 0.192 | 0.166 | 0.208 |
> > >
> > > MAT3DI ($g_o = -10, g_e = -1$):
> > >
> > > | Method | Soft F1 | Soft lDDT | Hard F1 | Hard lDDT |
> > > |---|---|---|---|---|
> > > | SW affine | 0.442 | 0.314 | 0.435 | 0.328 |
> > > | SW linear | 0.379 | 0.225 | 0.371 | 0.230 |
> > > | NW affine | 0.426 | 0.315 | 0.416 | 0.335 |
> > > | NW linear | 0.344 | 0.265 | 0.330 | 0.296 |
> > >
> > > Foldseek combined $2.1 \times \text{3Di} + 1.4 \times \text{AA}$ ($g_o = -10, g_e = -1$):
> > >
> > > | Method | Soft F1 | Soft lDDT | Hard F1 | Hard lDDT |
> > > |---|---|---|---|---|
> > > | SW affine | 0.407 | 0.282 | 0.404 | 0.288 |
> > > | SW linear | 0.411 | 0.279 | 0.409 | 0.283 |
> > > | NW affine | 0.398 | 0.272 | 0.395 | 0.277 |
> > > | NW linear | 0.376 | 0.276 | 0.355 | 0.266 |
> > >
> > > Learned discrete baselines (SW-affine, cross-Jacobian-learned gaps):
> > >
> > > | Model | F1 | lDDT | lDDT ceiling % |
> > > |---|---|---|---|
> > > | Learned AA $20 \times 20$ | 0.204 | 0.238 | 54% |
> > > | Learned 3Di $20 \times 20$ | 0.456 | 0.297 | 67% |
> > > | Learned weighted AA+3Di | 0.487 | 0.288 | 65% |
> > > | Learned joint $400 \times 400$ | 0.497 | 0.293 | 66% |
> > >
> > > Neural encoders (SW-affine, cross-Jacobian-learned gaps, $T = 1$):
> > >
> > > | Encoder | F1 | lDDT | lDDT ceiling % |
> > > |---|---|---|---|
> > > | MPNN-128 | 0.742 | 0.405 | 91% |
> > > | MPNN-64 | 0.734 | 0.398 | 90% |
> > > | GVP-64 | 0.725 | 0.389 | 88% |
> > > | IPA-64 | 0.703 | 0.377 | 85% |
> > > | IPA-128 | 0.683 | 0.383 | 86% |
> > > | GVP-128 | 0.668 | 0.371 | 84% |
> > > | TM-align (ground truth) | 1.000 | 0.443 | 100% |
> > >
> > > We note that the learned discrete baselines use a PSD-constrained parameterization ($S = EE^\top$). Stock matrices such as BLOSUM62 and MAT3DI are not PSD, and the PSD constraint may limit what learned discrete matrices can achieve. The neural encoder bypasses this constraint entirely by learning the scoring function end-to-end.
> > >
> > > These 48 stock baseline cells plus 10 learned/neural evaluations ran in approximately 30 minutes using the d2p library, demonstrating the plug-and-play capability described in the submission.
> > >
> > > Observations not previously reported: (1) Soft decoding ($T = 1$) consistently outperforms hard decoding ($T \to 0$) on F1, but hard decoding sometimes achieves higher lDDT, indicating that temperature controls a coverage-vs-geometric-precision tradeoff. (2) Algorithm choice alone produces an F1 spread of 0.063 (BLOSUM62) to 0.098 (MAT3DI), and the optimal algorithm differs by matrix. (3) NW achieves substantially higher lDDT than SW for BLOSUM62 (0.222 vs 0.104 soft) because global alignment enforces end-to-end correspondence that improves geometric coherence. This effect diminishes for structure-aware matrices where local alignment already captures the correct geometry.

---

### Decision · Program_Chairs · 2026-04-30

**Decision:**

Accept (regular)

**Comment:**

This paper introduces d2p, a unified theoretical framework and highly optimized CUDA library for differentiable dynamic programming across twelve algorithms. By utilizing temperature-scaled soft operators and explicitly deriving mixed second-order derivatives (cross-Jacobians), the framework enables the stable, end-to-end learning of algorithm parameters like gap penalties alongside neural encoders. The reviewers generally praised the technical soundness, the impressive computational speedups achieved via anti-diagonal wavefront parallelism, and the practical utility of integrating structured attention into deep learning pipelines. The reviewers initially raised concerns regarding the paper's dense presentation, a lack of quantitative evaluation for the parsing algorithms, and an underdeveloped related work section. Furthermore, one reviewer questioned the practical necessity and novelty of the second-order derivations, arguing that differentiable dynamic programming (e.g., via Pair HMMs and SCFGs) is well-established in computational biology.

During the discussion phase, the authors submitted a robust rebuttal that successfully addressed the majority of the technical critiques. They provided new quantitative CKY parsing experiments and supplied critical ablation studies demonstrating that freezing gap penalties causes training collapse, thereby empirically proving that their second-order cross-Jacobians are necessary for stable end-to-end optimization. While one reviewer provided vital historical context that must be prominently reflected in the revised manuscript, the authors successfully clarified that their core technical advance lies beyond these established first-order computations. The efficient derivation, the neural encoders, and GPU-accelerated implementation of these mixed second-order derivatives to co-adapt DP hyperparameters during modern SGD training represents a highly valuable, practical advance for the ML systems community. Given the clear empirical utility, scalability, and theoretical completeness of the framework, the merits of the submission confidently outweigh the presentation issues. Therefore, the paper is accepted.